# Label Distribution Shift-Aware Prediction Refinement for Test-Time Adaptation

**Minguk Jang**                                                                                      *mgjang@kaist.ac.kr*
*School of Electrical Engineering, KAIST*

**Hye Won Chung**                                                                                    *hwchung@kaist.ac.kr*
*School of Electrical Engineering, KAIST*

**Reviewed on OpenReview:** *https://openreview.net/forum?id=c7AAHdEYz5*

## Abstract

Test-time adaptation (TTA) is an effective approach to mitigate performance degradation of trained models when encountering input distribution shifts at test time. However, existing TTA methods often suffer significant performance drops when facing additional class distribution shifts. We first analyze TTA methods under label distribution shifts and identify the presence of class-wise confusion patterns commonly observed across different covariate shifts. Based on this observation, we introduce *label Distribution shift-Aware prediction Refinement for Test-time adaptation (DART)*, a novel TTA method that refines the predictions by focusing on class-wise confusion patterns. DART trains a prediction refinement module during an intermediate time by exposing it to several batches with diverse class distributions using the training dataset. This module is then used during test time to detect and correct class distribution shifts, significantly improving pseudo-label accuracy for test data. Our method exhibits 5-18% gains in accuracy under label distribution shifts on CIFAR-10C, without any performance degradation when there is no label distribution shift. Extensive experiments on CIFAR, PACS, OfficeHome, and ImageNet benchmarks demonstrate DART's ability to correct inaccurate predictions caused by test-time distribution shifts. This improvement leads to enhanced performance in existing TTA methods, making DART a valuable plug-in tool.

## 1 Introduction

Deep learning has achieved remarkable success across various tasks, including image classification (Krizhevsky et al., 2012; Radford et al., 2021; Simonyan & Zisserman, 2014) and natural language processing (Devlin et al., 2018; Vaswani et al., 2017). However, these models often suffer from significant performance degradation when there is a substantial shift between the training and test data distributions (Mendonca et al., 2020; Saenko et al., 2010; Taori et al., 2020). Test-time adaptation (TTA) methods (Boudiaf et al., 2022; Goyal et al., 2022; Jang et al., 2022; Wang et al., 2020; Zhao et al., 2022) have emerged as a prominent solution to mitigate performance degradation due to distribution shifts. TTA methods allow trained models to adapt to the test domains using only unlabeled test data. In particular, an approach known as BNAdapt (Nado et al., 2020; Schneider et al., 2020), which substitutes the batch statistics in the batch normalization (BN) layers of a trained classifier with those from the test batch, has proven to be effective in adapting to input distribution shifts in scenarios such as image corruption. Consequently, numerous TTA methods (Lee, 2013; Wang et al., 2020; Zhao et al., 2022; Zhou et al., 2023) are built upon the BNAdapt strategy.

However, recent studies (Gong et al., 2022; Niu et al., 2023; Zhou et al., 2023) have shown that the effectiveness of the BNAdapt strategy diminishes or even becomes harmful when both the class distribution and the input distribution in the test domain shift at test time. This reduction in effectiveness is due to BNAdapt's reliance on batch-level data statistics, which are influenced not only by class-conditional input distributions but

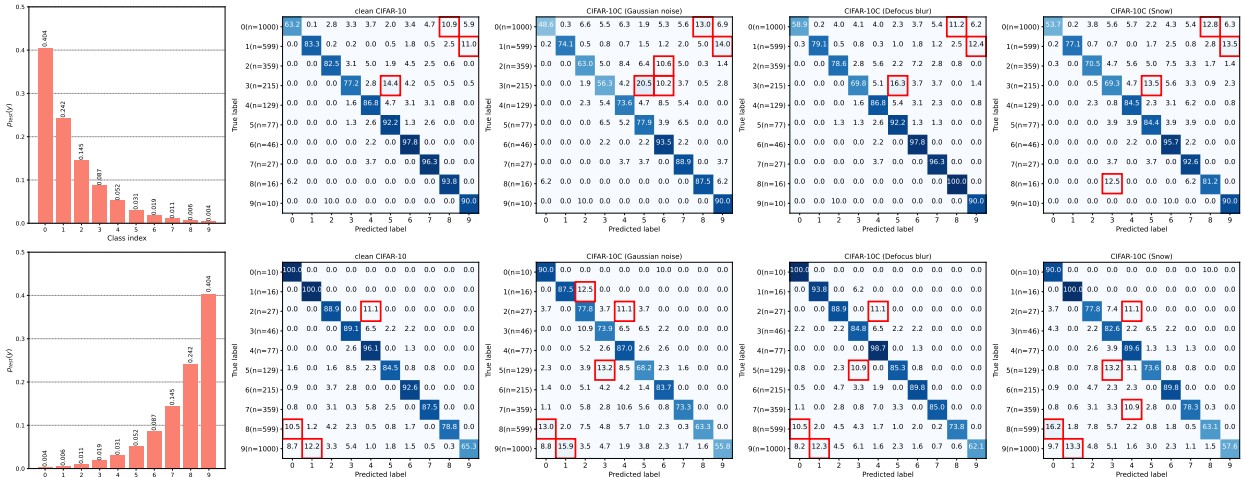

Figure 1: **Confusion patterns of BN-adapted classifier due to test-time label distribution shifts**. We present class-wise confusion matrices of BN-adapted classifiers, initially trained on class-balanced CIFAR-10 and then tested on CIFAR-10C with two long-tailed distributions (first column). The second column shows confusion patterns on the CIFAR-10 test dataset with only label shifts, while the third to fifth columns display patterns on CIFAR-10C under three types of corruptions combined with label shifts. Class pairs where the confusion rate exceeds 11% are highlighted in red. There is significant accuracy degradation in head classes (e.g., class 0 in the 1st row and class 9 in the 2nd row). Additionally, similar class-wise confusion patterns are observed across different types of corruption for each label distribution shift (each row). Confusion matrices for other 15 types of corruption and class imbalance ratios of $\rho = 1, 10, 100$ are also reported in Figure 9–11.

also by the configuration of data within the test batches. For instance, when a test batch predominantly contains samples from a few head classes, the batch statistics become biased towards these classes. To address these limitations, some recent methods have been designed to lessen the dependency of test-time adaptation strategies on batch-level statistics and to tackle class distribution shifts with additional adjustments. For instance, NOTE (Gong et al., 2022) employs instance-aware batch normalization, which diverges from traditional batch normalization, and uses a prediction-balanced memory to simulate a class-balanced test data stream. SAR (Niu et al., 2023) implements batch-agnostic normalization layers, such as group or layer norm, complemented by techniques designed to drive the model toward a flat minimum, enhancing robustness against noisy test samples. However, these methods still rely on pseudo labels of test samples, and their effectiveness is fundamentally limited by the accuracy of these pseudo labels, which particularly deteriorates under severe label distribution shifts.

In this work, we propose a more direct, simple yet effective method that can correct the inaccurate predictions generated by the BNAdapt strategy under label distribution shifts. Our method can be integrated with a variety of existing TTA methods (Boudiaf et al., 2022; Gong et al., 2022; Lee, 2013; Niu et al., 2023; Schneider et al., 2020; Wang et al., 2020; Zhao et al., 2022; Zhou et al., 2023), significantly enhancing their accuracies by 5-18 percentage points under label distribution shifts, while having minimal impact when label distributions remain unchanged. We achieve this by first identifying the existence of *class-wise confusion patterns*, i.e., specific misclassification trends between classes that the BNAdapt classifier exhibits under label distribution shifts, regardless of different input corruption patterns (Fig. 1). In particular, we show that these confusion patterns depend not only on the inter-class relationships but also on the magnitude and direction of the label distribution shifts from a class-balanced training dataset. Building on this insight, it might seem natural to correct the BN-adapted classifier's inaccurate predictions by applying a simple affine transformation that reverses the effects of class-wise confusion as in learning with label noise (Natarajan et al., 2013; Patrini et al., 2017; Zhu et al., 2021). However, a significant challenge arises because these class-wise confusion patterns are difficult to deduce using only the unlabeled test data, which has unknown label distributions. Additionally,

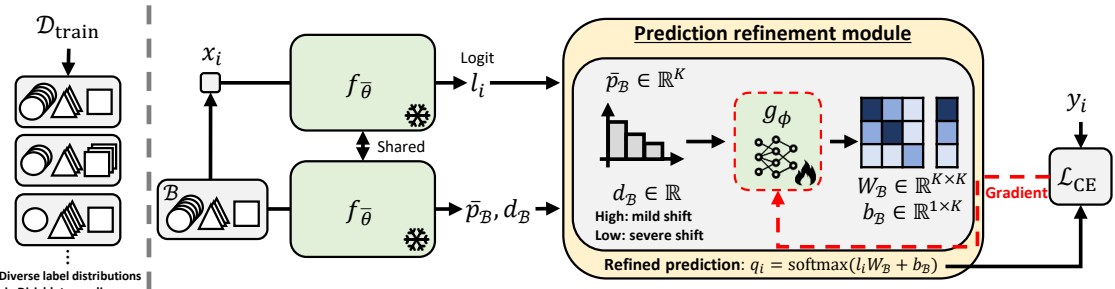

Figure 2: **Intermediate time training of DART.** At intermediate time, the period between the training and test times, DART trains a prediction refinement module $g_\phi$ to correct the inaccurate prediction caused by the class distribution shifts. *(left)* By sampling the training data from Dirichlet distributions, we generate batches with diverse class distributions. *(right)* The prediction refinement module $g_\phi$ takes the averaged pseudo label distribution $\bar{p}_{\mathcal{B}}$ and prediction deviation $d_{\mathcal{B}}$ for each batch $\mathcal{B}$, and outputs a square matrix $W_{\mathcal{B}}$ and a vector $b_{\mathcal{B}}$ of size $K$ (class numbers). Using labels of the training data, we optimize $g_\phi$ to minimize the cross-entropy loss between labels $y$ and the refined prediction $q = \mathrm{softmax}(f_{\bar{\theta}}(x)W_{\mathcal{B}} + b_{\mathcal{B}})$ for samples $(x, y) \in \mathcal{B}$ for the BN-adapted classifier $f_{\bar{\theta}}$.

these patterns can vary over time with online label distribution shifts, further complicating the adaptation process.

To tackle these challenges, we propose a novel method, named *label Distribution shift-Aware prediction Refinement for Test-time adaptation (DART)*, which detects test-time label distribution shifts and corrects the inaccurate predictions of BN-adapted classifier through an effective affine transformation depending on the label distribution shifts. Our key insight is that the model can learn how to adjust inaccurate predictions caused by label distribution shifts by experiencing several batches with diverse class distributions using the labeled training dataset before the start of test time. DART trains a prediction refinement module during an *intermediate time*, positioned between the end of training and the start of testing, by exposing multiple batches of labeled training data with diverse class distributions, sampled from the Dirichlet distribution (Figure 2). The module uses two inputs to detect label distribution shifts: the averaged pseudo-label distribution of each batch generated by the BN-adapted classifier and a new metric called *prediction deviation*, which measures average deviations of each sample's soft pseudo-label from uniformity to gauge the confidence of predictions made by the BN-adapted classifier. The module is trained to generate two outputs, a square matrix and a vector of class dimensions, which together transform the logit vector for prediction refinement. Since this module requires only soft pseudo-labels for test samples, it can be readily employed during test time using the pre-trained BN-adapted model. This approach enables DART to dynamically adapt to label distribution shifts at test time, enhancing the accuracy of predictions.

We evaluate the effectiveness of DART across a wide range of TTA benchmarks, including CIFAR-10/100C, ImageNet-C, CIFAR-10.1, PACS, and OfficeHome. Our results consistently demonstrate that DART significantly improves prediction accuracy across these benchmarks, particularly in scenarios involving test-time distribution shifts in both input and label distributions. This enhancement also boosts the performance of existing TTA methods, establishing DART as a valuable plug-in tool (Table 1). Specifically, DART achieves notable improvements in test accuracy, enhancing the BNAdapt method by 5.7% and 18.1% on CIFAR-10C-LT under class imbalance ratios of $\rho = 10$ and 100, respectively. Additionally, our ablation studies highlight the critical role of the prediction refinement module's design, particularly its inputs and outputs, in enhancing DART's effectiveness.

## 2 Label Distribution Shifts on TTA

TTA methods are extensively investigated to mitigate input data distribution shifts, yet the impact of label distribution shifts on these methods is less explored. In this section, we theoretically and experimentally

analyze the impact of label distribution shifts on BNAdapt (Nado et al., 2020; Schneider et al., 2020), a foundational strategy for many TTA methods (Lee, 2013; Wang et al., 2020; Zhao et al., 2022; Zhou et al., 2023). We also introduce a new metric designed to detect label distribution shifts at test time using only unlabeled test samples.

**Impact of label distribution shifts on TTA**   We begin with a toy example to explore the impact of test-time distribution shifts on a classifier trained for a four-class Gaussian mixture distribution. We assume that the classifier is adapted at test time by a mean centering function, modeling the effect of batch normalization in the BNAdapt. Consider the input distribution for class $i$ as $\mathcal{N}(\mu_i, \sigma^2 I_2)$, where $\mu_1 = (d, \beta d), \mu_2 = (-d, \beta d), \mu_3 = (d, -\beta d)$, and $\mu_4 = (-d, -\beta d)$ with $d \in \mathbb{R}^2$ and $\beta > 1$. Assume uniform priors $p_{\mathrm{tr}} = [1/4, 1/4, 1/4, 1/4]$ for training. At test time, we consider shifts in both input and label distributions. Consider the covariate shift by $\Delta \in \mathbb{R}^2$ as studied in prior works (Stojanov et al., 2021; Yi et al., 2023). This changes the input distribution for each class to $\mathcal{N}(\mu_i + \Delta, \sigma^2 I_2)$. Additionally, let the class distribution shift to $p_{\mathrm{te}} = [p, 1/4, 1/4, 1/2 - p]$ for some $p \in [1/4, 1/2)$. Let $h(\cdot)$ denote a Bayes classifier for the training distribution, and Norm($\cdot$) represent a mean centering function, mimicking the batch normalization at test time. When $p = 1/4$ (i.e., $p_{\mathrm{tr}} = p_{\mathrm{te}}$), the mean centering effectively mitigates the test-time input distribution shift, restoring the original performance of $h(\cdot)$. However, when $p > 1/4$ (i.e., $p_{\mathrm{tr}} \neq p_{\mathrm{te}}$), the BN-adapted classifier, modeled by $\bar{h}(\cdot) := h(\mathrm{Norm}(\cdot))$, begins to exhibit performance degradation with a distinct class-wise confusion pattern. This pattern reflects both the spatial distances between class means and the severity of the label distribution shifts:

(#1) The misclassification probability from the major class (class 1) to a minor class is higher than the reverse: $\mathrm{Pr}_{x \sim \mathcal{N}(\mu_1 + \Delta, \sigma^2 I_2)}[\bar{h}(x) = i] > \mathrm{Pr}_{x \sim \mathcal{N}(\mu_i + \Delta, \sigma^2 I_2)}[\bar{h}(x) = 1], \forall i \neq 1$.

(#2) The misclassification patterns are influenced by the spatial relationships between classes, e.g., the probability of misclassifying from class 1 to other classes is higher for those that are spatially closer: $\mathrm{Pr}_{x \sim \mathcal{N}(\mu_1 + \Delta, \sigma^2 I_2)}[\bar{h}(x) = 2] > \mathrm{Pr}_{x \sim \mathcal{N}(\mu_1 + \Delta, \sigma^2 I_2)}[\bar{h}(x) = 3] > \mathrm{Pr}_{x \sim \mathcal{N}(\mu_1 + \Delta, \sigma^2 I_2)}[\bar{h}(x) = 4]$.

(#3) As the class distribution imbalance $p > 1/4$ increases, the rate at which misclassification towards spatially closer classes increases is greater than towards more distant classes: $\frac{\partial}{\partial p} \mathrm{Pr}_{x \sim \mathcal{N}(\mu_1 + \Delta, \sigma^2 I_2)}[\bar{h}(x) = 2] > \frac{\partial}{\partial p} \mathrm{Pr}_{x \sim \mathcal{N}(\mu_1 + \Delta, \sigma^2 I_2)}[\bar{h}(x) = 3]$.

More additional explanations for these confusion patterns including the symbol table, illustrations, and detailed proofs are described in Appendix D. Through experiments with real datasets, we reaffirm the impact of label distribution shifts on a BN-adapted classifier. We consider a BN-adapted classifier originally trained on a class-balanced CIFAR-10 dataset (Krizhevsky & Hinton, 2009) and later tested using CIFAR-10-LT and CIFAR-10C-LT test datasets, which include label distribution shifts. CIFAR-10C (Hendrycks & Dietterich, 2019) serves as a benchmark for assessing model robustness against 15 different predefined types of corruptions, such as Gaussian noise. To analyze the effects of label distribution shifts, we define the number of samples for class $k$ as $n_k = n(1/\rho)^{k/(K-1)}$, where $n$ is the number of samples for the head class and $\rho$ is the class imbalance ratio. In Fig. 3, we observe substantial performance drops in test accuracy for BNAdapt (orange) as the class imbalance ratio $\rho$ increases. The BNAdapt exhibits even worse performance than NoAdapt (blue) when $\rho = 100$.

In Fig. 1, we provide more detailed analysis for misclassification patterns by presenting confusion matrices illustrating the impact of two distinct long-tailed distributions (each row) across four types of image corruptions (each column, including clean, Gaussian noise, Defocus blur, and Snow). These matrices show the fraction of test samples from class $i$ classified into class $j$. The trends observed in these real experiments are similar to those analyzed in our toy example. Firstly, significant accuracy degradation is evident, especially for head classes (with smaller/larger class indices in the 1st/2nd row, respectively). Secondly, the confusion patterns remain consistent across different corruption types and reflect class-wise relationships; for example, more frequent confusion occurs between classes with similar characteristics, such as airplane & ship (0 and 8), automobile & truck (1 and 9), and cat & dog (3 and 5). Additionally, we present confusion matrices for CIFAR-10C-LT with various levels of label distribution shifts ($\rho = 1, 10, 100$) in Figures 9-11 of Appendix F, revealing increasingly pronounced class-wise confusion patterns as the imbalance ratio $\rho$ rises from 1 to 100.

These observations suggest the potential to correct the inaccurate predictions of the BN-adapted classifier by reversing the effects of class-wise confusion. Similar problem has been considered in robust model training with label noise (Natarajan et al., 2013; Patrini et al., 2017; Zhu et al., 2021), where attempts have been made to adjust model outputs using an appropriate affine transformation, reversing the estimated label noise patterns in training dataset. However, a new challenge arises in TTA scenarios where only unlabeled test data with unknown (online) label distributions are available, which hinders the accurate estimation of class-wise confusion patterns. In particular, a commonly used label correction scheme in robust learning settings (Zhu et al., 2021) fails to estimate the confusion matrix resulting from label distribution shifts in TTA scenarios, as shown in Appendix H. To address these challenges, we next propose a new metric to detect label distribution shifts and correct the inaccurate predictions from BN-adapted classifiers.

**A new metric to detect label distribution shifts in TTA** During test time, although we only have access to unlabeled test samples, we can obtain pseudo soft labels for these samples using the BN-adapted pre-trained classifier. A common metric for detecting label distribution shifts is the average pseudo-label distribution of the test samples, as used in previous studies (Park et al., 2023; Zhao et al., 2022; Zhou et al., 2023). However, we find that relying solely on this metric is insufficient for accurately detecting label distribution shifts. To illustrate this, we use the CIFAR-10C-imb dataset (Niu et al., 2023), which models online label distribution shifts with varying severity. CIFAR-10C-imb consists of 10 subsets, each with a class distribution defined by $[p_1, p_2, \ldots, p_K]$ where $p_k = p_{\max}$ and $p_i = p_{\min} = (1 - p_{\max})/(K - 1)$ for $i \neq k$. The imbalance ratio (IR) is defined as $\text{IR} = p_{\max}/p_{\min}$. Each subset has 1,000 samples, totaling 10,000 for the test set (detailed in Appendix A). If the average pseudo-label distribution accurately estimates the shift from a uniform distribution, the distance $D(u, \bar{p}_{\mathcal{B}})$ (using cross-entropy) between the uniform distribution $u$ and the average pseudo-label distribution $\bar{p}_{\mathcal{B}} := \frac{1}{|\mathcal{B}|} \sum_{(x,\cdot) \in \mathcal{B}} \text{softmax}(f_{\bar{\theta}}(x))$, calculated for each test batch $\mathcal{B}$ using the BN-adapted classifier $f_{\bar{\theta}}$, should increase with IR. However, as shown in Figure 4 (left), $D(u, \bar{p}_{\mathcal{B}})$ increases from IR 1 (blue points) to 20 (green) but decreases again for more severe shifts (e.g., IR > 50), even though test accuracy continues to drop. This indicates that the average pseudo-label distribution $\bar{p}_{\mathcal{B}}$ alone is inadequate for distinguishing between no shifts (IR 1) and severe shifts (IR 5000). To address this limitation, we introduce a new metric, *prediction deviation (from uniformity)* $d_{\mathcal{B}} = \frac{1}{|\mathcal{B}|} \sum_{(x,\cdot) \in \mathcal{B}} D(u, \text{softmax}(f_{\bar{\theta}}(x)))$, which measures the average confidence of predictions for each batch. Figure 4 (right) shows that as IR increases, both prediction deviation and test accuracy decrease, effectively quantifying the severity of label distribution shifts. Furthermore, we can now see that the decline in $D(u, \bar{p}_{\mathcal{B}})$ under severe shifts is due to the unconfident predictions, making $\bar{p}_{\mathcal{B}}$ appear closer to uniform. We use both the average pseudo-label distribution $\bar{p}_{\mathcal{B}}$ and prediction variance $d_{\mathcal{B}}$ to detect the magnitude and direction of label distribution shifts.

## 3 Label Distribution Shift-Aware Prediction Refinement for TTA

We introduce a prediction refinement module that can detect test-time class distribution shifts and modify the predictions of the trained classifiers to effectively reverse the class-wise confusion patterns.

**Prediction refinement module** $g_{\phi}$ Our core idea is that if the prediction refinement module $g_{\phi}$ experiences batches with diverse class distributions before the test time, it can develop the ability to refine inaccurate predictions resulting from label distribution shifts. Based on this insight, we train $g_{\phi}$ during the intermediate time, between the training and testing times, by exposing it to several batches with diverse class distributions from the training datasets. The module $g_{\phi}$ takes as inputs the average pseudo-label distribution $\bar{p}_{\mathcal{B}} = \frac{1}{|\mathcal{B}|} \sum_{(x,\cdot) \in \mathcal{B}} \text{softmax}(f_{\bar{\theta}}(x)) \in \mathbb{R}^K$ and prediction variance $d_{\mathcal{B}} = \frac{1}{|\mathcal{B}|} \sum_{(x,\cdot) \in \mathcal{B}} D(u, \text{softmax}(f_{\bar{\theta}}(x))) \in \mathbb{R}$ for each batch, to detect the label distribution shifts from uniformity. The module then outputs a square matrix $W_{\mathcal{B}} \in \mathbb{R}^{K \times K}$ and a vector $b_{\mathcal{B}} \in \mathbb{R}^K$, which together transform the model's logits $f_{\bar{\theta}}(x)$ for a sample $x \in \mathcal{B}$ to $f_{\bar{\theta}}(x)W_{\mathcal{B}} + b_{\mathcal{B}}$.

**Training of** $g_{\phi}$ During the intermediate time, we use the labeled training dataset $\mathcal{D}$ as the intermediate dataset $\mathcal{D}_{\text{int}}$, assuming that $\mathcal{D}$ is available while the test dataset $\mathcal{D}_{\text{test}}$ remains unavailable, as is common in previous TTA settings (Choi et al., 2022; Lim et al., 2022; Park et al., 2023). For example, we use CIFAR-10 dataset during the intermediate time on CIFAR-10C-imb benchmark. To create batches with diverse class

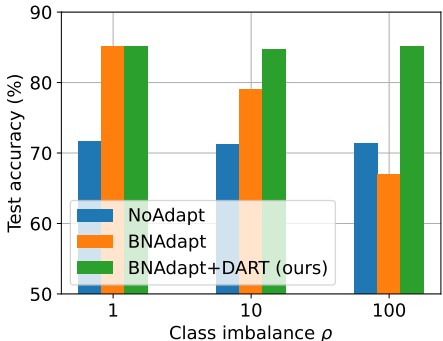

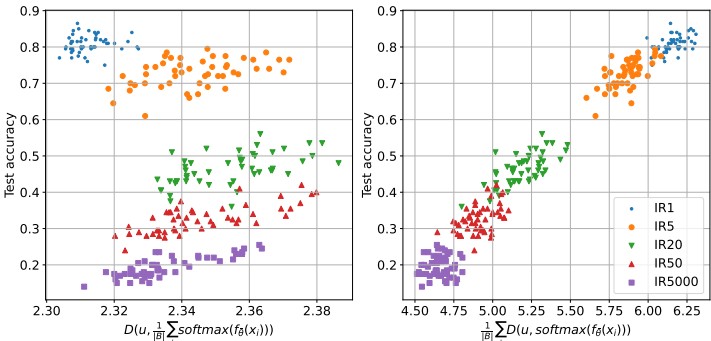

Figure 3: We observe performance degradation of BNAdapt (orange) as the class imbalance ratio $\rho$ increases on long-tailed CIFAR-10C. DART-applied BNAdapt (green) shows consistently improved performance regardless of class imbalance.

Figure 4: We demonstrate the relationship between (left) test accuracy and the difference between averaged pseudo label distribution, $\frac{1}{|\mathcal{B}|}\sum_{x_i\in\mathcal{B}}\text{softmax}(f_{\bar{\theta}}(x_i))$, and uniform distribution, $u$, and (right) the test accuracy and the prediction deviation $d_{\mathcal{B}}$, for each batch $\mathcal{B}$ with the BN-adapted classifier for CIFAR-10C-imb under Gaussian noise of 5 different imbalance ratios. Each point represents a single batch within the test dataset.

distributions during the intermediate time, we employ a Dirichlet distribution (Gong et al., 2022; Yurochkin et al., 2019). Batches sampled through i.i.d. sampling typically have class distributions resembling a uniform distribution. In contrast, batches sampled using the Dirichlet distribution exhibit a wide range of class distributions, including long-tailed distributions, as illustrated in Fig. 6 of Appendix A. If the original training dataset exhibits an imbalanced class distribution, as seen in datasets like PACS and OfficeHome (Fig. 5), this imbalance can inadvertently influence the class distributions within batches. To mitigate this, we create a class-balanced intermediate dataset $\mathcal{D}_{\text{int}}$ by uniformly sampling data from each class.

The training objective of $g_\phi$ for a Dirichlet-sampled batch $\mathcal{B}_{\text{Dir}} \subset \mathcal{D}_{\text{int}}$ is formulated as follows:

$$\mathcal{L}_{\text{imb}}(\phi) = \mathbb{E}_{(x,y)\in\mathcal{B}_{\text{Dir}}}[\text{CE}(y, \text{softmax}(f_{\bar{\theta}}(x)W_{\text{imb}}(\phi) + b_{\text{imb}}(\phi)))] \tag{1}$$

for $W_{\text{imb}}(\phi), b_{\text{imb}}(\phi) = g_\phi(\bar{p}_{\text{imb}}, d_{\text{imb}})$ where $\bar{p}_{\text{imb}}$ is the averaged pseudo label distribution, $d_{\text{imb}}$ is the prediction deviation of the batch $\mathcal{B}_{\text{Dir}}$, and CE denotes the cross-entropy loss. During the intermediate time, $g_\phi$ is optimized to minimize the CE loss between the ground truth labels of the training samples and the modified softmax probability. During this time, the parameters of the trained classifier $f_\theta$ are not updated, but the batch statistics in the classifiers are updated as in BNAdapt. Moreover, since $g_\phi$ training is independent of the test domains, e.g. test corruption types, we train $g_\phi$ only once for each classifier. Therefore, it requires negligible computation overhead (Appendix A).

**Regularization for $g_\phi$**  When there is no label distribution shift, there are no significant class-wise confusion patterns that the module needs to reverse (as shown in Fig. 9). However, when using only equation 1, the module may become biased towards class-imbalanced situations due to exposure to batches with imbalanced class configurations from Dirichlet sampling. To prevent this bias, we add a regularization term on the training objective of $g_\phi$, encouraging it to produce outputs similar to an identity matrix and a zero vector when exposed to batches i.i.d. sampled from the class-balanced intermediate dataset $\mathcal{D}_{\text{int}}$. The regularization for an i.i.d. sampled batch $\mathcal{B}_{\text{IID}} \subset \mathcal{D}_{\text{int}}$ is formulated as

$$\mathcal{L}_{\text{bal}}(\phi) = \mathbb{E}_{(x,y)\in\mathcal{B}_{\text{IID}}}[\text{MSE}(W_{\text{bal}}(\phi), \mathbb{I}_{K\times K}) + \text{MSE}(b_{\text{bal}}(\phi), \mathbf{0}_K)] \tag{2}$$

for $W_{\text{bal}}(\phi), b_{\text{bal}}(\phi) = g_\phi(\bar{p}_{\text{bal}}, d_{\text{bal}})$.

**Overall objective**  Combining these two losses, we propose a training objective for $g_\phi$:

$$\mathcal{L}(\phi) = \mathcal{L}_{\text{imb}}(\phi) + \alpha\mathcal{L}_{\text{bal}}(\phi), \tag{3}$$

where $\alpha$ is a hyperparameter that balances the two losses for class-imbalanced and balanced intermediate batches. We simply set $\alpha$ to 0.1 unless specified. The change in the outputs of $g_\phi$ with respect to different $\alpha$ can be observed in Appendix F. The pseudocode for DART is presented in Appendix B.

**Utilizing $g_\phi$ at test-time** Since the prediction refinement module $g_\phi$ only requires the averaged pseudo label distribution and the prediction deviation as inputs, it can be employed at test time when only unlabeled test data is available. To generate the affine transformation for prediction refinement, $g_\phi$ uses the outputs of the BN-adapted pre-trained classifier $f_{\bar{\theta}_0}$, not the classifier $f_\theta$ that is continually updated during test time by some TTA method. This is because $g_\phi$ has been trained to correct the inaccurate predictions of the BN-adapted classifier caused by label distribution shifts during the intermediate time. Specifically, $g_\phi$ generates $W_\mathcal{B}$ and $b_\mathcal{B}$ for prediction refinement of a test batch $\mathcal{B}$:

$$W_\mathcal{B}, b_\mathcal{B} = g_\phi(\bar{p}_\mathcal{B}, d_\mathcal{B}) \text{ where } \bar{p}_\mathcal{B} = \frac{1}{|\mathcal{B}|} \sum_{\hat{x} \in \mathcal{B}} \text{softmax}(f_{\bar{\theta}_0}(\hat{x})), d_\mathcal{B} = \frac{1}{|\mathcal{B}|} \sum_{\hat{x} \in \mathcal{B}} D(u, \text{softmax}(f_{\bar{\theta}_0}(\hat{x}))).$$

For a test data $\hat{x} \in \mathcal{B}$, we modify the prediction from $f_\theta$ from $\text{softmax}(f_\theta(\hat{x}))$ to $\text{softmax}(f_\theta(\hat{x})W_\mathcal{B} + b_\mathcal{B})$. These modified predictions can then be used for the adaptation of $f_\theta$. Consequently, our method can be integrated as a plug-in with any TTA methods that rely on pseudo labels obtained from the classifier. Details about the integration with existing TTA methods can be found in Appendix A.

## 4 Experiments

**Benchmarks** We consider two types of input data distribution shifts: synthetic and natural. Synthetic distribution shifts are artificially created using data augmentation techniques, such as image corruption with Gaussian noise. In contrast, natural distribution shifts arise from changes in image style, for instance, from artistic to photographic styles. We evaluate synthetic distribution shifts on CIFAR-10/100C and ImageNet-C (Hendrycks & Dietterich, 2019), and natural distribution shifts on CIFAR-10.1 (Recht et al., 2018), PACS (Li et al., 2017), and OfficeHome (Venkateswara et al., 2017) benchmarks. For synthetic distribution shifts, we apply 15 different types of common corruption, each at the highest severity level (i.e., level 5). To evaluate the impact of class distribution shifts, we use long-tailed distributions for CIFAR-10C (class imbalance ratio $\rho$) and online imbalanced distributions for CIFAR-100C and ImageNet-C (imbalance ratio IR). For datasets with a large number of classes, such as CIFAR-100C and ImageNet-C, the test batch size is often smaller than the number of classes, making the distribution of test batches significantly different from the assumed long-tailed distributions. Therefore, to evaluate the impact of class distribution shifts, we consider test batches modeled by online label distribution shifts, as described in Sec. 2 (with further details in Sec. A.1). For PACS and OfficeHome, we use the original datasets, which inherently include natural label distribution shifts across domains (Fig. 5).

**Baselines** Our method can be used as a plug-in for baseline methods that utilize test data predictions. We test the efficacy of DART combined with following baselines: (1) BNAdapt (Schneider et al., 2020) corrects batch statistics using test data; (2) TENT (Wang et al., 2020) fine-tunes BN layer parameters to minimize the prediction entropy of test data; (3) PL (Lee, 2013) fine-tunes the trained classifier using confident pseudo-labeled test samples; (4) NOTE (Gong et al., 2022) adapts classifiers while mitigating the effects of non-i.i.d test data streams through instance-aware BN and prediction-balanced reservoir sampling; (5) DELTA (Zhao et al., 2022) addresses incorrect BN statistics and prediction bias with test-time batch renormalization and dynamic online reweighting; (6) ODS (Zhou et al., 2023) estimates test data label distribution via Laplacian-regularized maximum likelihood estimation and adapts the model by weighting infrequent and frequent classes differently. (7) LAME (Boudiaf et al., 2022) modifies predictions with Laplacian-regularized maximum likelihood estimation using nearest neighbor information in the classifier's embedding space. (8) SAR (Niu et al., 2023) adapts models to lie in a flat region on the entropy loss surface. More details about baselines are available in Appendix A.

**Experimental details** We use ResNet-26 (He et al., 2016) for CIFAR and ResNet-50 for PACS, OfficeHome, and ImageNet benchmarks. For the distribution shift-aware module $g_\phi$, we use a 2-layer MLP (Haykin,

Table 1: Average accuracy on CIFAR-10C/10.1-LT, PACS, and OfficeHome. **Bold** indicates the best performance for each benchmark. The prediction refinement of DART consistently contributes to effective adaptation combined with existing entropy minimization-based TTA methods.

| Method | CIFAR-10C-LT | | | CIFAR-10.1-LT | | | PACS | OfficeHome |
|---|---|---|---|---|---|---|---|---|
| | $\rho = 1$ | $\rho = 10$ | $\rho = 100$ | $\rho = 1$ | $\rho = 10$ | $\rho = 100$ | | |
| NoAdapt | 71.7±0.0 | 71.3±0.1 | 71.4±0.2 | **87.7±0.0** | 87.3±0.7 | 88.3±0.7 | 60.7±0.0 | 60.8±0.0 |
| BNAdapt (Schneider et al., 2020) | 85.2±0.0 | 79.0±0.1 | 67.0±0.1 | 85.6±0.2 | 77.8±0.5 | 64.6±0.6 | 72.5±0.0 | 60.4±0.1 |
| BNAdapt+DART (ours) | 85.2±0.1 | 84.7±0.1 | 85.1±0.3 | 85.6±0.2 | 84.5±0.4 | 85.6±0.7 | 76.4±0.1 | 61.0±0.1 |
| TENT (Wang et al., 2020) | 86.3±0.1 | 82.9±0.4 | 70.4±0.1 | 85.8±0.3 | 77.6±1.5 | 64.6±1.1 | 76.8±0.6 | 60.9±0.2 |
| TENT+DART (ours) | 86.5±0.2 | 86.7±0.3 | **88.2±0.3** | 85.7±0.4 | 85.0±0.7 | 85.8±0.2 | 81.6±0.4 | 62.1±0.2 |
| PL (Lee, 2013) | 86.5±0.1 | 82.9±0.2 | 68.1±0.2 | 86.2±0.6 | 77.6±1.4 | 64.1±1.1 | 72.4±0.2 | 59.2±0.2 |
| PL+DART (ours) | 86.6±0.1 | **86.8±0.2** | 86.3±0.4 | 86.2±0.4 | 85.0±0.7 | 85.4±1.1 | 76.7±0.4 | 60.1±0.4 |
| NOTE (Gong et al., 2022) | 81.8±0.3 | 81.1±0.5 | 79.8±1.7 | 84.4±0.7 | 84.4±1.2 | 84.6±2.4 | 79.1±0.8 | 53.7±0.3 |
| NOTE+DART (ours) | 81.6±0.4 | 81.9±0.5 | 82.2±0.4 | 84.0±1.0 | 85.8±0.8 | 87.0±0.9 | 79.9±0.3 | 53.5±0.1 |
| LAME (Boudiaf et al., 2022) | 85.2±0.0 | 80.4±0.1 | 70.6±0.2 | 85.6±0.4 | 78.7±1.5 | 67.2±1.0 | 72.5±0.3 | 59.8±0.1 |
| LAME+DART (ours) | 85.2±0.0 | 82.8±0.1 | 81.2±0.2 | 85.6±0.5 | 81.9±1.0 | 80.7±0.5 | 76.0±0.2 | 60.2±0.2 |
| DELTA (Zhao et al., 2022) | 85.9±0.1 | 82.5±0.6 | 70.4±0.3 | 85.4±0.2 | 77.8±3.1 | 64.2±2.9 | 78.2±0.6 | 63.1±0.1 |
| DELTA+DART (ours) | 85.8±0.2 | 85.9±0.6 | 87.6±0.4 | 85.3±0.3 | 85.1±1.2 | 86.6±0.9 | **81.7±0.4** | **63.9±0.1** |
| ODS (Zhou et al., 2023) | 85.9±0.1 | 83.8±0.5 | 76.5±0.4 | 86.8±0.2 | 85.3±1.6 | 81.0±0.9 | 77.4±0.5 | 62.1±0.2 |
| ODS+DART (ours) | 85.9±0.1 | 85.0±0.4 | 84.9±0.2 | 86.7±0.3 | **87.4±1.4** | **88.5±1.2** | 79.7±0.4 | 62.5±0.2 |
| SAR (Niu et al., 2023) | **86.8±0.1** | 81.0±0.3 | 68.2±0.2 | 86.3±0.4 | 77.7±1.5 | 64.4±0.9 | 74.2±0.4 | 61.1±0.0 |
| SAR+DART (ours) | 86.7±0.1 | 86.4±0.3 | 87.0±0.4 | 86.1±0.3 | 84.9±0.8 | 85.6±0.4 | 78.5±0.2 | 61.7±0.2 |

1998) with a hidden dimension of 1,000. Fine-tuning layers, optimizers, and hyperparameters remain consistent with those introduced in each baseline for fair comparison. Implementation details for pre-training, intermediate-time training, and test-time adaptation are described in Appendix A.

## 4.1 Experimental Results

**DART-applied TTA methods**   In Table 1, we compare the experimental results for the original vs. DART-applied TTA methods across CIFAR-10C/10.1-LT, PACS and OfficeHome benchmarks. DART consistently enhances the performance of existing TTA methods for both synthetic and natural distribution shifts. In particular, DART achieves significant gains as the class imbalance ratio $\rho$ increases, while maintaining the original performance for $\rho = 1$. For instance, on CIFAR-10C-LT with class imbalance ratios $\rho = 10$ and 100, DART improves the test accuracy of BNAdapt by 5.7% and 18.1%, respectively. DART's performance improvement can vary depending on how the baseline method handles pseudo labels. The baseline methods discussed in this paper can be categorized into two categories: those designed to mitigate the effects of label distribution shift, such as NOTE, ODS, and DELTA, and the remaining baseline methods like TENT and SAR. As shown in Table 1, methods like TENT and SAR suffer significant performance degradation due to label distribution shifts because they do not consider the presence or intensity of test-time label distribution shifts. And, DART successfully recovers the predictions significantly degraded by label distribution shift. Conversely, methods such as NOTE, which builds a prediction balancing memory for adaptation, and DELTA and ODS, which compute class frequencies in the test data and assign higher weights to infrequent class data, show less performance degradation by label distribution shifts compared to other baseline methods. However, since these methods still rely on pseudo-labeled test data for memory construction or weight computation, the accuracy of pseudo labels significantly impacts their performance. Thus, as observed in Table 1, DART's prediction refinement scheme also contributes to performance improvement in these methods under label distribution shifts. In conclusion, the degree of performance degradation due to label distribution shifts and performance recovery by DART varies depending on the baseline methods used.

**DART on large-scale benchmarks**   To apply DART to larger-scale benchmarks, such as CIFAR-100C-imb and ImageNet-C-imb, we introduce a variant called DART-split. The main difference is that DART-split divides the prediction refinement module $g_\phi$ into two parts: $g_{\phi_1}$ for detecting the severity of label distribution shifts and $g_{\phi_2}$ for generating an effective affine transformation for prediction refinement. Originally, DART

Table 2: Average accuracy on CIFAR-100C, and ImageNet-C under online imbalance setup with several imbalance ratios (IR). Both DART and DART-split effectively correct inaccurate predictions. On the large-scale benchmarks such as CIFAR-100C and ImageNet-C, DART-split demonstrates superior performance. Refer to Table 8 for the combined results with other TTA methods.

| | CIFAR-100C-imb | | | | ImageNet-C-imb | | | |
|---|---|---|---|---|---|---|---|---|
| | IR1 | IR200 | IR500 | IR50000 | IR1 | IR1000 | IR5000 | IR500000 |
| NoAdapt | 41.5±0.3 | 41.5±0.3 | 41.4±0.3 | 41.5±0.3 | 18.0±0.0 | 18.0±0.1 | 18.0±0.0 | 18.1±0.1 |
| BNAdapt | 59.8±0.5 | 29.1±0.5 | 18.8±0.4 | 9.3±0.1 | 31.5±0.0 | 19.8±0.1 | 8.4±0.1 | 3.4±0.0 |
| BNAdapt+ DART (ours) | 59.9±0.3 | 41.8±0.8 | 32.2±0.7 | 22.7±0.6 | - | - | - | - |
| BNAdapt+ DART-split (ours) | 59.7±0.5 | 45.8±0.2 | 50.2±0.3 | 49.1±0.7 | 31.5±0.0 | 20.0±0.2 | 11.5±0.6 | 8.2±0.5 |

Table 3: Ablation studies highlighting the critical role of the prediction refinement module's design, particularly the significance of its inputs and outputs, in enhancing DART's effectiveness.

| | $g_\phi$ input | | $g_\phi$ output | | CIFAR-10C-LT | | |
|---|---|---|---|---|---|---|---|
| | $\bar{p}_\mathcal{B}$ | $d_\mathcal{B}$ | $W$ | $b$ | $\rho = 1$ | $\rho = 10$ | $\rho = 100$ |
| BNAdapt | | | | | 85.2±0.0 | 79.0±0.1 | 67.0±0.1 |
| BNAdapt+DART (ours) | ✓ | | ✓ | ✓ | 81.8±1.6 | 78.0±1.0 | 81.9±0.5 |
| | ✓ | ✓ | ✓ | | 85.0±0.1 | 84.0±0.2 | 83.0±0.5 |
| | ✓ | ✓ | | ✓ | **85.4±0.0** | 83.6±0.2 | 80.7±0.1 |
| | ✓ | ✓ | ✓ | ✓ | 85.2±0.1 | **84.7±0.1** | **85.1±0.3** |

used a single module $g_\phi$ to handle both roles, trained with the loss equation 1 for refining predictions on diverse class-imbalanced batches, regularized by equation 2 using nearly class-balanced batches. However, with a large number of classes, the module needs to experience numerous types of distribution shifts during intermediate training to generate effective transformations at test time, regardless of the severity of class distribution shifts. To address this, DART-split separates these tasks. $g_{\phi_2}$ is trained exclusively on Dirichlet-sampled batches, focusing on severely class-imbalanced batches for prediction refinement. The entire module then decides whether to apply $g_{\phi_2}$'s refinement for each test batch based on the output of $g_{\phi_1}$, which detects the severity of label distribution shifts using the prediction deviation $d_\mathcal{B}$ of each batch. $g_{\phi_1}$ outputs a severity score $s_\mathcal{B}$ ranging from 0 to 1, indicating the degree of label distribution shift. If $s_\mathcal{B}$ exceeds 0.5, indicating a severe shift, the batch's predictions are modified using the affine transformation $W_\mathcal{B}$ and $b_\mathcal{B}$ generated by $g_{\phi_2}$, i.e., softmax($f_\theta(x)W_\mathcal{B} + b_\mathcal{B}$). Otherwise, the prediction is not modified. More detailed explanation of DART-split is provided in Appendix B. In Table 2, we compare the experimental results for the original vs. DART-split-applied BNAdapt on the CIFAR-100C-imb and ImageNet-C-imb benchmarks. DART-split consistently improves performance, achieving a 16.7-39.8% increase on CIFAR-100C-imb with imbalance ratios of 200-50000 and a 0.2-4.8% increase on ImageNet-C-imb with imbalance ratios of 1000-500000. It notably outperforms BNAdapt under severe label distribution shifts and surpasses NoAdapt by over 4% across all imbalance ratios on CIFAR-100C-imb. In Table 8, we also show the results of DART-split combined with existing TTA methods demonstrating consistent performance gains, on CIFAR-100C-imb and ImageNet-C-imb. Additionally, we report the performance of DART-split compared to DART on CIFAR-10C-LT in Table 7.

## 4.2 Ablation Studies

**Importance of inputs and outputs of $g_\phi$ for label distribution detection and prediction refinement**
In Table 3, we present ablation studies demonstrating the effectiveness of the two inputs–the averaged pseudo-label distribution $\bar{p}_\mathcal{B}$ and prediction deviation $d_\mathcal{B}$–and the two outputs, $W \in \mathbb{R}^{K \times K}$ and $b \in \mathbb{R}^K$, of DART in prediction refinement. First, we observe that using the prediction deviation $d_\mathcal{B}$ as an input, in addition to the average pseudo-label, results in a gain of 3.2-6.7% when $g_\phi$ outputs both $W\&b$. Additionally, using the square matrix $W$ for prediction refinement outperforms using the bias vector $b$ as label distribution shifts become more severe, showing improvements of 0.4% and 2.3% on CIFAR-10C-LT with $\rho = 10$ and 100, respectively.

The best performance for $\rho = 10$ and $100$ is achieved when both $W$ and $b$ are used. These results align with the analysis in Section 2, indicating that as label distribution shifts become more severe, class-wise confusion patterns become clearer, making it crucial to reverse these patterns for improving BNAdapt's performance.

Table 4: Average accuracy (%) on CIFAR-10C and CIFAR-10C-imb of DART-applied BNAdapt with different sampling methods during intermediate time.

| | Sampling at int. time | | CIFAR-10C-LT | | | CIFAR-10C-imb | | | |
| | Dirichelt (used) | Unif&LT($\rho = 20$) | $\rho = 1$ | $\rho = 10$ | $\rho = 100$ | IR1 | IR20 | IR50 | IR5000 |
|---|---|---|---|---|---|---|---|---|---|
| BNAdapt | | | 85.2±0.0 | 79.0±0.1 | 67.0±0.1 | 85.3±0.1 | 50.0±0.4 | 34.6±0.4 | 20.3±0.1 |
| BNAdapt+DART (ours) | ✓ | | 85.2±0.1 | 84.7±0.1 | 85.1±0.3 | 85.3±0.2 | 76.2±0.3 | 78.2±0.1 | 82.4±0.7 |
| | | ✓ | 85.4±0.0 | 85.6±0.1 | 87.3±0.2 | 85.5±0.1 | 58.9±0.5 | 43.8±0.6 | 28.7±0.2 |

**Importance of Dirichlet sampling during intermediate time**   We use Dirichlet sampling during the intermediate time to make the prediction refinement module experience intermediate batches with diverse label distributions. To evaluate the effectiveness of the Dirichlet sampling, we consider a scenario where the module experiences only three types of batches during the intermediate time: uniform, long-tailed with a class imbalance ratio $\rho = 20$, and inversely long-tailed class distributions, inspired by Park et al. (2023). In Table 4, we report the performance of BNAdapt combined with this DART variant on CIFAR-10C-LT and CIFAR-10C-imb. When experiencing only uniform and long-tailed distributions during the intermediate time, the test accuracy of the variant of DART surpasses that of the original DART on CIFAR-10C-LT since it experiences only similar label distribution shifts during the intermediate time. However, on CIFAR-10C-imb, we can observe that the DART variant's performance is significantly lower since the test-time label distributions become more diverse and severely imbalanced. Thus, using Dirichlet sampling to experience a wide range of class distributions during the intermediate time is crucial for effectively addressing diverse test time label distribution shifts.

Table 5: Average accuracy (%) on CIFAR-10C-LT of DART with different $g_\phi$ outputs.

| | $g_\phi$ output | | CIFAR-10C-LT | | |
| | affine (used) | additive params | $\rho = 1$ | $\rho = 10$ | $\rho = 100$ |
|---|---|---|---|---|---|
| BNAdapt | | | 85.2±0.0 | 79.0±0.1 | 67.0±0.1 |
| BNAdapt+DART (ours) | ✓ | | 85.2±0.1 | 84.7±0.1 | 85.1±0.3 |
| | | ✓ | 82.0±2.2 | 79.9±2.0 | 82.3±1.6 |

**Importance of DART prediction refinement scheme using affine transformation**   To test the effectiveness of the DART's prediction refinement scheme using the affine transformation, we consider a variant of DART that modifies $g_\phi$ to generate the additive parameters for a part of the classifier, including affine parameters for the output of the feature extractor and the weight difference for the last linear classifier weights, inspired by LSA (Park et al., 2023). For this case, the output dimension of $g_\phi$ gets larger since it is proportional to the feature dimension of the trained model (e.g. 512 for ResNet26). For a fair comparison, we maintain the number of parameters of $g_\phi$. In Table 5, we report the performance of BNAdapt combined with the DART variant on CIFAR-10C-LT. The variant of DART shows worse performance on all $\rho$s and significant performance variability depending on different random seeds. We conjecture that this is because $g_\phi$ struggles to learn to generate additive parameters proportional to the feature dimensions in response to various class distribution shifts.

We also present additional experiments in Appendix G. These experiments demonstrate the effectiveness of DART in a more challenging TTA scenario, where test samples with multiple types of corruption are encountered simultaneously. We also verify the robustness of DART against changes in hyperparameters, such as the structure of $g_\phi$ and intermediate batch size. Moreover, we introduce a variant of DART that reduces the dependency on labeled training data during the intermediate training phase of DART by utilizing the condensed training dataset as in Kang et al. (2023).

## 5    Related Works

**Methods to handle label distribution shifts**    The scenario where class distributions differ between training and test datasets has been primarily studied in the context of long-tail learning. Long-tail learning addresses class imbalance through two main approaches: re-sampling (Chawla et al., 2002; Liu et al., 2008) and re-weighting (Hong et al., 2021; Menon et al., 2020). Recently, the impact of test-time label distribution shift in domain adaptation has gained attention. Specifically, RLSbench (Garg et al., 2023) reveals performance drops of TTA methods, BNAdapt and TENT, under label distribution shifts, and proposes a recovery method through re-sampling and re-weighting. However, the method is unsuitable for TTA settings due to the simultaneous use of training and test data. DART enhances the BN-adapted classifier predictions by correcting errors through a prediction refinement module, without relying on training data during test time.

**TTA method utilizing intermediate time**    Recent works (Choi et al., 2022; Lim et al., 2022) have explored methods to prepare for unknown test-time distribution shifts by leveraging the training dataset during an intermediate time. For instance, LSA (Park et al., 2023) conducts intermediate training to enhance classifier performance on long-tailed training datasets when applied to test datasets with different long-tailed distributions. Specifically, LSA trains a label shift adapter using batches with three types of class distributions (uniform, imbalanced training distribution, and inversely imbalanced) to produce additive parameters for the classifier. In contrast, DART exposes the prediction refinement module to a more diverse range of class distributions through Dirichlet sampling and generates an effective affine transformation for logit refinement. Additionally, while LSA uses only averaged pseudo-label distributions to estimate label distribution shifts, DART also considers prediction deviation to more effectively detect label distribution shifts. In Section G, we demonstrate the efficacy of DART's sampling and prediction modification schemes compared to LSA. More related works are reviewed in Appendix C.

## 6    Discussion and Conclusion

### 6.1    Discussion

**Performance on large-scale benchmarks**    While DART consistently enhances BN-adapted classifiers on large-scale datasets such as OfficeHome and PACS under test-time label distribution shifts (Table 1), the performance gains by DART may appear less noticeable on large-scale benchmarks compared to small-scale benchmarks. In these settings, the batch size is typically smaller than the total number of classes, so even if the test dataset's class distribution differs significantly from that of the training dataset, each batch experiences only minimal distribution shifts. Consequently, while DART's improvement may seem less significant compared to those on small-scale datasets like CIFAR-10C, it still effectively adjusts inaccurate predictions by label distribution shifts. Additionally, as the number of classes increases, the variety of potential class distribution shifts grows dramatically. Therefore, training the prediction refinement module at the intermediate time becomes more challenging on large-scale benchmarks with many classes, such as ImageNet-C (Table 2). This highlights a need for further research on more efficient methods to train the prediction refinement module, especially in extremely large label spaces.

**The use of training data during the intermediate time**    Although DART employs labeled training data to train its prediction refinement module before the test time, it does not violate TTA protocols and is commonly adopted by various TTA methods. In particular, DART's intermediate-time training with labeled data does not conflict with TTA guidelines because the training data is exclusively used to refine the prediction module during the intermediate time and is no longer accessible during the test time. Moreover, as described in Section 5, several recent TTA methods (Choi et al., 2022; Lim et al., 2022; Park et al., 2023) also utilize training data before test time to prepare for unknown distribution shifts. Nevertheless, in scenarios where the labeled training data for a given trained model is unclear, obtaining the training data can be challenging. To increase the scalability of DART in these situations, exploring approaches that leverage an auxiliary dataset instead of the original training data may be a promising direction for future work.

## 6.2 Conclusion

We proposed DART, a method to mitigate test-time class distribution shifts by accounting for class-wise confusion patterns. DART trains a prediction refinement module during an intermediate time to detect label distribution shifts and correct predictions of BN-adapted classifiers. Our experiments demonstrate DART's effectiveness across various benchmarks, including synthetic and natural distribution shifts. DART's intermediate-time training incurs additional costs compared to traditional TTA. However, these costs are relatively minimal, especially considering the significant performance improvements it offers when combined with traditional TTA methods (Table 1). The low cost is partly due to using the existing training dataset, rather than requiring an auxiliary dataset, and its short runtime. This paper highlights and effectively addresses the negative impact of test-time label distribution shifts on existing TTA methods. As label distribution shifts are common in real-world scenarios, we expect future studies to build on our approach to enhance real-world scalability by making models robust to test-time shifts in both input and label distributions.

**Broader Impact Statement**

This paper highlights and effectively resolves the negative impact of test-time label distribution shifts to existing TTA methods. Since the label distribution shifts between the training and test domains are common in the real world, we expect further studies that are robust to test-time shifts on both input data and label distributions to be built on our work to enhance real-world scalability.

**Acknowledgement**

This work was supported by the National Research Foundation of Korea (NRF) grant funded by the Korea government (MSIT) (No. RS-2024-00408003 and No. 2021R1C1C11008539).

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

# Appendix

## A Implementation Details

### A.1 Details about Dataset

We consider two types of input data distribution shifts: synthetic and natural distribution shifts. The synthetic and natural distribution shifts differ in their generation process. Synthetic distribution shift is artificially generated by data augmentation schemes including image corruption like Gaussian noise and glass blur. On the other hand, the natural distribution shift occurs due to changes in image style transfer, for example, the domain is shifted from artistic to photographic styles.

For the synthetic distribution shift, we test on CIFAR-10/100C and ImageNet-C, which is created by applying 15 types of common image corruptions (e.g. Gaussian noise and impulse noise) to the clean CIFAR-10/100 test and ImageNet validation datasets. We test on the highest severity (*i.e.*, level-5). CIFAR-10/100C is composed of 10,000 generic images of size 32 by 32 from 10/100 classes, respectively. ImageNet-C is composed of 50,000 generic images of size 224 by 224 from 1,000 classes. The class distributions of the original CIFAR-10/100C and ImageNet-C are balanced. Thus, we consider two types of new test datasets to change the label distributions between training and test domains. First, we consider CIFAR-10-LT, which has long-tailed class distributions, as described in Section 2. We set the number of images per class to decrease exponentially as the class index increases. Specifically, we set the number of samples for class $k$ as $n_k = n(1/\rho)^{k/(K-1)}$, where $n$ and $\rho$ denote the number of samples of class 0 and the class imbalance ratio, respectively. We also consider the test set of inversely long-tailed distribution in Figure 1. Specifically, we set the number of samples for class $k$ as $n_k = n(1/\rho)^{(K-1-k)/(K-1)}$, where $n$ and $\rho$ denote the number of samples of class $K-1$ and the class imbalance ratio, respectively. However, unlike CIFAR-10-LT, each test batch of CIFAR-100C-LT and ImageNet-C-LT tend to not have imbalanced class distributions, since the test batch size (*e.g.,* 32 or 64) is set to be smaller than the number of classes (100 and 1,000). Thus, we also consider CIFAR-100C-imb and ImageNet-C-imb, whose label distributions keep changing during test time, as described in Section 2. These datasets are composed of $K$ subsets, where $K$ is the number of classes. We assume a class distribution of the $k$-th subset as $[p_1, p_2, \ldots, p_K]$, where $p_k = p_{\max}$ and $p_i = p_{\min} = (1 - p_{\max})/(K-1)$ for $i \neq k$. Let IR $= p_{\max}/p_{\min}$ represent the imbalance ratio. Each subset consists of 100 samples from the CIFAR-100C and ImageNet-C test set based on the above class distribution. Thus, the new test set for CIFAR-100C/ImageNet-C is composed of 10,000/100,000 samples, respectively. Additionally, we shuffle the subsets to prevent predictions based on their order.

For the natural distribution shift, we test on CIFAR-10.1-LT, PACS, and OfficeHome benchmarks. CIFAR-10.1 (Recht et al., 2018) is a newly collected test dataset for CIFAR-10 from the TinyImages dataset (Torralba et al., 2008), and is known to exhibit a distribution shift from CIFAR-10 due to differences in data collection process and timing. Since the CIFAR-10.1 has a balanced class distribution, we construct a test set having a long-tailed class distribution, named CIFAR-10.1-LT, similar to CIFAR-10C-LT. PACS benchmark consists of samples from seven classes including dogs and elephants in four domains: photo, art, cartoon, and sketch. In PACS, we test the robustness of classifiers across 12 different scenarios, each using the four domains as training and test domains, respectively. The OfficeHome (Venkateswara et al., 2017) benchmark is one of the well-known large-scale domain adaptation benchmarks, comprising images from four domains (art, clipart, product, and the real world). Each domain consists of images belonging to 65 different categories. The number of data samples per class ranges from a minimum of 15 to a maximum of 99, with an average of 70. This ensures that the label distribution differences between domains are not substantial, as depicted in Figure 5. The data generation/collection process of PACS and OfficeHome benchmarks is different across domains, resulting in differently imbalanced class distribution, as illustrated in Figure 5.

### A.2 Details about Pre-training

We use ResNet-26 for CIFAR datasets, and ResNet-50 for PACS, OfficeHome, and ImageNet benchmarks as backbone networks. We use publicly released trained models and codes for a fair comparison. Specifically,

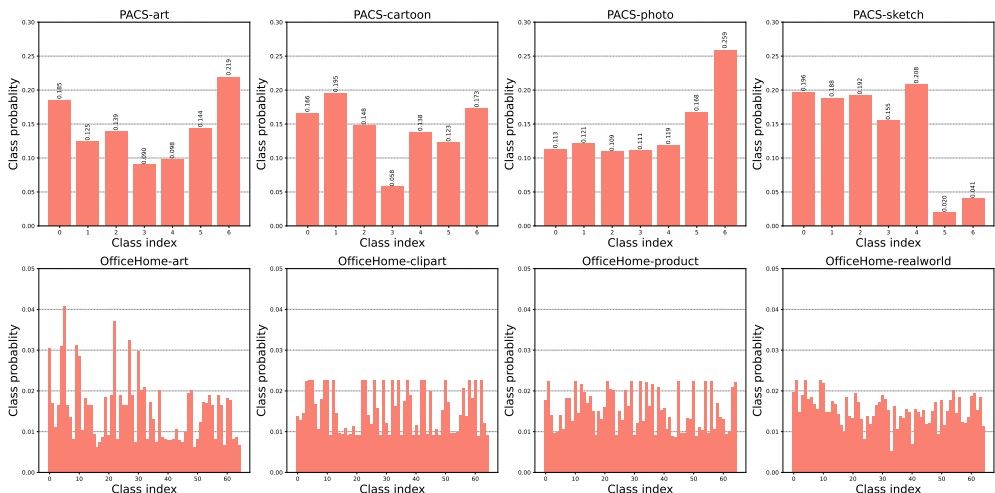

Figure 5: Class distribution of PACS and OfficeHome

for CIFAR-10/100 [1], we train the model with 200 epochs, batch size 200, SGD optimizer, learning rate 0.1, momentum 0.9, and weight decay 0.0005. For PACS and OfficeHome, we use released pre-trained models of TTAB (Zhao et al., 2023) [2]. For ImageNet-C, we use the released pre-trained models in the PyTorch library (Paszke et al., 2019) as described in (Niu et al., 2023).

### A.3 Details about Intermediate Time Training

We use the labeled dataset in the training domain to train the prediction refinement module $g_\phi$ for all benchmarks. If the original training dataset exhibits an imbalanced class distribution, e.g. PACS and OfficeHome, this imbalance can unintentionally influence the class distributions within the intermediate batches. To mitigate this, we create a class-balanced dataset $D_{\mathrm{int}}$ by uniformly sampling data from each class. For example, for the ImageNet benchmark, the intermediate dataset is a subset of the ImageNet training dataset, composed of 50 samples randomly selected from each of the classes. For the intermediate dataset, we use a simple augmentation such as normalization and center cropping.

We use a 2-layer MLP (Haykin, 1998) for the prediction refinement module $g_\phi$. $g_\phi$ is composed of two fully connected layers and ReLU (Agarap, 2018). The hidden dimension of the prediction refinement module is set to 1,000. During the intermediate time, we train $g_\phi$ by exposing it to several batches with diverse class distributions using the labeled training dataset. We train $g_\phi$ with Adam optimizer (Kingma & Ba, 2014), a learning rate of 0.001, and cosine annealing for 50 epochs. For large-scale benchmarks, such as CIFAR-100 and ImageNet, $g_\phi$ is trained for 100/200 epochs to expose it to diverse label distribution shifts of many classes, respectively. To make intermediate batches having diverse class distributions, we use Dirichlet sampling as illustrated in Figure 6 and 7 with two hyperparameters, the Dirichlet sampling concentration parameter $\delta$, and the number of chunks $N_{\mathrm{dir}}$. As these two hyperparameters increase, the class distributions of intermediate batches become similar to the uniform. $\delta$ is set to 0.001 for ImageNet-C, 0.1 for CIFAR-100, and 10 for other benchmarks. $N_{\mathrm{dir}}$ is set to 2000 for ImageNet-C, 1000 for CIFAR-100C, and 250 for other benchmarks. The intermediate batch size is set to 32/64 for ImageNet and all other benchmarks, respectively. On the other hand, we use a regularization on $g_\phi$ to produce $W$ and $b$ that are similar to the identity matrix of size $K$ and the zero vector, respectively, for the case when there are no label distribution shifts. The hyperparameter for the regularization $\alpha$ in equation 3 is generally set to 0.1. For PACS and OfficeHome, we set $\alpha$ to 10.

---

[1]https://github.com/locuslab/tta_conjugate

[2]https://github.com/LINs-lab/ttab

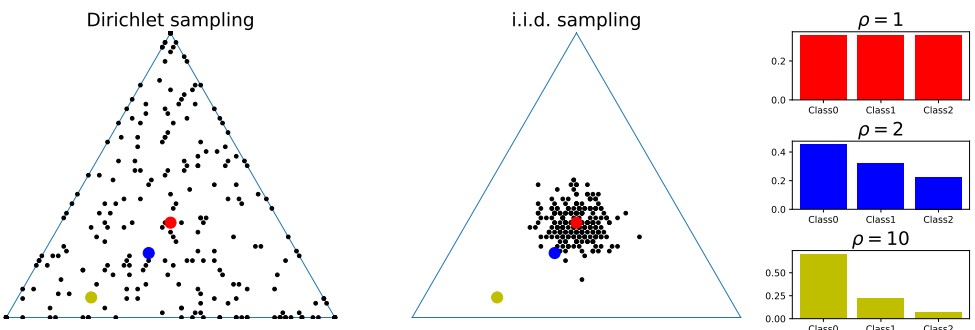

Figure 6: **Example of the Dirichlet distribution sampling**. IID (i.i.d.) sampling denotes standard uniform sampling. The black dots indicate the class distribution of the sampled batches. The red, blue, and yellow dots represent the class distributions of different class imbalance ratios $\rho$, namely 1,2, and 10, respectively. By employing the Dirichlet distribution for batch sampling, we can expose the model to numerous batches with diverse class distributions during the intermediate time, thereby enabling it to learn how to mitigate performance degradation caused by class distribution shifts.

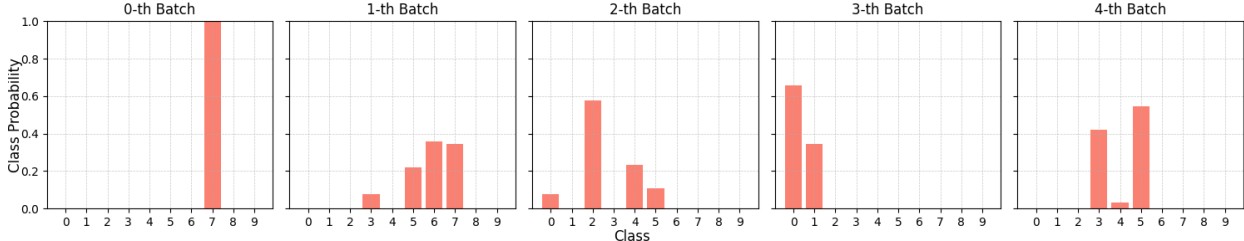

Figure 7: Class distributions of five batches sampled from CIFAR-10 using Dirichlet sampling. These sampled batches demonstrate diverse and varied class distributions, highlighting the effectiveness of Dirichlet sampling in generating a wide range of label distribution shifts.

### A.4    Details about Test-Time Adaptation

During test time, we fine-tune only the Batch Normalization (BN) layer parameters as is common with most TTA methods. We use the Adam optimizer with a learning rate of 0.001 for all TTA methods in every experiment, except on ImageNet benchmarks, following the approach in TENT (Wang et al., 2020). On ImageNet benchmarks, we use the SGD optimizer with a learning rate of 0.0005. We set the test batch size to 64 for PACS, OfficeHome, and ImageNet-C, and to 200 for CIFAR benchmarks. We run on 4 different random seeds for the intermediate-time and test-time training (0,1,2, and 3).

To generate the affine transformation for prediction refinement, $g_\phi$ uses the outputs of the BN-adapted pre-trained classifier $f_{\bar\theta_0}$, not the classifier $f_\theta$ that is continually updated during test time. This is because $g_\phi$ has been trained to correct the inaccurate predictions of the BN-adapted classifier caused by label distribution shifts during the intermediate time. Specifically, for a test batch $\mathcal{B}$,

$$W_\mathcal{B}, b_\mathcal{B} = g_\phi(\bar{p}_\mathcal{B}, d_\mathcal{B}) \quad \text{where} \quad \bar{p}_\mathcal{B} = \frac{1}{|\mathcal{B}|}\sum_{\hat{x}\in\mathcal{B}}\text{softmax}(f_{\bar\theta_0}(\hat{x})), d_\mathcal{B} = \frac{1}{|\mathcal{B}|}\sum_{\hat{x}\in\mathcal{B}}D(\text{softmax}(u, f_{\bar\theta_0}(\hat{x}))). \quad (4)$$

For a test data $\hat{x} \in \mathcal{B}$, we modify the prediction from $f_\theta$ from $\text{softmax}(f_\theta(\hat{x}))$ to $\text{softmax}(f_\theta(\hat{x})W_\mathcal{B} + b_\mathcal{B})$. These modified predictions can then be used for the adaptation of $f_\theta$. Since the affine transformation by $g_\phi$ focuses on prediction refinement, the confidence of predictions can be changed from the original predictions, potentially causing negative effects during test-time training (Goyal et al., 2022). Therefore, we use normalization to maintain the confidence level of predictions. For example, in the

case of TENT (Wang et al., 2020), we adapt the classifier $f_\theta$ using a training objective $\mathcal{L}_{\text{TENT}}(\theta) = \mathbb{E}_{\hat{x} \in \mathcal{B}}[-\sum_k \text{softmax}(\|f_\theta(\hat{x})\|_2 \frac{f_\theta(\hat{x})W_\mathcal{B}+b_\mathcal{B}}{\|f_\theta(\hat{x})W_\mathcal{B}+b_\mathcal{B}\|_2})_k \log \text{softmax}(\|f_\theta(\hat{x})\|_2 \frac{f_\theta(\hat{x})W_\mathcal{B}+b_\mathcal{B}}{\|f_\theta(\hat{x})W_\mathcal{B}+b_\mathcal{B}\|_2})_k]$.

During the test time, there often arises an issue where the logits/pseudo label distribution of unseen test data exhibits unconfidence, making it challenging to apply prediction refinement through $g_\phi$ directly. To address this, we utilize the softmax temperature scaling. For OfficeHome and CIFAR100, we multiply logits by 2 and 1.1 for test batches for softmax temperature scaling, respectively.

## A.5 Details about Baseline Methods

**BNAdapt (Schneider et al., 2020)** BNAdapt does not update the parameters in the trained model, but it corrects the BN statistics in the model using the BN statistics computed on the test batches.

**TENT (Wang et al., 2020)** TENT replaces the BN statistics of the trained classifier with the BN statistics computed in each test batch during test time. And, TENT fine-tunes the BN layer parameters to minimize the prediction entropy of the test data.

**PL (Lee, 2013)** PL regards the test data with confident predictions as reliable pseudo-labeled data and fine-tunes the BN layer parameters to minimize cross-entropy loss using these pseudo-labeled data. We set the confidence threshold to 0.95 for filtering out test data with unconfident predictions.

**NOTE (Gong et al., 2022)** NOTE aims to mitigate the negative effects of a non-i.i.d stream during test time by instance-aware BN (IABN) and prediction-balanced reservoir sampling (PBRS). IABN first detects whether a sample is from out-of-distribution or not, by comparing the instance normalization (IN) and BN statistics for each sample. For in-distribution samples, IABN uses the standard BN statistics, while for out-of-distribution samples, it corrects the BN statistics using the IN statistics. We set the hyperparameter to determine the level of detecting out-of-distribution samples to 4 as used in NOTE (Gong et al., 2022). Due to the non-i.i.d stream, class distribution within each batch is highly imbalanced. Thus, PBRS stores an equal number of predicted test data for each class and does test-time adaptation using the stored data in memory. We set the memory size same as the batch size, for example, 200 for CIFAR benchmarks. NOTE create prediction-balanced batches and utilize them for adaptation. Thus, the batches for adaptation in NOTE have different class distributions from the test dataset, unlike other baselines including TENT. Therefore, in NOTE, DART is used exclusively to enhance the prediction accuracy of the examples stored in memory. Since NOTE utilizes IABN layers instead of BN layers, we replace the BN layers of the pre-trained model with IABN layers and fine-tune the model for 30 epochs using the labeled training dataset as described in Gong et al. (2022).

**DELTA (Zhao et al., 2022)** DELTA aims to alleviate the negative effects such as wrong BN statistics and prediction bias by test-time batch renormalization (TBR) and dynamic online reweighting (DOT). Since the BN statistics computed in the test batch are mostly inaccurate, TBR corrects the BN statistics with renormalization using test-time moving averaged BN statistics with a factor of 0.95. DOT computes the class prediction frequency in exponential moving average with a factor of 0.95 during test time and uses the estimated class prediction frequency to assign low/high weights to frequent/infrequent classes, respectively.

**LAME (Boudiaf et al., 2022)** LAME modifies the prediction by Laplacian regularized maximum likelihood estimation considering nearest neighbor information in the embedding space of the trained classifier. Note that LAME does not update network parameters of the classifiers. We compute the similarity among samples for the nearest neighbor information with k-NN with $k = 5$.

**ODS (Zhou et al., 2023)** ODS consists of two modules: a distribution tracker and a prediction optimizer. The distribution tracker estimates the label distribution $w$ and modified label distribution $z$ by LAME, leveraging features of adapted classifier $f_\theta$ and predictions from a pre-trained classifier $f_{\theta_0}$. The prediction optimizer produces refined predictions by taking an average of the modified label distribution $z$ and the

prediction of the adapted classifier $f_\theta$, which is trained by a weighted loss that assigns high/low weights $w$ on infrequent/frequent classes, respectively.

Since DELTA and ODS utilize predictions to estimate label distribution shifts, our prediction refinement scheme can enhance the label distribution shift estimation, leading to more effective test-time adaptation.

**SAR (Niu et al., 2023)**   SAR adapts the trained models to lie in a flat region on the entropy loss surface using the test samples for which the entropy minimization loss does not exceed 0.4*log(number of classes). We only reset the model for the experiments of online imbalance when the exponential moving average (EMA) of the 2nd loss is smaller than 0.2. The authors of SAR observe a significant performance drop when label distribution shifts or batch sizes decrease on BNAdapt, explain this drop by the BN layer's property that uses the test batch statistics as shown in Section 2. Thus, they recommend the usage of other normalization layers instead of BN layers. However, in this paper, we analyze the error patterns of BN-adapted classifiers under label distribution shifts and propose a new method to refine the inaccurate prediction of the BN-adapted classifiers. We expect that it can greatly enhance the scalability of BN-adapted classifiers to real-world problems with shifted label distributions.

NOTE, ODS, and DELTA, which address class imbalances by re-weighting, the prediction-balancing memory, and the modified BN layers, can be combined with any entropy minimization test-time adaptation methods like TENT, PL, and SAR. For a fair comparison, we report the results of combining these methods with TENT, specifically TENT+NOTE, TENT+ODS, and TENT+DELTA, in Table 1.

Table 6: **# of parameters, FLOPs, wall clock at the test time.** DART-applied BNAdapt does not require additional cost except the memory for $g_\phi$. On the other hand, applying DART to other TTA methods requires more computation costs due to the need for an additional BN-adapted classifier $f_{\bar\theta_0}$ to compute affine transformations. However, the additional cost caused by $f_{\bar\theta_0}$ is negligible compared to the performance increases.

| Method | # params (# trainable params) | FLOPs | Wall clock |
|---|:---:|:---:|:---:|
| NoAdapt | 17.4 M (0 M) | 0.86 G | 2.24 sec |
| BNAdapt | 17.4 M (0 M) | 0.86 G | 2.36 sec |
| BNAdapt+DART | 17.6 M (0 M) | 0.86 G | 2.35 sec |
| TENT | 17.4 M (0.013 M) | 0.86 G | 3.93 sec |
| TENT+DART | 35.0 M (0.013 M) | 1.72 G | 4.81 sec |

### A.6   Additional Costs of DART

The additional cost incurred by DART during intermediate and test times is minimal, while it brings significant performance gains. DART's intermediate time training has a very low computation overhead. This is because the prediction refinement module $g_\phi$ is trained independently of the test domains, requiring training only once for each classifier. The parameters of the pre-trained classifier, except for BN statistics, are kept frozen while a shallow $g_\phi$ (a 2-layer MLP) is trained. For instance, training $g_\phi$ during the intermediate time for CIFAR-10 takes only 7 minutes and 9 seconds on RTX A100. Moreover, the DART-applied TTA methods also do not require significant additional costs compared to naive TTA methods. We summarize the number of parameters, FLOPs, and wall clocks on balanced CIFAR-10C in Table 6. When applying DART to BNAdapt, there is almost no additional cost aside from the memory for $g_\phi$. This is because $g_\phi$ remains fixed during the test time and is shallow. On the other hand, applying DART to other TTA methods requires additional costs due to the need for an additional classifier $f_{\bar\theta_0}$ to compute affine transformations, aside from the continuously adapted classifier. The additional costs by the classifier are insignificant since its parameters are kept frozen during the test time. The small additional cost is negligible (1-2 seconds) compared to the performance gains of 5.7/18.1% for CIFAR-10C-LT with $\rho = 10/100$, respectively.

# B  Comparison of DART and DART-split

---

**Algorithm 1:** Intermediate time training of DART

---

Input:  pre-trained classifier $f_\theta$, labeled training dataset $\mathcal{D}$, hyperparameter $\alpha$ for balancing two losses, learning rate $\beta$

Set class-balanced training dataset as an intermediate dataset $\mathcal{D}_{\text{int}}$
Initialize 2-layer MLP $g_\phi$ for logit refinement
for iterations
    # (1) Class-imbalanced intermediate batch
    Sample a batch $\mathcal{B}_{\text{Dir}} = (X_{\text{imb}}, Y_{\text{imb}})$ of size $M$ from $\mathcal{D}_{\text{int}}$ using Dirichlet distribution
    $\bar{\theta} \leftarrow \text{BNAdapt}(\theta, \mathcal{B}_{\text{Dir}})$ # Adapt BN layer statistics using the current batch
    $l_{\text{imb}} \leftarrow f_{\bar{\theta}}(X_{\text{imb}}) \in \mathbb{R}^{M \times K}$ # logit of class-imbalanced intermediate batch
    $W_{\text{imb}}(\phi), b_{\text{imb}}(\phi) \leftarrow \text{LogitModifier-DART}(l_{\text{imb}}, g_\phi)$ (Algorithm 2)
    $\mathcal{L}_{\text{imb}}(\phi) = \frac{1}{|\mathcal{B}_{\text{Dir}}|} \sum_i \text{CE}(Y_{\text{imb},i}, \text{softmax}(l_{\text{imb},i} W_{\text{imb}}(\phi) + b_{\text{imb}}(\phi)))$
    # (2) Class-balanced intermediate batch
    Sample a batch $\mathcal{B}_{\text{IID}} = (X_{\text{bal}}, Y_{\text{bal}})$ of size $M$ i.i.d.  from $\mathcal{D}_{\text{int}}$
    $\bar{\theta} \leftarrow \text{BNAdapt}(\theta, \mathcal{B}_{\text{IID}})$ # Adapt BN layer statistics using the current batch
    $l_{\text{bal}} \leftarrow f_{\bar{\theta}}(X_{\text{bal}}) \in \mathbb{R}^{M \times K}$ # logit of class-balanced intermediate batch
    $W_{\text{bal}}, b_{\text{bal}} \leftarrow \text{LogitModifier-DART}(l_{\text{bal}}, g_\phi)$ (Algorithm 2)
    $\mathcal{L}_{\text{bal}}(\phi) = \text{MSE}(W_{\text{bal}}(\phi), I_K) + \text{MSE}(b_{\text{bal}}(\phi), 0_K)$
    # Update $g_\phi$
    $\phi \leftarrow \phi - \beta\nabla_\phi\{\mathcal{L}_{\text{imb}}(\phi) + \alpha\mathcal{L}_{\text{bal}}(\phi)\}$
Output:  trained $g_\phi$

---

**Algorithm 2:** LogitModifier-DART$(l, g_\phi)$

---

Input:  BN-adapted classifier's logit $l_{\mathcal{B}}$ for current batch $\mathcal{B}$, prediction refinement module $g_\phi$
$p \leftarrow \text{softmax}(l_{\mathcal{B}}) \in \mathbb{R}^{M \times K}$ # softmax output
# Compute an averaged pseudo label distribution $\bar{p} \in \mathbb{R}^K$ and prediction deviation $d_{\mathcal{B}} \in \mathbb{R}$
$\bar{p}_{\mathcal{B}} \leftarrow \frac{1}{|\mathcal{B}|} \sum_i p_i \in \mathbb{R}^K$
$d_{\mathcal{B}} \leftarrow \frac{1}{|\mathcal{B}|} \sum_i D(u, p_i) \in \mathbb{R}$, where $u$ is an uniform distribution of size $K$.
# Obtain $W \in \mathbb{R}^{K \times K}$ and $b \in \mathbb{R}^{1 \times K}$ by feeding $\bar{p}_{\mathcal{B}}$ and $d_{\mathcal{B}}$ into $g_\phi$
$W_{\mathcal{B}}, b_{\mathcal{B}} \leftarrow g_\phi(\bar{p}_{\mathcal{B}}, d_{\mathcal{B}})$
Output:  $W_{\mathcal{B}}, b_{\mathcal{B}}$

---

**Algorithm 3:** Intermediate time training of DART-split

---

Input:  pre-trained classifier $f_\theta$, labeled training dataset $\mathcal{D}$, learning rate $\beta_1$ and $\beta_2$
Set class-balanced training dataset as an intermediate dataset $\mathcal{D}_{\text{int}}$
Initialize $g_{\phi_1}$ of a single layer and 2-layer MLP $g_{\phi_2}$ for logit refinement
for iterations
    # (1) Class-imbalanced intermediate batch
    Sample a batch $\mathcal{B}_{\text{Dir}} = (X_{\text{imb}}, Y_{\text{imb}})$ of size $M$ from $\mathcal{D}_{\text{int}}$ using Dirichlet distribution
    $\bar{\theta} \leftarrow \text{BNAdapt}(\theta, \mathcal{B}_{\text{Dir}})$ # Adapt BN layer statistics using the current batch
    $l_{\text{imb}} \leftarrow f_{\bar{\theta}}(X_{\text{imb}}) \in \mathbb{R}^{M \times K}$ # logit of class-imbalanced intermediate batch
    $d_{\text{imb}} \leftarrow \frac{1}{|\mathcal{B}_{\text{Dir}}|} \sum_i D(u, \text{softmax}(l_{\text{imb},i})) \in \mathbb{R}$, where $u$ is an uniform distribution of size $K$.
    $W_{\text{imb}}(\phi_2), b_{\text{imb}}(\phi_2) \leftarrow \text{LogitModifier-DART-split}(l_{\text{imb}}, g_{\phi_2})$ (Algorithm 4)
    $\mathcal{L}_{\text{imb}}(\phi_2) = \frac{1}{|\mathcal{B}_{\text{Dir}}|} \sum_i \text{CE}(Y_{\text{imb},i}, \text{softmax}(l_{\text{imb},i} W_{\text{imb}}(\phi_2) + b_{\text{imb}}(\phi_2)))$
    # (2) Class-balanced intermediate batch
    Sample a batch $\mathcal{B}_{\text{IID}} = (X_{\text{bal}}, Y_{\text{bal}})$ of size $M$ i.i.d.  from $\mathcal{D}_{\text{int}}$
    $\bar{\theta} \leftarrow \text{BNAdapt}(\theta, \mathcal{B}_{\text{IID}})$ # Adapt BN layer statistics using the current batch
    $l_{\text{bal}} \leftarrow f_{\bar{\theta}}(X_{\text{bal}}) \in \mathbb{R}^{M \times K}$ # logit of class-balanced intermediate batch
    $d_{\text{bal}} \leftarrow \frac{1}{|\mathcal{B}_{\text{IID}}|} \sum_i D(u, \text{softmax}(l_{\text{bal},i})) \in \mathbb{R}$, where $u$ is an uniform distribution of size $K$.
    $\mathcal{L}_{\text{detect}}(\phi_1) = \text{BCE}(0, \sigma(g_{\phi_1}(d_{\text{bal}}))) + \text{BCE}(1, \sigma(g_{\phi_1}(d_{\text{imb}})))$, where $\sigma$ is a sigmoid function and BCE is binary cross entropy loss
    # Update $g_\phi$
    $\phi_1 \leftarrow \phi_1 - \beta_1\nabla_{\phi_1}\mathcal{L}_{\text{detect}}(\phi_1)$
    $\phi_2 \leftarrow \phi_2 - \beta_2\nabla_{\phi_2}\mathcal{L}_{\text{imb}}(\phi_2)$
Output:  trained $g_\phi = \{g_{\phi_1}, g_{\phi_2}\}$

---

## B.1  Detailed Explanation about DART-split

We propose a variant of DART, named DART-split, for large-scale benchmarks, which splits the prediction refinement module $g_\phi$ into two parts $g_{\phi_1}$ and $g_{\phi_2}$. In this setup, $g_{\phi_1}$ takes the prediction deviation as an input to detect label distribution shifts, while $g_{\phi_2}$ generates affine transformations using the averaged pseudo

---

**Algorithm 4:** LogitModifier-DART-split$(l, g_{\phi_2})$

---

Input: BN-adapted classifier's logit $l_{\mathcal{B}}$ for current batch $\mathcal{B}$, a part of prediction refinement module $g_{\phi_2}$
$p \leftarrow \texttt{softmax}(l_{\mathcal{B}}) \in \mathbb{R}^{M \times K}$ # softmax output
# Compute an averaged pseudo label distribution $\bar{p} \in \mathbb{R}^K$
$\bar{p}_{\mathcal{B}} \leftarrow \frac{1}{|\mathcal{B}|} \sum_i p_i \in \mathbb{R}^K$
# Obtain $W \in \mathbb{R}^{K \times K}$ and $b \in \mathbb{R}^{1 \times K}$ by feeding $\bar{p}_{\mathcal{B}}$ into $g_{\phi_2}$
$W_{\mathcal{B}}, b_{\mathcal{B}} \leftarrow g_{\phi_2}(\bar{p}_{\mathcal{B}})$
Output: $W_{\mathcal{B}}, b_{\mathcal{B}}$

---

label distribution. Specifically, $g_{\phi_1}$ takes the prediction deviation of each batch $\mathcal{B}$ as an input, and outputs a severity score $s_{\mathcal{B}} = \sigma(g_{\phi_1}(d_{\mathcal{B}}))$ ranging from 0 to 1, with higher values indicating more severe shifts. On the other hand, $g_{\phi_2}$ uses the averaged pseudo label distribution $\bar{p}_{\mathcal{B}}$ as an input to produce an affine transformation $W_{\mathcal{B}}$ and $b_{\mathcal{B}}$, effectively correcting predictions affected by the label distribution shifts. DART-split can address the scaling challenges faced by the original DART as the number of classes, $K$, increases.

At the intermediate time, $g_{\phi_1}$ and $g_{\phi_2}$ are trained individually for detecting label distribution shifts and generating affine transformation, respectively, as detailed in Algorithm 3. $g_{\phi_1}$ is trained to classify batches to either severe label distribution shift (output close to 1) or no label distribution shift (output close to 0), taking the prediction deviation as an input. Specifically, $g_{\phi_1}$ is trained to minimize $\mathrm{BCE}(0, \sigma(g_{\phi_1}(d_{\mathrm{bal}}))) + \mathrm{BCE}(1, \sigma(g_{\phi_1}(d_{\mathrm{imb}})))$, where $\sigma$ is a sigmoid function, BCE is the binary cross entropy loss, and $d_{\mathrm{bal}}, d_{\mathrm{imb}}$ are the prediction deviation values of the batches i.i.d. sampled and Dirichlet-sampled, respectively. $g_{\phi_1}$ is composed of a simple single layer, based on the observation in Fig. 4 that the prediction deviation almost linearly decreases as the imbalance ratio increases. On the other hand, $g_{\phi_2}$, is composed of a two-layer MLP and focuses on generating affine transformations solely from the averaged pseudo label distribution, following the training objectives similar to those of the original DART's $g_\phi$.

During testing, each test batch $\mathcal{B}$ is first evaluated by $g_{\phi_1}$ to determine whether there is a severe label distribution shift or not. If $s_{\mathcal{B}} = \sigma(g_{\phi_1}(d_{\mathcal{B}}))$ exceeds 0.5, indicating a severe label distribution shift, the predictions for $\mathcal{B}$ are adjusted using the affine transformations from $g_{\phi_2}$, i.e., for a test data $x \in \mathcal{B}$, the prediction is modified from $\mathrm{softmax}(f_\theta(x))$ to $\mathrm{softmax}(f_\theta(x)W_{\mathcal{B}} + b_{\mathcal{B}})$. Otherwise, we do not modify the prediction.

For ImageNet-C, the output dimension of $g_{\phi_2}$ increases as the number of classes increases, becoming more challenging to learn and generate a higher-dimensional square matrix $W$ and $b$ for large-scale datasets. To address this challenge, we modify the prediction refinement module to produce $W$ with all off-diagonal entries set to 0 and fix all entries of $b$ to 0 for ImageNet benchmarks. Moreover, to alleviate the discrepancy in the softmax prediction confidence between the training and test datasets, we store the confidence of the pre-trained classifier's softmax probabilities and perform softmax temperature scaling to ensure that the prediction confidence of the pre-trained classifier are maintained for the test dataset for each test corruption for ImageNet-C. Specifically, for the pre-trained classifier $f_{\theta_0}$, training dataset $\mathcal{D}$, and test dataset $\mathcal{D}_{\mathrm{test}}$, we aim to find $T$ that achieves the following:

$$\mathbb{E}_{(x,\cdot) \sim \mathcal{D}}[\max_k \mathrm{softmax}(f_{\theta_0}(x))_k] = \mathbb{E}_{\hat{x} \sim \mathcal{D}_{\mathrm{test}}}[\max_k \mathrm{softmax}(f_{\theta_0}(\hat{x})/T)_k]. \tag{5}$$

By Taylor series approximation, we can re-write equation 5 as in Hinton et al. (2015):

$$\mathbb{E}_{(x,\cdot) \sim \mathcal{D}}\left[\frac{1 + \max_k f_{\theta_0}(x)_k}{\sum_{k'}(1 + f_{\theta_0}(x)_{k'})}\right] = \mathbb{E}_{\hat{x} \sim \mathcal{D}_{\mathrm{test}}}\left[\frac{1 + \max_k f_{\theta_0}(\hat{x})_k/T}{\sum_{k'}(1 + f_{\theta_0}(\hat{x})_{k'}/T)}\right]. \tag{6}$$

With the assumption that the sum of logits, $\sum_{k'} f_{\theta_0}(\cdot)_{k'}$, are constant $l^{\mathrm{tr, \, sum}}$ and $l^{\mathrm{te, \, sum}}$ for any training and test data, respectively, we can re-write equation 6 and compute $T$ as

$$\frac{1 + \mathbb{E}_{(x,\cdot) \sim \mathcal{D}}\left[\max_k f_{\theta_0}(x)_k\right]}{K + l^{\mathrm{tr, \, sum}}} = \frac{1 + \mathbb{E}_{\hat{x} \sim \mathcal{D}_{\mathrm{test}}}\left[\max_k f_{\theta_0}(\hat{x})_k/T\right]}{K + l^{\mathrm{te, \, sum}}/T} \tag{7}$$

$$T = \frac{\mathbb{E}_{\hat{x} \sim \mathcal{D}_{\mathrm{test}}}\left[\max_k f_{\theta_0}(\hat{x})_k\right] - l^{\mathrm{te, \, sum}} \frac{1 + \mathbb{E}_{(x,\cdot) \sim \mathcal{D}}\left[\max_k f_{\theta_0}(x)_k\right]}{K + l^{\mathrm{tr, \, sum}}}}{K \frac{1 + \mathbb{E}_{(x,\cdot) \sim \mathcal{D}}\left[\max_k f_{\theta_0}(x)_k\right]}{K + l^{\mathrm{tr, \, sum}}} - 1}, \tag{8}$$

where $K$ is the number of classes. We store the average of the maximum logits and the sum of logits of the pre-trained classifier on the training data, and use them to calculate the softmax temperature $T$ at test time. For each test batch, we calculate $T$ by equation 8 for the test batch using the maximum logits and the sum of logits of the test batch using the pre-trained classifier. For the $(t+1)$-th test batch, we perform softmax temperature scaling using the average value of $T$ calculated from the first to the $(t+1)$-th test batches. Calculating the softmax temperature $T$ in equation 8 does not require the labels of the test data, ensuring that it does not violate the test-time adaptation setup.

## B.2    Comparison of DART and DART-split

Table 7: Test accuracy of DART-applied BNAdapt and DART-split-applied BNAdapt on CIFAR-10C-LT

|                               | $\rho = 1$     | $\rho = 10$    | $\rho = 100$   |
|-------------------------------|----------------|----------------|----------------|
| BNAdapt                       | 85.2±0.0       | 79.0±0.1       | 67.0±0.1       |
| BNAdapt+DART (ours)           | 85.2±0.1       | 84.7±0.1       | 85.1±0.3       |
| BNAdapt+DART-split (ours)     | 85.0±0.0       | 78.4±0.2       | 76.1±0.4       |

Both DART and DART-split utilize the averaged pseudo label distribution and prediction deviation to detect label distribution shifts and correct the inaccurate predictions of a BN-adapted classifier due to the label distribution shifts, by considering class-wise confusion patterns. However, they differ in the structure of the prediction refinement module $g_\phi$. The original DART fed two inputs into a single prediction refinement module that detects shifts and generates the appropriate affine transformation for each batch. On the other hand, DART-split divides this module into two distinct modules, each detecting the existence and severity of the label distribution shift and generating the affine transformation. We use the same hyperparameters for DART-split as much as possible to those of DART, but $\delta$ for CIFAR-100C is set to 0.01, and the softmax temperature for CIFAR-10C during testing is set to 1.1.

As shown in Table 1 and 2, we demonstrate that DART and DART-split show consistent performance improvements under several label distribution shifts in small to mid-scale benchmarks and large-scale benchmarks, respectively. A possible follow-up question is the efficiency of DART-split in benchmarks like CIFAR-10C-LT, which are not large-scale benchmarks. As summarized in Table 7, DART-split shows about a 10% improvement under severe label distribution shifts ($\rho = 100$) in CIFAR-10C-LT, and it performs comparably to naive BNAdapt at lower shifts ($\rho = 1, 10$). While original DART outperforms DART-split on CIFAR-10C-LT for all $\rho$s, DART-split's success in significantly enhancing predictions under severe shifts while maintaining performance under milder conditions is notable. On the other hand, in Table 2, DART-split shows superior performance on large-scale benchmarks compared to the original DART. Therefore, we recommend using the original DART for small to mid-scale benchmarks and DART-split for large-scale benchmarks.

## B.3    DART-split on Large-scale Benchmark

In Table 8, we present the experimental results on CIFAR-100C-imb and ImageNet-C-imb of several imbalance ratios. We can observe that DART-split achieves consistent performance improvement of 16.7-39.8% on CIFAR-100C-imb of imbalance ratios 200-50000, and 0.2-4.8% on ImageNet-C-imb of imbalance ratios 1000-500000, respectively, when combined with BNAdapt. Specifically, on CIFAR-100C-imb, BNAdapt's performance significantly decreases as the imbalance ratio increases, falling well below that of NoAdapt. However, due to the effective prediction refinement by DART-split, it consistently outperforms NoAdapt by more than 4% across all imbalance ratios. DART-split can also be combined with existing TTA methods as reported in Table 8. We confirm that our proposed method consistently contributes to performance improvement in all scenarios. These experiments demonstrate that our proposed method effectively addresses the performance degradation issues of the classifiers using the BN layer, which were significantly impacted by label distribution shifts.

Table 8: Average accuracy (%) of DART-split-applied TTA methods on CIFAR-100C-imb and ImageNet-C-imb

| | CIFAR-100C-imb | | | | ImageNet-C-imb | | | |
|---|---|---|---|---|---|---|---|---|
| | IR1 | IR200 | IR500 | IR50000 | IR1 | IR1000 | IR5000 | IR500000 |
| NoAdapt | 41.5±0.3 | 41.5±0.3 | 41.4±0.3 | 41.5±0.3 | 18.0±0.0 | 18.0±0.1 | 18.0±0.0 | 18.1±0.1 |
| BNAdapt | 59.8±0.5 | 29.1±0.5 | 18.8±0.4 | 9.3±0.1 | 31.5±0.0 | 19.8±0.1 | 8.4±0.1 | 3.4±0.0 |
| BNAdapt+DART-split (ours) | 59.7±0.5 | 45.8±0.2 | 50.2±0.3 | 49.1±0.7 | 31.5±0.0 | 20.0±0.2 | 11.5±0.6 | 8.2±0.5 |
| TENT | 62.0±0.5 | 25.4±0.4 | 15.4±0.5 | 7.1±0.1 | 43.1±0.2 | 18.7±0.2 | 4.3±0.1 | 1.2±0.0 |
| TENT+DART-split (ours) | 61.9±0.5 | 43.9±0.4 | 48.7±0.5 | 47.7±0.7 | 43.1±0.2 | 19.2±0.2 | 9.0±0.7 | 6.1±0.5 |
| SAR | 61.0±0.6 | 26.7±0.4 | 16.5±0.4 | 7.8±0.1 | 40.5±0.1 | 24.3±0.1 | 9.6±0.1 | 3.5±0.0 |
| SAR+DART-split (ours) | 60.9±0.6 | 44.7±0.3 | 50.0±0.3 | 49.1±0.7 | 40.5±0.1 | 24.4±0.2 | 12.1±0.6 | 8.3±0.6 |
| ODS | 62.4±0.4 | 49.6±0.5 | 43.7±0.4 | 38.7±0.3 | 43.5±0.2 | 30.3±0.1 | 14.6±0.3 | 9.1±0.1 |
| ODS+DART-split (ours) | 62.4±0.4 | 52.5±0.7 | 50.7±0.7 | 49.3±0.3 | 43.5±0.2 | 30.3±0.2 | 17.1±0.6 | 13.3±0.6 |

## C   More Related Works

### C.1   TTA method Utilizing Intermediate Time

Some recent works (Choi et al., 2022; Lim et al., 2022; Park et al., 2023) try to prepare an unknown test-time distribution shift by utilizing the training dataset at the time after the training phase and before the test time, called intermediate time. SWR (Choi et al., 2022) and TTN (Lim et al., 2022) compute the importance of each layer in the trained model during intermediate time and prevent the important layers from significantly changing during test time. SWR and TTN compute the importance of each layer by computing cosine similarity between gradient vectors of training data and its augmented data. TTN additionally updates the importance with subsequent optimization using cross-entropy. Layers with lower importance are encouraged to change significantly during test time, while layers with higher importance are constrained to change minimally. On the other hand, our method DART trains a prediction refinement module during intermediate time by experiencing several batches with diverse class distributions and learning how to modify the predictions generated by pre-trained classifiers to mitigate the negative effects caused by the class distribution shift of each batch.

### C.2   TTA methods considering Sample-wise Relationships

Some recent works (Boudiaf et al., 2022; Iwasawa & Matsuo, 2021; Jang et al., 2022) focus on prediction modification using the nearest neighbor information based on the idea that nearest neighbors in the embedding space of the trained classifier share the same label. T3A (Iwasawa & Matsuo, 2021) replaces the last linear layer of the trained classifier with the prototypical classifier, which predicts the label of test data to the nearest prototype representing each class in the embedding space. LAME (Boudiaf et al., 2022) modifies the prediction of test data by Laplacian-regularized maximum likelihood estimation considering clustering information.

### C.3   TTA methods considering Class-wise Relationships

Some TTA methods (Iwasawa & Matsuo, 2021; Kang et al., 2023; Zhang et al., 2023) focus on preserving class-wise relationships as domain-invariant information during test time. For instance, the method in (Kang et al., 2023) aims to minimize differences between the class-wise relationships of the test domain and the stored one from the training domain. CRS (Zhang et al., 2023) estimates the class-wise relationships and embeds the source-domain class relationship in contrastive learning. While these methods utilize class-wise relationships to prevent their deterioration during test-time adaptation, DART takes a different approach by focusing on directly modifying the predictions. DART considers class-wise confusion patterns to refine predictions, effectively addressing performance degradation due to label distribution shifts, without explicitly enforcing preservation of class-wise relationships.

### C.4 Loss Correction Methods for Learning with Label Noise

In learning with label noise (LLN), it is assumed that there exists a noise transition matrix $T$, which determines the label-flipping probability of a sample from one class to other classes. For LLN, two main strategies have been widely used in estimating $T$: 1) using anchor points (Xia et al., 2019; Yao et al., 2020), which are defined as the training examples that belong to a particular class almost surely, and 2) using the clusterability of nearest neighbors of a training example belonging to the same true label class (Zhu et al., 2021). LLN uses the empirical pseudo-label distribution of the anchor points or nearest neighbors to estimate $T$.

For TTA, on the other hand, the misclassification occurs not based on a fixed label-flipping pattern, but from the combination of covariate shift and label distribution shift. To adjust the pre-trained model against the covariate shifts, most TTA methods apply the BN adaptation, which updates the Batch Norm statistics using the test batches. However, when there exists a label distribution shift in addition to the covariate shift, since the updated BN statistics follow the test label distribution, it induces bias in the classier (by pulling the decision boundary closer to the head classes and pushing the boundary farther from the tail classes as in Appendix 2). Thus, the resulting class-wise confusion pattern depends not only on the class-wise relationship in the embedding space but also on the classifier bias originated from the label distribution shift and the updated BN statistics. Such a classifier bias has not been a problem for LLN, where we do not modify the BN statistics of the classifier at the test time.

Our proposed method, DART, focuses on this new class-wise confusion pattern and is built upon the idea that if the module experiences various batches with diverse class distributions before the test time, it can develop the ability to refine inaccurate predictions resulting from label distribution shifts.

## D Theorectical Analysis

### D.1 Motivating Toy Example - Confusion Patterns Reflecting the Class Relationship by BNAdapt

Table 9: Notation and definitions used in Section D.1

| Symbol | Meaning |
|---|---|
| $\mathcal{N}(\mu, \sigma^2 I_2)$ | Multivariate Gaussian distribution with mean $\mu$ and variance $\sigma^2$ |
| $\mu_i$ | Mean vector of class $i$ ($i = 1, 2, 3, 4$) |
| $\sigma^2$ | Variance shared across all classes |
| $d, \beta$ | Constants controlling class separation ($d > 1, \beta > 1$) |
| $\Delta$ | Test-time distribution shift vector |
| $h(\cdot)$ | Bayes classifier for prediction |
| $\bar{h}(\cdot)$ | Mean-centered Bayes classifier for prediction |
| $p_{\mathrm{tr}}(y)$ | Class prior probability during training (uniform: $1/4$) |
| $p_{\mathrm{te}}(y)$ | Class prior probability during testing (imbalanced distribution) |
| $\mu_i'$ | Shifted mean of class $i$ after test-time mean centering |
| $\Phi(\cdot)$ | Cumulative distribution function (CDF) of the standard normal distribution |

To understand the effects of test-time distribution shifts, we consider a Bayes classifier for four-class Gaussian mixture distribution with mean centering, which mimics batch normalization. Let the distribution of class $i$ be $\mathcal{N}(\mu_i, \sigma^2 I_2)$ at training time for $i = 1, 2, 3$, and 4, where $\mu_i \in \mathbb{R}^2$ is the mean of each class distribution. We set the mean of each class as $\mu_1 = (d, \beta d), \mu_2 = (-d, \beta d), \mu_3 = (d, -\beta d)$, and $\mu_4 = (-d, -\beta d)$, where $\beta$ controls the distances between the classes, and we assume that $\beta > 1$. Moreover, we assume that the four classes have the same prior probability at training time, $i.e., p_{\mathrm{tr}}(y = i) = 1/4, i = 1, 2, 3$, and 4. Since the class priors for the training data are equal, the Bayes classifier $h$ predicts $x$ to the class $i$ when

$$p_{\mathrm{tr}}(x|y = i) > p_{\mathrm{tr}}(x|y = j), \quad j \neq i \tag{9}$$

due to Bayes' rule. Then, we have

$$h(x) = \begin{cases} 1, & \text{if } x_1 > 0, x_2 > 0; \\ 2, & \text{if } x_1 < 0, x_2 > 0; \\ 3, & \text{if } x_1 > 0, x_2 < 0; \\ 4, & \text{if } x_1 < 0, x_2 < 0. \end{cases} \tag{10}$$

Initially, we assume that the input data distribution is shifted by $\Delta \in \mathbb{R}^2$ at test time as studied in prior works such as Stojanov et al. (2021); Yi et al. (2023). Specifically, the distribution of class $i$ is modeled as $\mathcal{N}(\mu_i + \Delta, \sigma^2 I_2)$ for $i = 1, 2, 3$, and 4 at the test time. Employing mean centering moves the distribution of class $i$ from $\mathcal{N}(\mu_i + \Delta, \sigma^2 I_2)$ to $\mathcal{N}(\mu_i, \sigma^2 I_2)$. Consequently, we demonstrate that the mean centering effectively mitigates the test-time input data distribution shift.

Furthermore, we consider a scenario where the class distribution is also shifted along with the input data distribution. We assume that the class distribution is imbalanced, similar to the long-tailed distribution, as

$$p_{\text{te}}(y = 1) = p, \tag{11}$$
$$p_{\text{te}}(y = 2) = 1/4, \tag{12}$$
$$p_{\text{te}}(y = 3) = 1/4, \tag{13}$$
$$p_{\text{te}}(y = 4) = 1/2 - p. \tag{14}$$

Assume that $1/4 < p < 1/2$. Due to the mean centering, the distribution of class $i$ is shifted to $\mathcal{N}(\mu_i', \sigma^2 I_2)$, where $\mu_i'$ is the shifted class mean as follows:

$$\mu_1' = ((3/2 - 2p)d, (3/2 - 2p)\beta d), \tag{15}$$
$$\mu_2' = ((-1/2 - 2p)d, (3/2 - 2p)\beta d), \tag{16}$$
$$\mu_3' = ((3/2 - 2p)d, (-1/2 - 2p)\beta d), \tag{17}$$
$$\mu_4' = ((-1/2 - 2p)d, (-1/2 - 2p)\beta d). \tag{18}$$

Then, the probability that the samples from class 1 are wrongly classified to class 2 can be computed as

$$\Pr_{x \sim \mathcal{N}(\mu_1 + \Delta, \sigma^2 I_d)}[h(\text{Norm}(x)) = 2] = \Pr_{x \sim \mathcal{N}(\mu_1', \sigma^2 I_d)}[h(x) = 2] \tag{19}$$

$$= \Pr_{x = (x_1, x_2) \sim \mathcal{N}(\mu_1', \sigma^2 I_2)}[x_1 < 0, x_2 > 0] \tag{20}$$

$$= \Phi\left(-\frac{(3/2 - 2p)d}{\sigma}\right)\left\{1 - \Phi\left(-\frac{(3/2 - 2p)\beta d}{\sigma}\right)\right\}, \tag{21}$$

where Norm is a mean centering function and $\Phi$ is the standard normal cumulative density function. Similarly, the probability that the samples from class 2 are wrongly classified to class 1 can be computed as

$$\Pr_{x \sim \mathcal{N}(\mu_2 + \Delta, \sigma^2 I_d)}[h(\text{Norm}(x)) = 1] = \left\{1 - \Phi\left(-\frac{(-1/2 - 2p)d}{\sigma}\right)\right\}\left\{1 - \Phi\left(-\frac{(3/2 - 2p)\beta d}{\sigma}\right)\right\}. \tag{22}$$

Since $1/4 < p < 1/2$, we have $\Pr_{x \sim \mathcal{N}(\mu_1 + \Delta, \sigma^2 I_d)}[\bar{h}(x) = 2] > \Pr_{x \sim \mathcal{N}(\mu_2 + \Delta, \sigma^2 I_d)}[\bar{h}(x) = 1]$, where $\bar{h}(\cdot) := h(\text{Norm}(\cdot))$ is modeled BN-adapted classifier. With similar computations, we can obtain $\Pr_{x \sim \mathcal{N}(\mu_1 + \Delta, \sigma^2 I_d)}[\bar{h}(x) = i] > \Pr_{x \sim \mathcal{N}(\mu_i + \Delta, \sigma^2 I_d)}[\bar{h}(x) = 1], \forall i = 2, 3$, and 4. In other words, the probability that the samples from the class of a larger number of samples are confused to the rest of the classes is greater than the inverse direction.

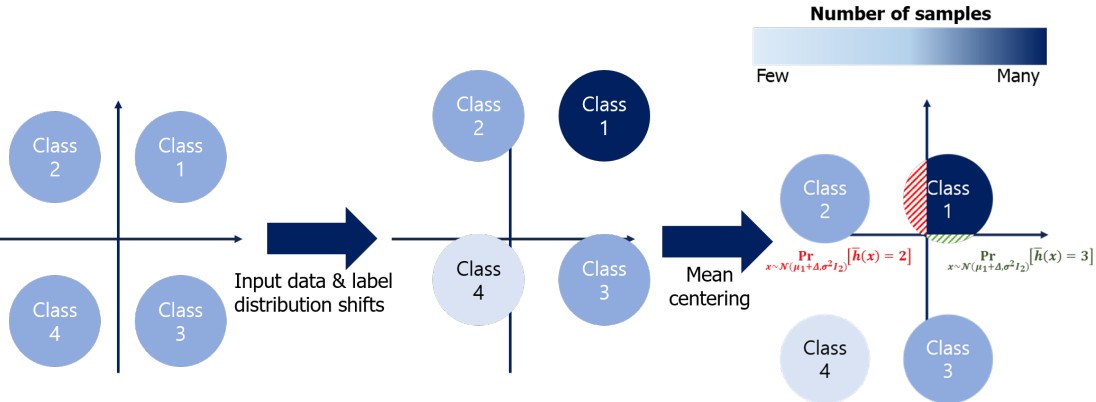

Figure 8: Illustration of the 4-class Gaussian mixture model from Section D.1. The two lines indicate the Bayes classifier decision boundaries, partitioning the feature space into regions corresponding to different classes. The patterned areas highlight regions where misclassifications occur. Specifically, the red patterned area denotes regions where samples from Class 1 are misclassified as Class 2, while the green patterned area represents regions where samples from Class 1 are misclassified as Class 3. Two notable confusion patterns can be observed when both input data and label distribution shifts occur, and mean centering is applied. First, Class 1, which has a larger sample size, is more prone to misclassification into other classes compared to smaller classes. Second, misclassifications are more likely to occur toward close classes in the feature space. For instance, Class 1 is more frequently confused with the closer class (Class 2) than with other classes.

The probability that the samples from class 1 are wrongly classified by $\bar{h}$ as class 2,3, and 4 can be calculated as follows:

$$\Pr_{x \sim \mathcal{N}(\mu_1+\Delta, \sigma^2 I_d)}[\bar{h}(x) = 2] = \Phi\left(-\frac{(3/2-2p)d}{\sigma}\right)\left\{1 - \Phi\left(-\frac{(3/2-2p)\beta d}{\sigma}\right)\right\}, \tag{23}$$

$$\Pr_{x \sim \mathcal{N}(\mu_1+\Delta, \sigma^2 I_d)}[\bar{h}(x) = 3] = \Phi\left(-\frac{(3/2-2p)\beta d}{\sigma}\right)\left\{1 - \Phi\left(-\frac{(3/2-2p)d}{\sigma}\right)\right\}, \tag{24}$$

$$\Pr_{x \sim \mathcal{N}(\mu_1+\Delta, \sigma^2 I_d)}[\bar{h}(x) = 4] = \Phi\left(-\frac{(3/2-2p)\beta d}{\sigma}\right)\Phi\left(-\frac{(3/2-2p)d}{\sigma}\right). \tag{25}$$

Note that $\Phi\left(-\frac{(3/2-2p)\beta d}{\sigma}\right)$ has the following properties: Since $1/4 < p < 1/2$, $\Phi\left(-\frac{(3/2-2p)\beta d}{\sigma}\right) < 1/2$; Since $\beta > 1$, $\Phi\left(-\frac{(3/2-2p)\beta d}{\sigma}\right) < \Phi\left(-\frac{(3/2-2p)d}{\sigma}\right)$; $\frac{\partial}{\partial p}\Phi\left(-\frac{(3/2-2p)\beta d}{\sigma}\right) = C_1 \beta \exp\left(-\frac{(3/2-2p)^2 \beta^2 d^2}{2\sigma^2}\right)$, where $C_1$ is a positive constant, independent of $p$ and $\beta$, which decreases as $\beta$ grows for $\beta > \frac{\sigma}{(3/2-2p)d}$.

Thus, we can say that

(1) The probability that the samples from the head class (class 1) are confused to tail classes is greater than the reverse direction, specifically, $\Pr_{x \sim \mathcal{N}(\mu_1+\Delta, \sigma^2 I_d)}[\bar{h}(x) = i] > \Pr_{x \sim \mathcal{N}(\mu_i+\Delta, \sigma^2 I_d)}[\bar{h}(x) = 1], \forall i \neq 1$, where $\bar{h}(\cdot) := h(\text{Norm}(\cdot))$ is a composition of mean-centering function and a Bayes classifier obtained using training dataset, modeling the BN-adapted classifier.

(2) The probability that a sample from the head class is confused to the closer class is larger than that to the farther classes. Specifically, $\Pr_{x \sim \mathcal{N}(\mu_1+\Delta, \sigma^2 I_d)}[\bar{h}(x) = 2] > \Pr_{x \sim \mathcal{N}(\mu_1+\Delta, \sigma^2 I_d)}[\bar{h}(x) = 3] > \Pr_{x \sim \mathcal{N}(\mu_1+\Delta, \sigma^2 I_d)}[\bar{h}(x) = 4]$.

(3) As the class distribution imbalance $p > 1/4$ increases, the rate at which misclassification towards spatially closer classes increases is greater than towards more distant classes. Specifically, $\frac{\partial}{\partial p}\Pr_{x \sim \mathcal{N}(\mu_1+\Delta, \sigma^2 I_d)}[\bar{h}(x) = 2] > \frac{\partial}{\partial p}\Pr_{x \sim \mathcal{N}(\mu_1+\Delta, \sigma^2 I_d)}[\bar{h}(x) = 3]$ when $2\sigma < d$.

The illustration of the class-wise confusion pattern caused by test-time input data and label distribution shift in a 4-class Gaussian mixture model can be found in Figure 8. The effects of test-time label distribution shift can be consistently observed not only in this toy example but also in real dataset experiments, e.g., CIFAR-10C-LT (Section 2).

## D.2  Meaning of DART's Logit Refinement Scheme

DART outputs a square matrix $W$ of size $K$ and a vector $b$ of size $K$ to correct the inaccurate BN-adapted predictions caused by the test time distribution shifts through an affine transformation of the classifier output (logit). Since the affine transformation can be interpreted as a simple matrix multiplication, achieved by modifying the input vector by adding 1 and including the vector into the square matrix, we focus on analyzing the theoretical meaning of the square matrix $W$ generated by DART. To understand the theoretical meaning of $W$ obtained by DART's training, we consider $K$-class classification problem with mean centering, which has a similar effect as the batch normalization as we discussed in Section D.1. As demonstrated in Section D.1, the mean centering effectively mitigates a test-time input data distribution shift. Thus, we focus only on the label distribution shifts in the following analysis. Let the distribution of class $i$ be $\mathcal{N}(\mu_i, \sigma I_d)$. Generally, the classifier output (logit) is computed as the product of features and classifier weights. With the mean centering, the logit for an example $x$ for class $i$ can be represented by the product of mean-centered features and mean-centered class centroids, i.e.,

$$l(x)_i = (x - \sum_{a \in [K]} p_a \mu_a)(\mu_i - \sum_{b \in [K]} p_b \mu_b)^T, \forall i \in [K] \tag{26}$$

where $p = [p_1, p_2, \ldots, p_K] \in \mathbb{R}^{1 \times K}$ is the label distribution of training dataset. In the matrix form, the logit for $x$ can be computed as

$$l(x) = (x - p\mu)(\mu - 1_{K \times 1} p\mu)^T = (x - p\mu)\mu^T (I_K - 1_{K \times 1} p)^T, \tag{27}$$

where $1_{K \times 1}$ is a matrix of size $K \times 1$ for which all the elements are 1, and $\mu = [\mu_1, \mu_2, \ldots, \mu_K]^T \in \mathbb{R}^{K \times d}$ is the concatenated class centroids. We assume that the logits are robust to the changes in the class distribution of the training dataset, i.e.,

$$l(x) = (x - p\mu)\mu^T (I_K - 1_{K \times 1} p)^T = (x - r\mu)\mu^T (I_K - 1_{K \times 1} r)^T, \tag{28}$$

when $r = [r_1, r_2, \ldots, r_K]$ satisfies $r_i \geq 0$ and $\sum_i r_i = 1$.

At the test time, when the label distribution is shifted from $p = [p_1, p_2, \ldots, p_K]$ to $q = [q_1, q_2, \ldots, q_K]$, the mean-centered logit for an example $x$ after the shift can be computed as

$$l(x) = (x - q\mu)\mu^T (I_K - 1_{K \times 1} p)^T, \tag{29}$$

since only the features are newly mean-centered while the classifier weights are unchanged similar to BNAdapt (Nado et al., 2020; Schneider et al., 2020).

DART learns the square matrix $W \in \mathbb{R}^{K \times K}$ to refine the BN-adapted classifier's output to the original one by multiplying it with $W$. Specifically, we obtain $W^*$ which minimizes the below optimization problem when we use the L2 norm,

$$W^* = \underset{W}{\arg\min} \, \mathbb{E}_{x \in \mathbb{R}^{1 \times d}} [\|(x - p\mu)\mu^T (I_K - 1_{K \times 1} p)^T - (x - q\mu)\mu^T (I_K - 1_{K \times 1} p)^T W\|_2^2] \tag{30}$$

$$\approx \underset{W}{\arg\min} \, \mathbb{E}_{x \in \mathbb{R}^{1 \times d}} [\|(x - q\mu)\mu^T (I_K - 1_{K \times 1} q)^T - (x - q\mu)\mu^T (I_K - 1_{K \times 1} p)^T W\|_2^2] \tag{31}$$

$$= \underset{W}{\arg\min} \, \mathbb{E}_{x \in \mathbb{R}^{1 \times d}} [\|(x - q\mu)\{\mu^T (I_K - 1_{K \times 1} q)^T - \mu^T (I_K - 1_{K \times 1} p)^T W\}\|_2^2]. \tag{32}$$

Since we match these two logits for any $x$, we can rewrite the above equation as follows:

$$W^* \approx \underset{W}{\arg\min} \, \|\mu^T (I_K - 1_{K \times 1} q)^T - \mu^T (I_K - 1_{K \times 1} p)^T W\|_2^2. \tag{33}$$

Since $d$ is greater than $K$ generally, the least-square solution for the above equation is

$$W^* = \{(I_K - 1_{K \times 1}p)\mu\mu^T(I_K - 1_{K \times 1}p)^T\}^{-1}(I_K - 1_{K \times 1}p)\mu\mu^T(I_K - 1_{K \times 1}q)^T, \qquad (34)$$

with an assumption that $(I_K - 1_{K \times 1}p)\mu\mu^T(I_K - 1_{K \times 1}p)^T$ is invertible. Then, we can observe that $W^*$ is determined by only three different components: the label distributions of training and test datasets ($p$ and $q$, resp.) and the class-wise relationship, i.e., the relationship among the class centroids ($\mu\mu^T \in \mathbb{R}^{K \times K}$). In particular, we can find that $W^*$ is related to the relationships among class centroids, not the exact locations of the class centroids.

In contrast to the re-weighting methods (Garg et al., 2023; Hong et al., 2021; Menon et al., 2020), which do not take into account class relationships for prediction refinement, DART considers class relationships for prediction refinement as evidenced by the above $W^*$. Thus, DART can address the errors with confusion patterns arising from test-time distribution shifts, in which the label distribution is also shifted.

## E    Confusion Matrices of BN-adapted Classifiers on CIFAR-10C-LT

In Figure 1, we present confusion matrices of the BN-adapted classifiers for three different corruption types (Gaussian noise, defocus blur, snow) from each of the three corruption categories (noise, blur, and weather) among 15 pre-defined corruptions. In Figure 9-11, we present confusion matrices for all 15 corruptions including clean CIFAR-10 test dataset with various levels of label distribution shifts ($\rho = 1, 10, 100$), respectively. While test accuracy may not perfectly be matched across the corruptions, we can observe that (1) the accuracy in head classes (with smaller class index) is significantly decreased and (2) confusion patterns tend to be consistent across different corruptions as we described in Section 2. Additionally, Figure 9-11 reveal increasingly pronounced class-wise confusion patterns as the imbalance ratio $\rho$ rises from 1 to 100. Based on these observations, we conjecture that the prediction refinement module trained only on the training data at the intermediate time remains consistently effective in modifying predictions for the test datasets.

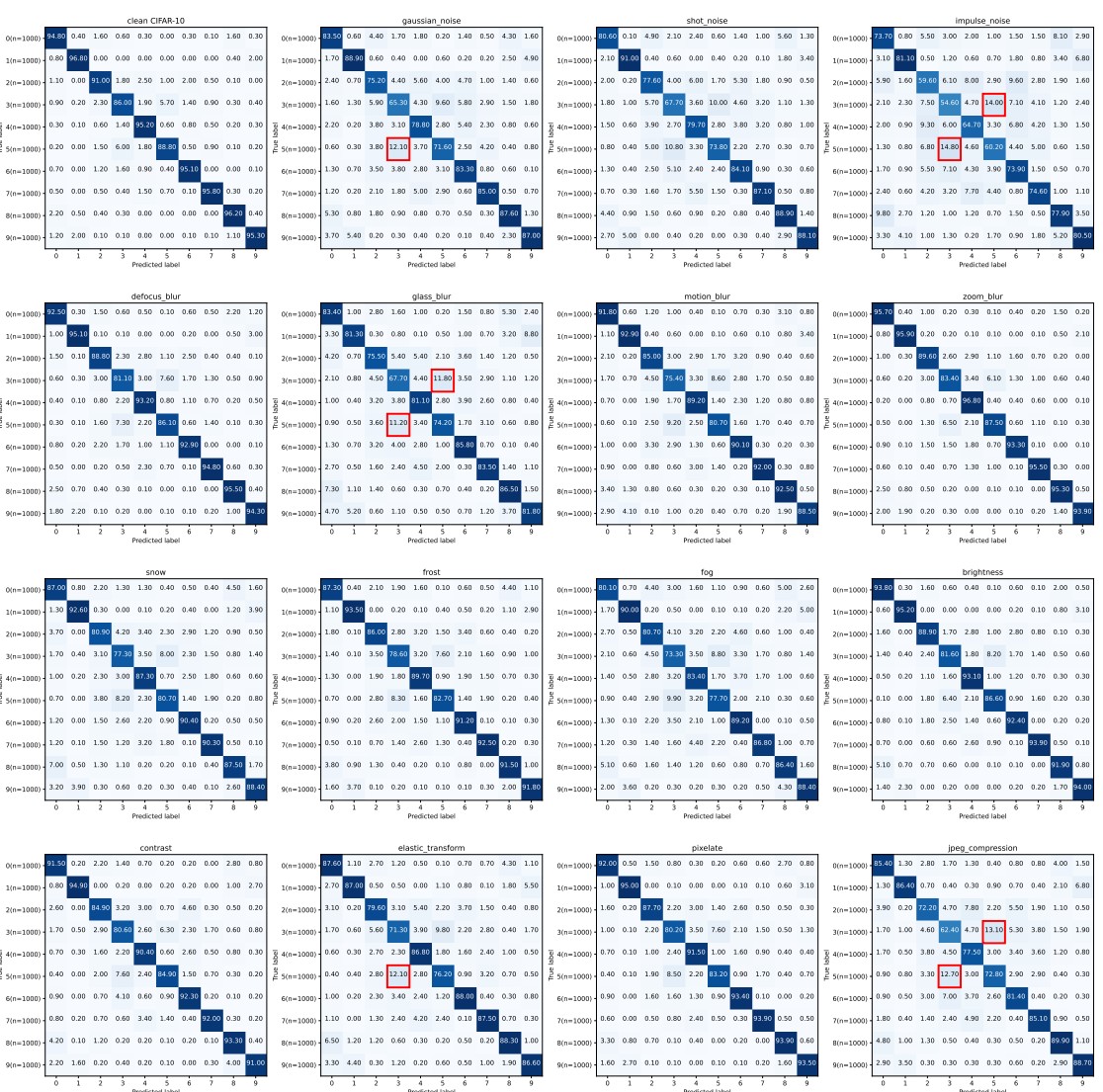

Figure 9: Confusion matrices of BN-adapted classifiers on CIFAR-10C-LT with $\rho = 1$. We mark the cases where the confusion rate exceeds 11% with red squares.

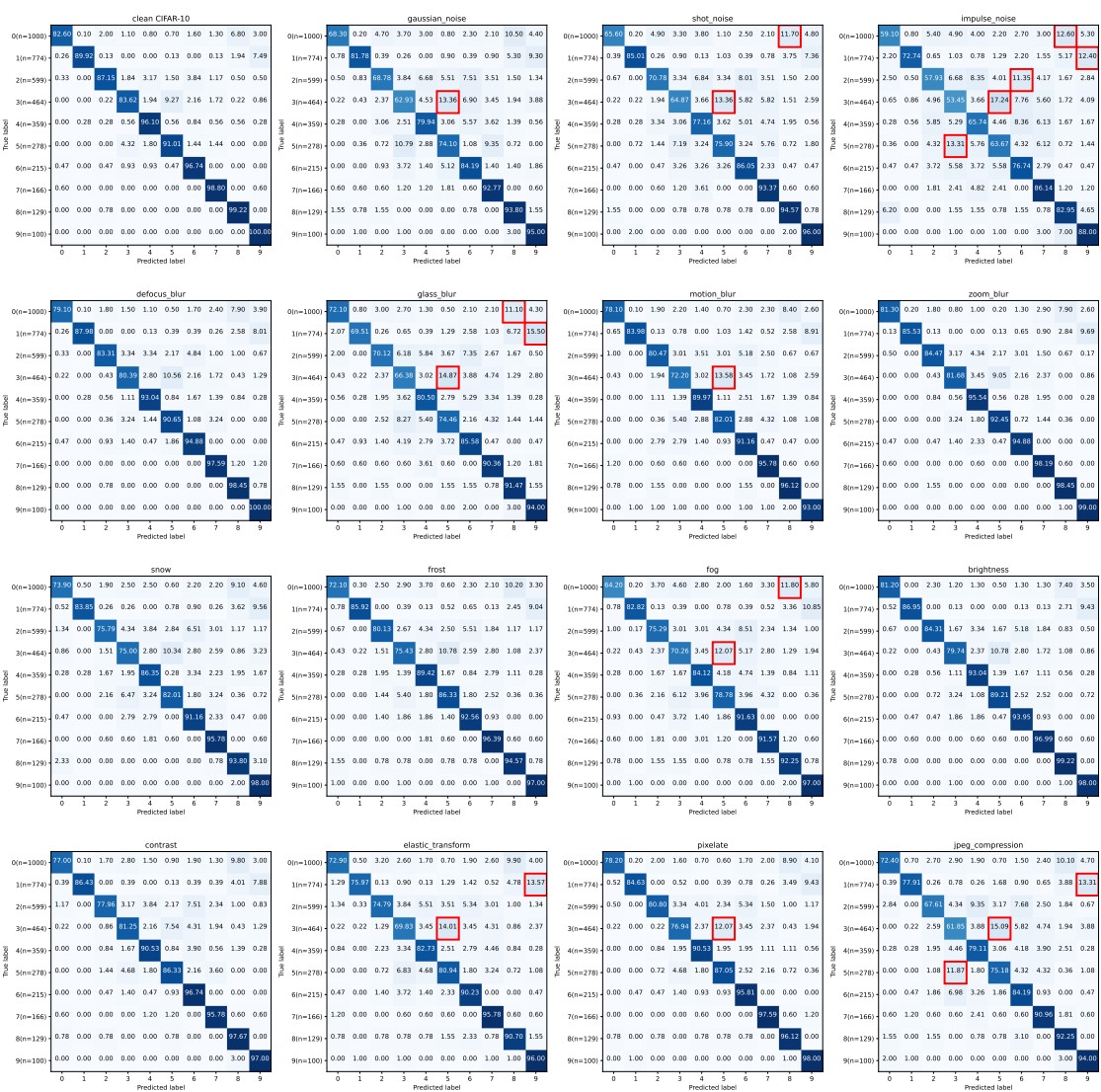

Figure 10: Confusion matrices of BN-adapted classifiers on CIFAR-10C-LT with $\rho = 10$. We mark the cases where the confusion rate exceeds 11% with red squares. We can observe notable accuracy decreases in classes with large amounts of data, and similar confusing patterns regardless of the corruption types. Compared to $\rho = 1$, we can observe more pronounced class-wise confusion patterns.

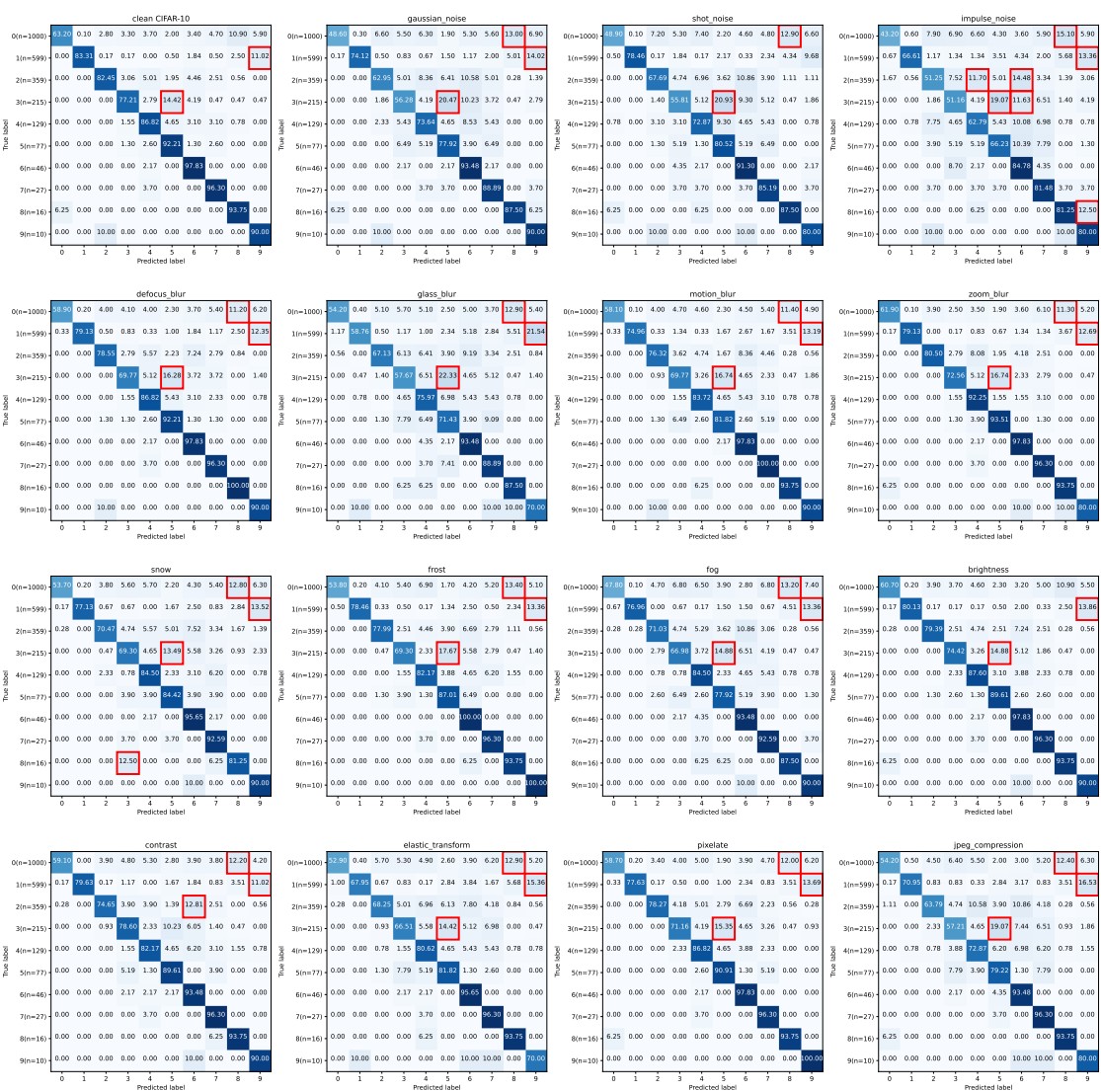

Figure 11: Confusion matrices of BN-adapted classifiers on CIFAR-10C-LT with $\rho = 100$. We mark the cases where the confusion rate exceeds 11% with red squares. We can observe notable accuracy decreases in classes with large amounts of data, and similar confusing patterns regardless of the corruption types under the class distribution shift. Compared to $\rho = 1, 10$, we can observe clearer class-wise confusion patterns.

## F Prediction Refinement Module Outputs on CIFAR-10C-LT

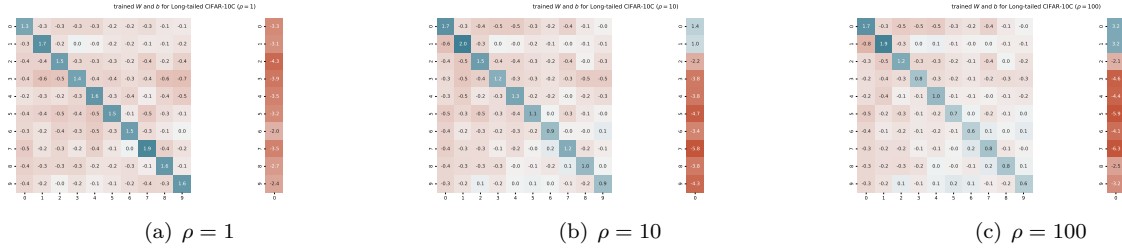

(a) $\rho = 1$      (b) $\rho = 10$      (c) $\rho = 100$

Figure 12: Prediction refinement module outputs on CIFAR-10C-LT of $\rho = 1, 10, 100$ without regularization (*i.e.*, $\alpha = 0$)

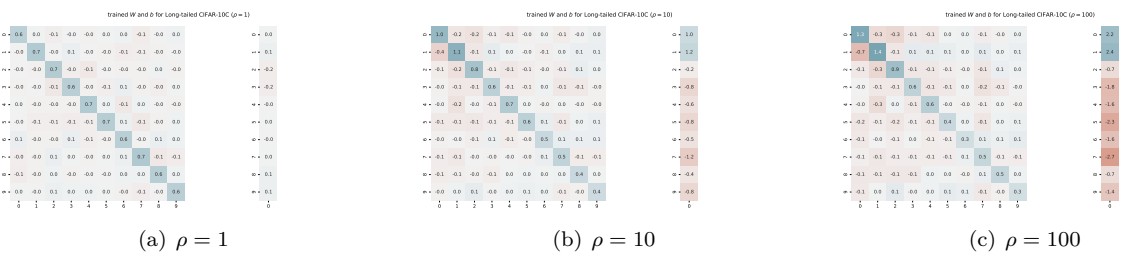

(a) $\rho = 1$      (b) $\rho = 10$      (c) $\rho = 100$

Figure 13: Prediction refinement module outputs on CIFAR-10C-LT of $\rho = 1, 10, 100$ when $\alpha = 0.1$

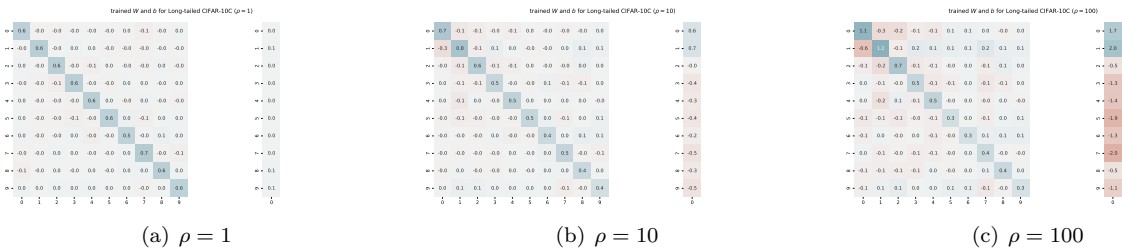

(a) $\rho = 1$      (b) $\rho = 10$      (c) $\rho = 100$

Figure 14: Prediction refinement module outputs on CIFAR-10C-LT of $\rho = 1, 10, 100$ when $\alpha = 1$

In Figures 12-14, we present the $W$ and $b$, generated by $g_\phi$ when inputting the averaged pseudo label distribution and prediction deviation computed on the whole test data for CIFAR-10C-LT under Gaussian noise with different $\alpha$ values (0, 0.1, and 1) in equation 3. The (a), (b), and (c) of each figure show the outputs of $g_\phi$ when class imbalance $\rho$ is set to 1, 10, and 100, respectively. In Figures 12-14, we commonly observe that both $W$ and $b$ assign high values for the head classes (classes 0-1). This prediction refinement of DART can correct the predictions of instances from the head classes that are misclassified to other classes, recovering the reduced test accuracy of the head classes (shown in Figure 1). As shown in Figure 15 of Appendix G, this prediction refinements scheme of DART can modify the averaged pseudo label distribution to be closely matched with the ground truth label distribution, which is originally significantly disagreed.

Moreover, by comparing Figure 12 and Figure 13- 14, we can observe the impact of the regularization of equation 2, which encourages $g_\phi$ to output an identity matrix and a zero vector for nearly class-balanced batches. Specifically, the values of the off-diagonal of $W$ and $b$ become almost 0 for $\rho = 1$. The classifier logit of each data point reflects not only the class to which the data point belongs but also the characteristics of the individual data point (Hinton et al., 2015). For example, even for data points in the same class, the classifier logit can vary depending on factors such as background or color, thereby containing information

specific to each data point. Therefore, in the absence of a test-time class distribution shift, where there is no class-wise confusion pattern to reverse, the classifier logit modified by the prediction refinement poses a risk of damaging the information of each data point, even if the prediction itself remains unchanged. By adding the regularization, we can prevent the unintended logit refinements for the nearly class-balanced batches since the batches might have no common class-wise confusion patterns originated from label distribution shifts (Figure 9).

We experimentally demonstrate the effect of regularization (equation 2) and present the results in Table 10. Without regularization ($\alpha = 0$), DART-applied TENT shows approximately 2% lower performance under no label distribution shift ($\rho = 1$) compared to other $\alpha$ values. It indicates that DART without regularization makes unnecessary adjustments, harming the subsequent test-time adaptation when there is no class distribution shift. On the other hand, the TTA performance remains stable across all $\rho$s when $\alpha$ is set between 0.01 and 1. This result demonstrates the effectiveness of regularization and the robustness of DART to changes in $\alpha$.

Table 10: Accuracy (%) of DART-applied BNAdapt and TENT with different $\alpha$ values on CIFAR-10C-LT.

| Method | $\alpha$ | CIFAR-10C-LT | | |
|---|---|---|---|---|
| | | $\rho = 1$ | $\rho = 10$ | $\rho = 100$ |
| BNAdapt + DART | 0 | 85.3 | 85.5 | 85.8 |
| | 0.01 | 85.4 | 85.1 | 85.7 |
| | 0.1 (used) | 85.2 | 84.7 | 85.1 |
| | 0.5 | 85.3 | 83.9 | 84.8 |
| | 1 | 85.3 | 83.5 | 84.7 |
| TENT + DART | 0 | 84.4 | 86.2 | 88.3 |
| | 0.01 | 86.1 | 86.9 | 88.7 |
| | 0.1 (used) | 86.4 | 86.8 | 88.2 |
| | 0.5 | 86.6 | 86.5 | 87.9 |
| | 1 | 86.5 | 86.2 | 87.8 |

## G  More Experiments

Table 11: **Average accuracy (%) on CIFAR-10C-LT using condensed training data.** DART with condensed training data achieves similar performance gains as with original training data, improving BN-adapted classifiers under label distribution shifts without any degradation in performance when there is no label distribution shift.

| Method | Avail. data at int. time | $\rho = 1$ | $\rho = 10$ | $\rho = 100$ |
|---|---|---|---|---|
| BNAdapt | - | 85.2±0.0 | 79.0±0.1 | 67.0±0.1 |
| BNAdapt+DART | Training data | 85.2±0.1 | 84.7±0.1 | 85.1±0.3 |
| | Condensed tr. data | 85.2±0.0 | 82.4±0.3 | 81.4±0.4 |

### G.1  Reducing the Dependency of Labeled Training Data in the Intermediate Time Training of DART

As described in Section 5 and Appendix C, some recent TTA methods utilize the training data to prepare adaptations to unknown test-time distribution shifts during the intermediate time. DART also uses the training data to construct batches with various label distributions to learn how to detect label distribution shifts and correct the degraded predictions caused by the shifts. Under a strict situation in which training data is unavailable due to privacy issues, we can use condensed training data, which is small synthetic data that preserves essential classification features from the original training data while discarding private information.

It could prevent privacy leakage (Kang et al., 2023). We report experimental results on CIFAR-10C-LT for the case when using the condensed training data instead of the original one in Table 11. We use the publicly released condensed dataset of CIFAR-10 by Cazenavette et al. (2022) consisting of only 50 images for each class. Even when using the condensed data, DART achieves similar performance gains as when trained with the original data. Specifically, DART improves the degraded performance caused by label distribution shifts ($+3.4/14.4\%$ for $\rho = 10/100$) without any degradation in performance when there is no label distribution shift ($\rho = 1$). This demonstrates that DART training is possible without the original training data.

Table 12: **Average accuracy (%) on mixed CIFAR-10C.** Even in the mixed CIFAR-10C dataset, DART effectively mitigates performance degradation caused by label distribution shifts resulting from Dirichlet distribution sampling.

| Method | No label dist. shifts | Under label dist. shifts |
|---|---|---|
| NoAdapt | 71.7±0.0 | 71.7±0.0 |
| BNAdapt | 75.7±0.1 | 20.4±0.1 |
| BNAdapt+DART | 75.8±0.1 | 81.5±0.8 |
| ROID (Marsden et al., 2024) | 80.3±0.2 | 12.9±1.2 |
| ROID+DART | 80.4±0.1 | 63.8±2.3 |

## G.2 DART in Mixed Domain TTA Settings

To verify DART's effectiveness in a complex scenario, we consider a "mixed domain" setting. Here, corrupted test data from 15 types of corruptions are combined into a single test data, as in Marsden et al. (2024). Under this mixed domain setting, we consider both no label distribution shifts and label distribution shifts, where for label distribution shifts, the test data is non-i.i.d sampled using Dirichlet sampling with $\delta = 0.1$. We summarize the results for the original v.s. DART-applied TTA methods in Table 12. The results indicate that 1) DART effectively addresses the performance degradation of the BN-adapted classifier even in the mixed domain setup ($+61.1\%$), and 2) while ROID (Marsden et al., 2024), a state-of-the-art method in mixed domains, shows a significant performance drop of 67.4% under label distribution shifts, DART leads to an improvement of 50.9% in performance of ROID. DART's prediction refinement could be effective in mixed domains since it is based on the finding that the BN-adapted classifier's predictions exhibit consistent class-wise confusion patterns due to label distribution shifts, regardless of corruption type (Fig. 1). We expect that DART's prediction refinement will also be effective in other challenging situations where both covariate and label distribution shifts occur.

## G.3 Robustness to Changes in Test Batch Size

Table 13: Experiments on balanced CIFAR-10C with different batch sizes.

| Method | Test batch size | | | | | |
|---|---|---|---|---|---|---|
| | 4 | 8 | 16 | 32 | 64 | 200 |
| NoAdapt | 71.7±0.0 | 71.7±0.0 | 71.7±0.0 | 71.7±0.0 | 71.7±0.0 | 71.7±0.0 |
| BNAdapt | 71.5±0.3 | 76.7±0.1 | 81.1±0.1 | 83.4±0.1 | 84.5±0.0 | 85.2±0.0 |
| BNAdapt+DART (ours) | 74.3±0.1 | 80.4±0.1 | 82.0±0.1 | 83.5±0.1 | 84.6±0.1 | 85.2±0.1 |
| TENT | 19.7±0.6 | 37.6±1.2 | 60.2±1.2 | 75.7±0.5 | 82.5±0.3 | 86.3±0.1 |
| TENT+DART (ours) | 51.3±2.3 | 68.1±1.0 | 75.6±0.4 | 80.2±0.5 | 83.6±0.2 | 86.5±0.2 |

In Table 13, we evaluate the effectiveness of DART under small test batch sizes on balanced CIFAR-10C. Even if the label distribution of the test dataset remains balanced, when the test batch size becomes small,

class imbalance naturally occurs within the test batch as discussed in SAR (Niu et al., 2023). As the test batch size gets smaller from 200 to 4, the performance of BNAdapt gets worse. For example, the performance of BNAdapt is slightly worse than the one of NoAdapt when the test batch size is set to 4. However, DART successfully improves the degraded performance of BNAdapt by about 3% under a small test batch size (e.g. 4 or 8).

### G.4 Experimental Results using Different Hyperparameters on CIFAR-10C-LT

Table 14: Experimental results using different combinations of the hyperparameters on CIFAR-10C-LT

| | intermediate batch size | hidden dimension of $g_\phi$ | | | | | | | | | | | |
| | | 250 | | | 500 | | | 1000 (used) | | | 2000 | | |
| | | $\rho=1$ | $\rho=10$ | $\rho=100$ | $\rho=1$ | $\rho=10$ | $\rho=100$ | $\rho=1$ | $\rho=10$ | $\rho=100$ | $\rho=1$ | $\rho=10$ | $\rho=100$ |
|---|---|---|---|---|---|---|---|---|---|---|---|---|---|
| BNAdapt | | 85.2±0.0 | 79.0±0.1 | 67.0±0.1 | 85.2±0.0 | 79.0±0.1 | 67.0±0.1 | 85.2±0.0 | 79.0±0.1 | 67.0±0.1 | 85.2±0.0 | 79.0±0.1 | 67.0±0.1 |
| BNAdapt+DART (ours) | 32 | 85.0±0.0 | 83.4±0.3 | 83.1±1.2 | 85.0±0.0 | 83.3±0.4 | 83.5±1.2 | 85.2±0.0 | 82.9±0.3 | 84.0±0.3 | 85.2±0.0 | 82.3±0.3 | 84.1±0.7 |
| | 64 (used) | 85.3±0.0 | 84.8±0.1 | 84.3±0.5 | 85.2±0.1 | 84.7±0.2 | 84.8±0.2 | 85.2±0.1 | 84.7±0.1 | 85.1±0.3 | 85.3±0.0 | 84.6±0.2 | 85.1±0.4 |
| | 128 | 85.4±0.0 | 84.4±0.1 | 80.6±0.3 | 85.4±0.0 | 84.5±0.0 | 81.0±0.3 | 85.4±0.0 | 84.7±0.1 | 81.9±0.2 | 85.4±0.0 | 84.9±0.1 | 83.3±0.4 |

In Table 1, we report the experimental results when the intermediate batch size and the hidden dimension of $g_\phi$ are set to 64 and 1,000, respectively. In Table 14, we summarize the experimental results using different combinations of the hyperparameters on CIFAR-10C-LT. We can observe that the performance gain of DART is robust to changes in the hyperparameters for most cases. We observe that the performance improvement is about 14% when the intermediate batch size is set to 128 for CIFAR-10C of $\rho = 100$, while it shows about 18% when the intermediate batch size is set to 32 or 64. This is because the prediction refinement module $g_\phi$ does not experience sufficiently severe class imbalance under the same Dirichlet distribution when the intermediate batch size increases. For example, when there are two batches of the same size having class distributions [0.9, 0.1] and [0.1, 0.9], the combined batch has a class distribution of [0.5, 0.5]. Therefore, this can be addressed by modifying the Dirichlet distribution to experience severe class imbalance by reducing $\delta$ or $N_{\text{dir}}$.

Table 15: Performance of DART-applied BNAdapt on CIFAR-10C-LT with different intermediate-time training epochs

| | CIFAR-10C-LT | | |
| # of epochs | $\rho=1$ | $\rho=10$ | $\rho=100$ |
|---|---|---|---|
| 1 | 82.0 | 80.6 | 75.7 |
| 5 | 85.3 | 84.5 | 81.7 |
| 10 | 85.4 | 84.9 | 83.3 |
| 25 | 85.3 | 84.8 | 84.8 |
| 50 (used) | 85.2 | 84.7 | 85.1 |

We summarize the experimental results on the effect of training epochs for the prediction refinement module $g_\phi$ during the intermediate-time training in Table 15. We set the training epochs for intermediate-time training to 50 for CIFAR-10. To assess the convergence of $g_\phi$ training, we conduct experiments by reducing the number of training epochs from 1 to 25. Notably, training for only 10 epochs achieves comparable results to 50 epochs, with just a 1.8% gap observed for CIFAR-10C-LT with $\rho = 100$. This demonstrates the efficiency of DART's intermediate-time training, significantly enhancing its scalability for practical use.

### G.5 DART on CIFAR-10C-imb

We summarize the experimental results on CIFAR-10C under online imbalance setup in Table 16. We observe that the DART-applied TTA methods consistently show improved performance than naive TTA methods as shown in the experiments on CIFAR-10C-LT of Table 1. BNAdapt's performance decreases as the imbalance ratio increases, and it shows worse performance than NoAdapt when the imbalance ratios are set to 20-5000. However, DART-applied BNAdapt consistently outperforms NoAdapt across all imbalance ratios.

Table 16: Average accuracy (%) of DART-applied TTA methods on CIFAR-10C-imb

| | CIFAR-10c-imb | | | | |
|---|---|---|---|---|---|
| | IR1 | IR5 | IR20 | IR50 | IR5000 |
| NoAdapt | 71.6±0.1 | 71.7±0.1 | 71.7±0.1 | 71.6±0.1 | 71.7±0.1 |
| BNAdapt | 85.3±0.1 | 77.1±0.3 | 50.0±0.4 | 34.6±0.4 | 20.3±0.1 |
| BNAdapt+DART (ours) | 85.3±0.2 | 83.3±0.1 | 76.2±0.3 | 78.2±0.1 | 82.4±0.7 |
| TENT | 86.5±0.2 | 75.8±0.6 | 47.3±0.1 | 32.6±0.4 | 19.1±0.1 |
| TENT+DART (ours) | 86.5±0.3 | 83.2±0.1 | 73.6±0.3 | 71.4±0.6 | 71.5±1.3 |

Table 17: **Average accuracy (%) on DomainNet-126**. *X2Y* refers to a scenario where a pre-trained model, trained in domain *X*, is tested in domain *Y*. We can observe that DART achieved consistent performance improvements on DomainNet-126.

| Method | r2c | r2p | r2s | c2r | c2p | c2s | p2r | p2c | p2s | s2r | s2c | s2p | avg |
|---|---|---|---|---|---|---|---|---|---|---|---|---|---|
| BNAdapt | 52.65 | 61.57 | 46.38 | 63.12 | 48.68 | 47.93 | 73.54 | 52.67 | 50.96 | 66.32 | 57.86 | 57.56 | 56.60 |
| BNAdapt+DART | 53.40 | 62.20 | 46.95 | 66.07 | 51.74 | 50.21 | 75.05 | 54.49 | 52.00 | 69.73 | 60.73 | 59.82 | 58.53 (+1.93%) |

## G.6 DART on DomainNet-126

We test the effectiveness of DART on DomainNet-126, and the experimental results are summarized in Table 17. We follow the experimental setting described in Marsden et al. (2024) and set the test batch size to 32. We can observe that DART achieved consistent performance improvements of 1.93% on average on DomainNet-126. We would like to emphasize that DART consistently achieves performance gains in large-scale benchmarks such as OfficeHome and DomainNet. In the large-scale benchmarks, the batch size is generally significantly smaller than the number of classes. Then, even if the class distribution of the test dataset differs greatly from that of the training dataset, the change in class distribution within each test batch would be minimal. In such cases, the performance improvement due to DART seems to be less significant compared to small-scale datasets like CIFAR-10C-LT in Table 1, but DART is still effectively adjusting inaccurate predictions by label distribution shifts.

Table 18: **Average accuracy (%) of MEMO on CIFAR-10C-LT**. MEMO (stored BN) performs worse than NoAdapt while it does not show a performance decline under label distribution shifts. On the other hand, MEMO (adapted BN) exhibits performance degradation due to label distribution shifts, similar to BNAdapt. DART can effectively address these performance degradations.

| Method | $\rho = 1$ | $\rho = 10$ | $\rho = 100$ |
|---|---|---|---|
| NoAdapt | 71.7±0.0 | 71.3±0.1 | 71.4±0.2 |
| MEMO (stored BN) | 67.6±0.1 | 68.9±0.3 | 71.4±0.1 |
| MEMO (adapted BN) | 87.0±0.1 | 85.0±0.2 | 73.6±0.1 |
| MEMO (adapted BN) + DART | 86.8±0.3 | 87.8±0.1 | 87.1±0.4 |

## G.7 Applying DART to MEMO

We test the effectiveness of MEMO on CIFAR-10C-LT and try to combine MEMO and DART. MEMO fine-tunes classifiers to minimize the marginal entropy of the classifier outputs across different data augmentations. We consider two variants of MEMO introduced in (Goyal et al., 2022). The first variant, MEMO (stored BN), uses stored BN statistics and resets the classifier for each test batch. The second variant, MEMO (adapted BN), uses the BN statistics computed on each test batch and continuously adapts the classifier like TENT. In Table 18, we can observe that MEMO (stored BN) does not show a performance decline under

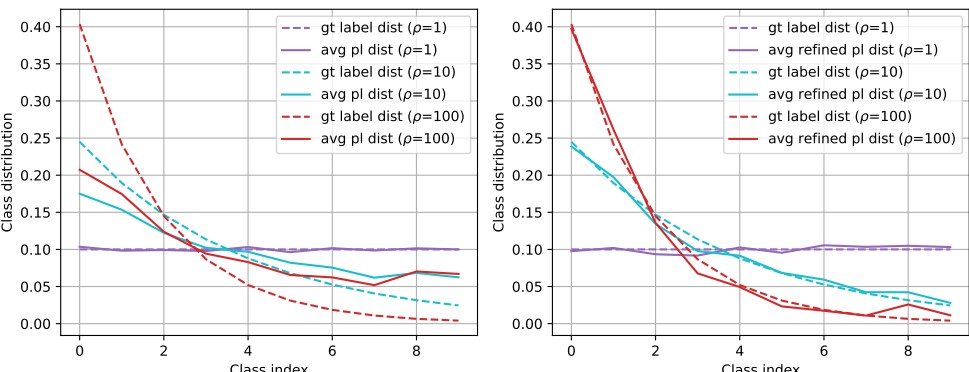

Figure 15: **Visualizations of ground truth and averaged pseudo label distributions of (left) BNAdapt and (right) DART-applied BNAdapt on CIFAR-10C-LT** The averaged refined pseudo label distribution (solid line) generated by the DART-applied BNAdapt closely matches the ground truth label distribution (dashed line), but the pseudo label distribution of BNAdapt does not match.

label distribution shifts on CIFAR-10C as shown in Zhao et al. (2023), but it performs worse than NoAdapt. On the other hand, MEMO (adapted BN) exhibits performance degradation due to label distribution shifts, similar to BNAdapt (-2/13.4% for $\rho = 10/100$). However, we can observe that DART can effectively address these performance degradations. Through the above experiments, we explored the potential for combining DART and MEMO (adapted BN).

### G.8 Comparison of Averaged Pseudo Label Distribution of BNAdapt and DART-applied BNAdapt

In Figure 15, we present the ground truth label distribution and the averaged pseudo label distributions generated by (left) the BN-adapted classifier and (right) the DART-applied BN-adapted classifier for CIFAR-10C-LT with different $\rho$s. In Figure 15 (left), we can observe that although the averaged pseudo-label distribution of the BN-adapted classifier does not perfectly match the ground truth label distribution, it still contains information about the direction and severity of the label distribution shift. However, except when $\rho$ is 1 (*i.e.,* balanced), the averaged pseudo label distribution differs significantly from the ground truth label distribution. Specifically, the averaged pseudo-label distribution is slightly closer to uniform compared to the ground truth label distribution. On the other hand, in Figure 15 (right), we observe that the averaged pseudo label distribution generated by the DART-applied BN-adapted classifier almost perfectly matches the ground truth label distribution. Through these experiments, we can see that the prediction refinement by DART can accurately estimate the label distribution, which is advantageous for subsequent test-time adaptation.

## H Transition Matrix Estimation by Noisy Label Learning Method

HOC (Zhu et al., 2021) estimates the noisy label transition matrix $T$, which represents the probabilities of clean labels being flipped to noisy labels, for a given noisy label dataset under the intuition that the nearest neighbor in the embedding space of a trained classifier $f_\theta$ shares the same ground truth label. In the case of noisy labels caused by random transitions, the noisy transition matrix can be predicted using feature clusterability or nearest neighbor information in the embedding space. However, as shown in Figure 16 (a) left, the nearest neighbors in the embedding space already have the same predictions of BN-adapted classifiers, making it difficult for HOC to estimate the noisy transition matrix using feature clusterability (Figure 16 (a) right). Thus, the estimated $T$ by HOC is similar to the identity matrix (Figure 16 (b)). In this paper, we theoretically and experimentally analyze the misclassification patterns and propose a TTA method DART to address performance degradation by learning the prediction refinement module which modifies the inaccurate BNAdapt predictions due to label distribution shifts.

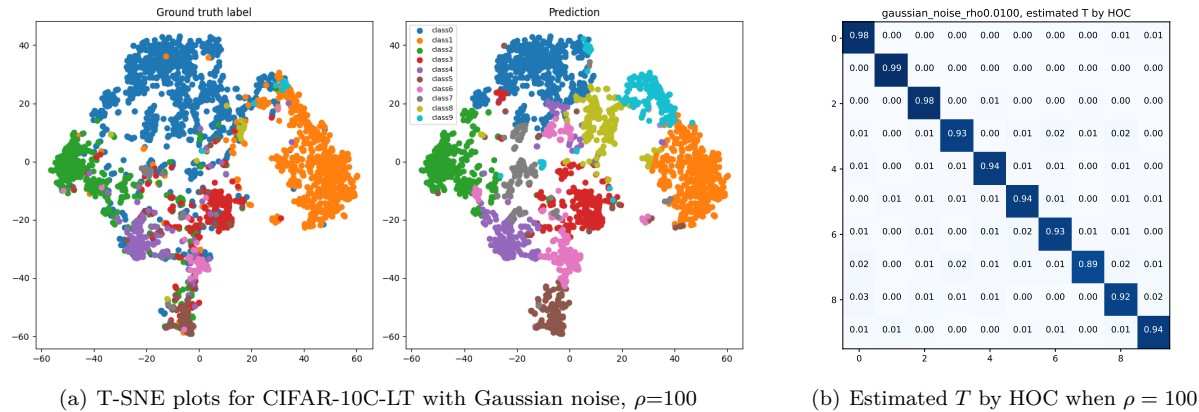

(a) T-SNE plots for CIFAR-10C-LT with Gaussian noise, $\rho$=100    (b) Estimated $T$ by HOC when $\rho = 100$

Figure 16: (a) T-SNE plots of test data with ground truth labels (left) and their predictions (right) for CIFAR-10C-LT with Gaussian noise, $\rho$=100 (b) Estimated $T$ by HOC for CIFAR-10C-LT with Gaussian noise, $\rho$=100

# I    Expansion to Vision Transformer (ViT)

Since DART can be applied to the adaptation of classifiers using batch normalization (BN), we believe that DART is applicable to Vision Transformer (ViT) if it uses BN as well. ViTs typically use Layer Normalization (LN) instead of BN. However, since BN generally leads to faster convergence and has shown strong performance in vision tasks, there have been attempts to replace LN with BN in ViTs (Chen et al., 2021). For instance, it was shown that replacing LN with BN in ViTs for the ImageNet classification task resulted in a 1% increase in classification performance, evaluated by linear probing (Chen et al., 2021). For the ViTs replacing LN with BN, we expect that applying DART will lead to significant improvements in the classification performance of the classifiers under label distribution shifts.

## J  Full Results

Table 19: Average accuracy (%) on CIFAR-10C-LT ($\rho = 1$)

| Method | gaussian_noise | shot_noise | impulse_noise | defocus_blur | glass_blur | motion_blur | zoom_blur | snow | frost | fog | brightness | contrast | elastic_transform | pixelate | jpeg_compression |
|---|---|---|---|---|---|---|---|---|---|---|---|---|---|---|---|
| NoAdapt | 46.4±0.0 | 51.6±0.0 | 27.1±0.0 | 90.7±0.0 | 67.9±0.0 | 82.1±0.0 | 91.9±0.0 | 83.8±0.0 | 80.2±0.0 | 68.8±0.0 | 91.2±0.0 | 51.6±0.0 | 82.8±0.0 | 81.2±0.0 | 77.9±0.0 |
| BNAdapt | 80.6±0.1 | 81.8±0.1 | 69.9±0.1 | 91.5±0.1 | 80.2±0.1 | 88.0±0.2 | 92.7±0.0 | 86.2±0.1 | 88.4±0.2 | 83.6±0.2 | 91.2±0.1 | 89.5±0.1 | 84.0±0.2 | 90.5±0.1 | 80.4±0.1 |
| BNAdapt+DART (ours) | 80.5±0.3 | 81.7±0.2 | 68.1±0.7 | 91.6±0.1 | 80.4±0.1 | 88.1±0.2 | 92.9±0.1 | 86.4±0.0 | 88.6±0.1 | 83.7±0.1 | 91.3±0.1 | 89.7±0.1 | 84.1±0.1 | 90.6±0.1 | 80.5±0.2 |
| TENT | 81.8±0.2 | 83.2±0.6 | 74.0±0.4 | 91.2±0.3 | 81.3±0.6 | 88.5±0.5 | 92.3±0.3 | 87.4±0.3 | 88.4±0.1 | 87.7±0.3 | 91.4±0.2 | 90.7±0.3 | 83.9±0.1 | 90.1±0.3 | 83.3±0.2 |
| TENT+DART (ours) | 82.0±0.5 | 83.5±0.5 | 71.8±1.5 | 91.6±0.4 | 81.9±0.3 | 88.6±0.6 | 92.5±0.2 | 87.6±0.4 | 89.0±0.1 | 87.6±0.8 | 91.8±0.2 | 91.1±0.2 | 84.4±0.0 | 90.5±0.3 | 83.2±0.6 |
| PL | 82.3±0.3 | 83.1±0.3 | 74.1±0.5 | 91.5±0.2 | 81.2±0.7 | 88.8±0.1 | 92.3±0.3 | 87.5±0.2 | 89.0±0.1 | 87.8±0.2 | 91.3±0.4 | 90.7±0.4 | 83.7±0.1 | 90.6±0.3 | 83.3±0.2 |
| PL+DART (ours) | 82.3±0.3 | 83.7±0.4 | 72.5±0.8 | 91.8±0.3 | 81.8±0.5 | 88.9±0.2 | 92.3±0.3 | 87.6±0.3 | 89.1±0.2 | 88.0±0.2 | 91.7±0.2 | 91.0±0.5 | 84.3±0.3 | 90.6±0.2 | 83.6±0.4 |
| NOTE | 73.1±2.2 | 76.5±0.4 | 65.9±0.5 | 88.6±0.5 | 74.0±1.8 | 85.9±0.4 | 89.2±0.7 | 83.7±0.4 | 85.2±0.7 | 83.3±1.1 | 89.0±0.8 | 88.6±0.2 | 78.6±1.1 | 87.3±0.5 | 77.8±0.9 |
| NOTE+DART (ours) | 74.2±1.4 | 76.5±2.0 | 63.0±2.9 | 88.0±0.7 | 73.6±1.2 | 85.5±0.2 | 89.2±0.7 | 84.0±0.3 | 84.9±0.6 | 83.3±0.8 | 89.0±0.8 | 88.7±0.5 | 79.4±0.4 | 86.6±0.5 | 78.1±0.7 |
| LAME | 80.6±0.1 | 81.8±0.0 | 70.1±0.1 | 91.5±0.1 | 80.1±0.1 | 88.0±0.2 | 92.7±0.1 | 86.3±0.1 | 88.4±0.1 | 83.6±0.2 | 91.3±0.1 | 89.5±0.1 | 84.0±0.2 | 90.5±0.1 | 80.3±0.1 |
| LAME+DART (ours) | 80.6±0.1 | 81.9±0.1 | 69.2±0.5 | 91.5±0.1 | 80.2±0.1 | 87.9±0.2 | 92.7±0.1 | 86.4±0.1 | 88.5±0.1 | 83.7±0.1 | 91.3±0.2 | 89.6±0.1 | 84.1±0.1 | 90.6±0.1 | 80.4±0.2 |
| DELTA | 81.0±0.2 | 82.7±0.4 | 72.4±1.0 | 91.3±0.3 | 80.7±0.5 | 88.2±0.3 | 92.0±0.3 | 86.9±0.3 | 88.3±0.5 | 87.4±0.6 | 91.2±0.4 | 90.5±0.4 | 83.3±0.4 | 90.1±0.2 | 82.4±0.5 |
| DELTA+DART (ours) | 80.8±0.8 | 82.6±0.2 | 71.3±1.0 | 91.3±0.1 | 80.8±0.7 | 88.1±0.3 | 91.8±0.2 | 87.4±0.1 | 88.2±0.1 | 87.2±0.4 | 91.0±0.8 | 90.2±0.4 | 83.6±0.5 | 90.2±0.4 | 82.8±0.6 |
| ODS | 80.4±0.4 | 82.1±0.6 | 72.4±0.5 | 91.5±0.1 | 80.8±0.6 | 88.4±0.7 | 92.5±0.2 | 87.7±0.3 | 88.5±0.2 | 87.1±0.4 | 91.7±0.3 | 88.7±0.4 | 83.9±0.2 | 90.0±0.3 | 83.0±0.2 |
| ODS+DART (ours) | 80.3±0.3 | 82.0±0.7 | 71.5±0.8 | 91.5±0.0 | 80.9±0.5 | 88.4±0.7 | 92.5±0.2 | 87.8±0.3 | 88.4±0.2 | 87.1±0.4 | 91.8±0.3 | 88.7±0.4 | 84.0±0.2 | 90.0±0.4 | 83.0±0.3 |
| SAR | 82.5±0.5 | 83.8±0.3 | 74.6±0.4 | 91.7±0.3 | 81.6±0.2 | 89.0±0.1 | 92.4±0.3 | 87.9±0.4 | 89.0±0.2 | 88.1±0.2 | 91.5±0.2 | 90.8±0.4 | 84.6±0.2 | 90.7±0.1 | 83.8±0.2 |
| SAR+DART (ours) | 82.5±0.2 | 83.7±0.5 | 72.8±1.0 | 91.9±0.1 | 81.7±0.6 | 89.2±0.2 | 92.4±0.3 | 87.9±0.1 | 89.1±0.1 | 88.0±0.3 | 91.6±0.1 | 90.9±0.4 | 84.5±0.3 | 90.8±0.3 | 83.6±0.4 |

Table 20: Average accuracy (%) on CIFAR-10C-LT ($\rho = 10$)

| Method | gaussian_noise | shot_noise | impulse_noise | defocus_blur | glass_blur | motion_blur | zoom_blur | snow | frost | fog | brightness | contrast | elastic_transform | pixelate | jpeg_compression |
|---|---|---|---|---|---|---|---|---|---|---|---|---|---|---|---|
| NoAdapt | 45.8±0.4 | 50.1±0.2 | 29.2±0.5 | 89.8±0.3 | 66.1±0.2 | 84.9±0.0 | 92.8±0.2 | 82.8±0.2 | 77.0±0.2 | 66.8±0.4 | 91.6±0.2 | 54.7±0.4 | 82.8±0.2 | 79.7±0.1 | 75.7±0.3 |
| BNAdapt | 73.9±0.3 | 75.0±0.5 | 63.9±0.5 | 85.1±0.3 | 73.8±0.2 | 82.5±0.2 | 87.0±0.1 | 80.2±0.4 | 81.8±0.3 | 76.4±0.6 | 85.8±0.1 | 83.5±0.4 | 77.5±0.4 | 84.3±0.1 | 74.0±0.6 |
| BNAdapt+DART (ours) | 80.1±0.5 | 80.5±0.5 | 67.2±0.4 | 90.5±0.4 | 80.3±0.4 | 88.4±0.2 | 92.3±0.2 | 86.5±0.4 | 87.5±0.1 | 82.8±0.5 | 90.7±0.2 | 88.8±0.2 | 84.5±0.3 | 89.7±0.4 | 80.0±0.6 |
| TENT | 77.6±2.0 | 79.3±0.8 | 68.4±1.0 | 88.9±0.9 | 77.2±1.0 | 85.8±0.6 | 89.9±0.6 | 84.3±0.8 | 86.3±0.3 | 82.8±0.5 | 88.9±0.7 | 87.6±1.1 | 81.4±0.9 | 87.3±0.6 | 78.5±1.0 |
| TENT+DART (ours) | 82.3±1.1 | 83.0±1.1 | 71.3±1.3 | 91.9±0.2 | 82.2±0.2 | 89.4±0.7 | 93.3±0.1 | 88.7±0.5 | 89.2±0.5 | 87.2±1.0 | 92.1±0.5 | 90.7±0.3 | 85.6±0.2 | 91.2±0.4 | 83.1±0.8 |
| PL | 77.9±1.0 | 79.1±0.7 | 68.9±0.8 | 88.8±0.5 | 77.1±0.5 | 85.4±0.5 | 90.0±0.2 | 84.7±0.8 | 85.3±0.7 | 82.6±0.7 | 88.7±0.4 | 86.8±0.6 | 81.4±1.1 | 87.4±0.5 | 79.5±0.7 |
| PL+DART (ours) | 82.3±0.6 | 82.8±0.8 | 71.7±0.7 | 92.3±0.4 | 82.7±0.4 | 89.7±0.3 | 93.2±0.0 | 88.7±0.6 | 89.2±0.4 | 87.8±0.4 | 91.6±0.6 | 90.9±0.0 | 85.6±0.1 | 90.9±0.5 | 83.0±0.4 |
| NOTE | 72.8±1.5 | 74.2±1.7 | 58.8±2.8 | 88.2±1.4 | 72.9±4.5 | 87.5±0.7 | 90.8±0.4 | 84.1±0.8 | 85.6±1.6 | 81.4±1.6 | 89.7±1.4 | 87.9±1.1 | 78.9±0.7 | 86.6±1.2 | 76.8±1.5 |
| NOTE+DART (ours) | 74.2±0.3 | 75.4±1.2 | 62.1±1.5 | 88.6±0.9 | 76.1±1.5 | 86.6±0.3 | 90.0±0.3 | 85.5±1.0 | 85.9±0.7 | 82.6±0.8 | 89.5±1.4 | 87.9±1.4 | 79.6±1.6 | 87.0±0.8 | 77.1±0.6 |
| LAME | 75.3±0.5 | 76.6±0.5 | 65.3±0.4 | 86.6±0.2 | 75.3±0.3 | 83.9±0.2 | 88.4±0.1 | 81.8±0.4 | 83.3±0.2 | 78.0±0.5 | 87.1±0.2 | 84.8±0.6 | 78.9±0.3 | 85.6±0.2 | 75.3±0.6 |
| LAME+DART (ours) | 78.1±0.7 | 78.6±0.5 | 66.2±0.6 | 88.6±0.2 | 78.1±0.2 | 86.2±0.3 | 90.7±0.3 | 84.5±0.4 | 85.7±0.2 | 80.7±0.3 | 89.1±0.2 | 87.0±0.3 | 82.2±0.4 | 87.9±0.4 | 78.1±0.4 |
| DELTA | 74.6±2.7 | 78.8±1.1 | 66.5±2.0 | 89.6±1.2 | 77.0±0.9 | 85.4±1.5 | 89.8±1.2 | 83.5±1.5 | 86.1±2.1 | 82.4±1.2 | 90.1±0.6 | 86.9±1.6 | 80.8±1.2 | 87.4±1.4 | 78.4±2.0 |
| DELTA+DART (ours) | 81.1±1.4 | 81.3±2.1 | 67.5±4.9 | 91.0±0.5 | 81.5±0.5 | 88.7±0.3 | 92.8±0.7 | 88.1±0.7 | 89.1±0.6 | 86.6±0.6 | 91.8±0.6 | 90.9±0.1 | 84.2±0.7 | 90.7±0.3 | 82.8±1.3 |
| ODS | 75.4±2.3 | 78.0±0.6 | 65.7±2.8 | 91.0±0.7 | 78.2±1.7 | 87.0±0.5 | 92.6±0.4 | 86.4±1.2 | 87.3±0.6 | 84.0±0.6 | 91.7±0.6 | 85.0±1.0 | 83.9±0.7 | 89.1±0.8 | 80.3±1.5 |
| ODS+DART (ours) (ours)rs | 78.0±1.3 | 79.6±0.4 | 68.6±1.7 | 91.6±0.5 | 80.2±1.3 | 88.7±0.6 | 92.9±0.5 | 87.4±1.0 | 88.2±0.5 | 85.2±0.7 | 92.0±0.5 | 85.9±0.9 | 85.1±0.8 | 89.9±0.6 | 82.0±1.2 |
| SAR | 76.1±1.8 | 77.4±0.9 | 67.2±0.9 | 86.6±0.7 | 75.5±1.7 | 83.8±0.7 | 87.9±0.3 | 82.2±1.1 | 82.8±0.4 | 80.8±0.4 | 87.3±0.5 | 85.2±0.5 | 79.4±1.0 | 85.6±0.3 | 77.4±1.1 |
| SAR+DART (ours) | 82.2±0.9 | 82.7±0.5 | 71.5±0.5 | 91.9±0.5 | 81.7±0.3 | 89.2±0.3 | 92.8±0.2 | 88.4±0.4 | 88.8±0.6 | 86.7±0.6 | 91.3±0.4 | 90.2±0.3 | 85.3±0.3 | 90.6±0.4 | 82.6±0.8 |

Table 21: Average accuracy (%) on CIFAR-10C-LT ($\rho = 100$)

| Method | gaussian_noise | shot_noise | impulse_noise | defocus_blur | glass_blur | motion_blur | zoom_blur | snow | frost | fog | brightness | contrast | elastic_transform | pixelate | jpeg_compression |
|---|---|---|---|---|---|---|---|---|---|---|---|---|---|---|---|
| NoAdapt | 43.4±0.8 | 46.7±0.4 | 27.8±0.4 | 90.4±0.7 | 66.8±0.9 | 86.9±0.3 | 93.6±0.5 | 82.6±0.5 | 75.2±0.4 | 63.6±0.6 | 92.8±0.1 | 59.6±0.3 | 83.1±0.0 | 82.1±0.9 | 76.6±0.7 |
| BNAdapt | 62.5±1.1 | 63.5±0.1 | 54.1±0.4 | 73.1±0.3 | 62.0±0.6 | 69.7±0.5 | 74.7±0.2 | 66.8±0.4 | 69.8±0.3 | 64.6±0.5 | 73.4±0.4 | 72.0±0.4 | 65.2±0.4 | 73.2±0.2 | 64.0±0.6 |
| BNAdapt+DART (ours) | 80.8±0.8 | 80.7±1.2 | 70.3±0.8 | 91.2±0.6 | 79.8±1.3 | 89.4±0.7 | 92.5±0.7 | 86.7±0.2 | 88.0±0.4 | 82.8±0.4 | 91.5±0.6 | 89.2±0.3 | 84.4±0.5 | 90.1±0.6 | 79.3±1.2 |
| TENT | 65.6±1.7 | 68.1±0.8 | 57.5±1.8 | 76.2±1.3 | 66.2±1.6 | 72.9±1.1 | 77.6±0.7 | 69.7±0.4 | 73.0±0.7 | 68.4±0.9 | 76.0±0.7 | 74.8±0.2 | 68.7±0.8 | 75.3±0.8 | 66.1±1.2 |
| TENT+DART (ours) | 84.1±1.0 | 85.0±1.4 | 73.3±2.1 | 93.2±0.6 | 84.6±1.0 | 91.6±0.8 | 93.9±0.4 | 89.8±0.8 | 90.5±0.6 | 87.8±0.4 | 93.2±0.2 | 91.8±0.8 | 87.5±0.5 | 92.1±0.6 | 84.0±0.7 |
| PL | 63.4±2.0 | 65.8±1.3 | 56.1±0.8 | 73.1±1.3 | 63.2±0.9 | 70.6±0.7 | 75.3±1.7 | 68.7±0.8 | 70.0±0.9 | 66.9±0.9 | 73.8±1.3 | 73.0±0.6 | 65.5±1.0 | 72.7±0.3 | 64.5±0.6 |
| PL+DART (ours) | 81.5±0.9 | 81.9±1.2 | 70.8±1.9 | 91.9±0.8 | 82.0±1.2 | 90.5±0.9 | 92.9±0.3 | 88.0±0.6 | 89.0±0.6 | 84.9±0.5 | 91.9±0.7 | 90.3±0.4 | 86.0±0.3 | 90.7±0.7 | 81.9±0.8 |
| NOTE | 68.3±6.1 | 67.2±3.1 | 56.2±5.8 | 88.4±1.4 | 72.8±3.8 | 86.5±2.3 | 90.7±2.2 | 83.6±3.7 | 81.8±6.3 | 79.6±1.5 | 90.2±0.9 | 87.2±2.8 | 80.5±1.4 | 86.3±1.6 | 77.3±4.4 |
| NOTE+DART (ours) | 76.8±1.5 | 78.7±1.7 | 66.7±2.0 | 88.1±0.4 | 74.0±1.9 | 86.4±0.3 | 88.6±1.0 | 84.5±0.3 | 86.6±0.3 | 83.7±1.0 | 88.7±2.0 | 86.4±1.4 | 81.4±0.6 | 85.7±1.1 | 76.0±1.3 |
| LAME | 66.0±1.3 | 67.4±0.2 | 57.4±0.5 | 76.3±0.9 | 65.4±1.0 | 73.6±0.8 | 78.6±0.5 | 70.5±0.7 | 73.3±0.5 | 68.0±0.5 | 77.2±0.3 | 75.4±0.6 | 68.8±0.6 | 75.5±0.4 | 66.1±0.6 |
| LAME+DART (ours) | 76.3±0.7 | 77.2±1.0 | 66.6±1.3 | 87.5±0.2 | 75.9±0.9 | 85.0±0.9 | 88.7±0.4 | 82.9±0.3 | 83.9±0.4 | 78.6±0.5 | 87.6±0.6 | 84.7±1.0 | 80.9±0.6 | 85.9±0.5 | 75.8±1.0 |
| DELTA | 63.8±4.5 | 67.2±2.6 | 56.4±5.8 | 77.1±2.3 | 70.5±4.6 | 71.1±4.1 | 76.4±2.7 | 71.4±2.5 | 73.2±0.4 | 69.3±5.1 | 76.4±3.1 | 73.7±2.2 | 65.3±4.7 | 75.9±4.9 | 68.0±4.7 |
| DELTA+DART (ours) | 83.3±0.7 | 84.0±0.8 | 70.1±2.1 | 93.4±0.5 | 83.3±1.9 | 91.3±0.9 | 94.2±0.6 | 89.9±0.9 | 90.4±0.6 | 87.4±0.7 | 93.5±0.5 | 92.8±0.6 | 85.5±0.8 | 92.5±0.9 | 82.0±2.9 |
| ODS | 64.7±4.1 | 68.4±2.8 | 57.0±2.8 | 85.0±1.6 | 73.6±3.4 | 83.0±1.6 | 86.9±0.5 | 79.1±0.7 | 79.1±1.2 | 71.4±2.1 | 86.9±1.0 | 77.8±1.9 | 77.5±0.9 | 82.4±0.7 | 74.8±1.2 |
| ODS+DART (ours) | 77.5±1.9 | 79.4±2.0 | 70.1±1.7 | 91.0±1.2 | 82.8±2.0 | 89.1±1.3 | 92.9±0.5 | 87.5±0.9 | 87.3±1.2 | 81.0±2.0 | 92.1±0.8 | 84.6±1.8 | 86.3±1.1 | 89.0±0.7 | 83.2±0.8 |
| SAR | 64.7±1.7 | 64.6±1.4 | 55.4±1.1 | 74.0±0.5 | 62.9±1.1 | 70.6±0.5 | 75.3±1.4 | 68.3±0.6 | 70.4±0.7 | 66.4±0.6 | 74.5±0.4 | 73.1±0.4 | 66.2±0.3 | 72.4±0.5 | 63.9±1.4 |
| SAR+DART (ours) | 82.9±1.0 | 83.2±1.4 | 72.9±1.8 | 92.1±0.4 | 82.5±1.0 | 91.1±0.9 | 93.3±0.8 | 88.6±0.4 | 89.7±0.4 | 85.7±0.6 | 92.4±0.5 | 90.6±0.6 | 86.3±0.6 | 91.1±0.6 | 82.5±1.2 |

Table 22: Average accuracy (%) on PACS. *X2Y* refers to a scenario where a pre-trained model, trained in domain *X*, is tested in domain *Y*. For example, a2c refers to a scenario where a pre-trained model, trained in the art domain, is tested in the cartoon domain.

| Method | a2c | a2p | a2s | c2a | c2p | c2s | p2a | p2c | p2s | s2a | s2c | s2p | avg |
|---|---|---|---|---|---|---|---|---|---|---|---|---|---|
| NoAdapt | 66.0±0.0 | 97.8±0.0 | 57.3±0.0 | 75.6±0.0 | 90.2±0.0 | 72.2±0.0 | 73.2±0.0 | 39.7±0.0 | 43.9±0.0 | 23.5±0.0 | 50.3±0.0 | 38.0±0.0 | 60.7±0.0 |
| BNAdapt | 75.8±0.3 | 97.0±0.2 | 74.0±0.2 | 82.5±0.3 | 95.1±0.4 | 73.5±0.7 | 77.8±0.2 | 65.0±0.4 | 46.9±0.2 | 59.4±0.5 | 69.0±0.4 | 57.8±0.2 | 72.5±0.0 |
| BNAdapt+DART (ours) | 76.0±0.3 | 96.8±0.1 | 72.1±0.5 | 84.4±0.6 | 95.4±0.2 | 74.7±0.3 | 76.1±0.3 | 62.0±0.4 | 55.0±0.7 | 74.1±0.2 | 76.7±0.2 | 74.0±0.6 | 76.4±0.1 |
| TENT | 78.6±0.9 | 97.8±0.5 | 77.1±1.9 | 88.0±0.3 | 97.1±0.5 | 79.1±1.2 | 81.7±0.6 | 75.5±1.0 | 59.2±2.5 | 60.1±1.1 | 71.0±1.0 | 56.8±1.6 | 76.8±0.6 |
| TENT+DART (ours) | 80.8±0.8 | 97.7±0.4 | 79.2±0.7 | 88.9±0.2 | 97.4±0.3 | 79.2±1.2 | 79.6±1.0 | 75.4±1.7 | 69.8±1.5 | 81.3±0.6 | 77.6±0.7 | 71.6±0.5 | 81.6±0.4 |
| PL | 77.6±0.4 | 96.9±0.6 | 70.0±0.3 | 83.8±1.2 | 95.4±0.6 | 68.9±1.5 | 78.0±0.5 | 66.3±0.8 | 46.9±0.2 | 59.9±0.4 | 68.6±1.4 | 56.8±0.2 | 72.4±0.2 |
| PL+DART (ours) | 78.0±0.8 | 96.8±0.2 | 71.7±0.8 | 85.2±0.7 | 96.0±0.4 | 72.9±1.3 | 76.5±0.3 | 63.5±1.1 | 55.0±0.7 | 74.1±0.2 | 76.9±0.6 | 74.0±2.6 | 76.7±0.4 |
| NOTE | 79.2±0.8 | 97.2±0.4 | 74.8±4.6 | 86.1±1.0 | 96.6±0.4 | 77.3±4.8 | 77.4±1.0 | 72.5±2.8 | 68.5±0.9 | 73.8±2.8 | 76.1±2.2 | 69.2±0.6 | 79.1±0.8 |
| NOTE+DART (ours) | 80.7±1.0 | 97.1±0.8 | 77.9±1.9 | 85.7±0.8 | 96.8±0.6 | 79.0±1.0 | 75.7±1.6 | 70.8±1.9 | 70.6±2.1 | 78.5±1.3 | 76.2±0.3 | 69.3±0.3 | 79.9±0.3 |
| LAME | 78.6±0.4 | 98.0±0.1 | 72.8±0.6 | 84.1±0.5 | 96.9±0.2 | 72.5±0.9 | 80.4±0.8 | 61.0±0.6 | 29.4±1.2 | 59.9±2.1 | 71.6±0.8 | 65.3±1.1 | 72.5±0.3 |
| LAME+DART (ours) | 78.7±0.7 | 97.8±0.2 | 70.7±0.9 | 84.8±0.5 | 96.5±0.7 | 71.0±1.1 | 78.5±1.1 | 59.8±0.4 | 41.3±1.9 | 78.6±0.7 | 76.0±0.7 | 78.8±3.4 | 76.0±0.2 |
| DELTA | 79.6±1.0 | 98.5±0.2 | 75.4±2.2 | 89.3±0.2 | 97.8±0.2 | 80.9±1.2 | 82.3±0.5 | 77.4±1.4 | 62.5±0.9 | 60.6±1.8 | 72.0±1.1 | 62.2±5.0 | 78.2±0.6 |
| DELTA+DART (ours) | 81.7±1.1 | 98.3±0.1 | 77.0±1.3 | 89.8±0.3 | 97.7±0.2 | 80.5±1.6 | 79.5±1.2 | 76.7±0.9 | 66.6±1.8 | 83.6±0.4 | 76.7±2.1 | 71.8±1.1 | 81.7±0.4 |
| ODS | 79.9±1.3 | 98.4±0.3 | 78.7±1.2 | 88.8±0.4 | 98.1±0.3 | 78.4±0.8 | 82.8±0.7 | 74.9±0.8 | 50.9±2.6 | 59.2±2.1 | 73.4±0.8 | 65.0±1.0 | 77.4±0.5 |
| ODS+DART (ours) | 80.1±1.3 | 98.4±0.2 | 78.6±0.7 | 88.8±0.2 | 97.8±0.3 | 77.8±0.8 | 81.4±0.5 | 74.0±0.3 | 58.6±2.0 | 73.3±1.6 | 74.9±0.3 | 73.2±1.9 | 79.7±0.4 |
| SAR | 79.6±0.8 | 97.3±0.4 | 70.3±1.6 | 85.4±0.4 | 95.8±0.6 | 74.9±2.1 | 78.0±0.7 | 67.8±0.3 | 46.4±0.8 | 62.6±0.5 | 72.5±0.5 | 59.7±0.6 | 74.2±0.4 |
| SAR+DART (ours) | 80.3±0.6 | 97.1±0.1 | 70.9±1.9 | 86.7±0.4 | 96.2±0.2 | 77.2±1.1 | 76.6±0.7 | 66.6±0.9 | 55.3±1.3 | 76.5±0.4 | 80.3±0.4 | 78.6±0.3 | 78.5±0.2 |

Table 23: Average accuracy (%) on OfficeHome. *X2Y* refers to a scenario where a pre-trained model, trained in domain *X*, is tested in domain *Y*. For example, a2c refers to a scenario where a pre-trained model, trained in the art domain, is tested in the clipart domain.

| Method | a2c | a2p | a2r | c2a | c2p | c2r | p2a | p2c | p2r | r2a | r2c | r2p | avg |
|---|---|---|---|---|---|---|---|---|---|---|---|---|---|
| NoAdapt | 47.9±0.0 | 65.8±0.0 | 73.2±0.0 | 52.4±0.0 | 63.2±0.0 | 64.2±0.0 | 49.8±0.0 | 46.3±0.0 | 71.8±0.0 | 65.2±0.0 | 51.7±0.0 | 77.5±0.0 | 60.8±0.0 |
| BNAdapt | 48.2±0.1 | 62.4±0.2 | 71.4±0.2 | 51.2±0.3 | 62.6±0.2 | 62.9±0.1 | 51.3±0.1 | 48.3±0.4 | 72.3±0.4 | 64.5±0.4 | 53.6±0.5 | 76.3±0.2 | 60.4±0.1 |
| BNAdapt+DART (ours) | 47.8±0.3 | 63.5±0.1 | 72.4±0.2 | 52.7±0.4 | 63.2±0.2 | 63.7±0.2 | 51.9±0.3 | 48.9±0.2 | 72.8±0.4 | 64.5±0.4 | 53.8±0.4 | 76.8±0.2 | 61.0±0.1 |
| TENT | 48.1±0.4 | 62.0±0.8 | 69.5±0.6 | 54.6±0.4 | 63.7±0.7 | 64.3±0.8 | 52.2±0.3 | 47.7±0.7 | 72.3±0.5 | 65.5±0.7 | 54.0±0.6 | 76.6±0.4 | 60.9±0.2 |
| TENT+DART (ours) | 49.8±0.4 | 65.3±0.5 | 71.4±0.4 | 55.8±0.5 | 65.0±0.6 | 65.7±0.4 | 53.3±0.4 | 48.4±0.4 | 72.8±0.3 | 66.0±0.6 | 54.8±0.7 | 77.2±0.4 | 62.1±0.2 |
| PL | 46.3±0.8 | 60.3±0.1 | 68.7±0.6 | 51.5±0.4 | 61.7±0.5 | 62.8±0.2 | 51.1±0.1 | 46.1±1.2 | 71.2±0.4 | 64.7±0.5 | 51.7±0.4 | 75.0±0.5 | 59.2±0.2 |
| PL+DART (ours) | 46.8±0.4 | 62.5±0.7 | 71.0±0.3 | 51.9±1.1 | 62.1±0.3 | 63.7±0.6 | 51.8±0.5 | 46.8±1.7 | 72.0±0.6 | 64.7±0.5 | 52.1±1.5 | 75.1±0.4 | 60.1±0.4 |
| NOTE | 41.1±0.9 | 51.4±0.6 | 61.4±0.3 | 47.7±1.3 | 56.1±1.0 | 57.2±0.8 | 45.5±1.1 | 43.2±1.3 | 62.7±0.5 | 60.5±0.5 | 48.3±1.7 | 69.7±1.1 | 53.7±0.3 |
| NOTE+DART (ours) | 40.4±1.5 | 50.0±1.0 | 62.1±0.7 | 48.8±0.4 | 55.8±1.4 | 57.4±1.1 | 44.6±1.4 | 42.3±2.4 | 62.7±0.6 | 60.0±1.1 | 48.6±1.1 | 69.5±1.0 | 53.5±0.1 |
| LAME | 46.5±0.5 | 60.2±0.4 | 68.3±0.4 | 51.7±0.1 | 62.2±0.5 | 62.2±0.1 | 51.9±0.3 | 48.4±0.4 | 71.9±0.4 | 64.5±0.2 | 53.5±0.3 | 75.9±0.1 | 59.8±0.1 |
| LAME+DART (ours) | 45.7±0.2 | 61.3±0.2 | 68.4±0.4 | 53.2±0.6 | 62.8±0.2 | 62.9±0.1 | 52.7±0.4 | 49.0±0.4 | 72.3±0.4 | 64.7±0.1 | 53.8±0.4 | 76.0±0.3 | 60.2±0.2 |
| DELTA | 50.6±0.6 | 65.6±0.4 | 72.3±0.4 | 56.3±0.9 | 67.1±0.7 | 66.8±0.1 | 52.8±0.7 | 49.9±0.4 | 74.7±0.2 | 66.6±1.0 | 56.7±0.5 | 78.1±0.3 | 63.1±0.1 |
| DELTA+DART (ours) | 51.5±0.4 | 67.6±0.2 | 73.6±0.3 | 57.4±0.5 | 67.4±0.6 | 67.6±0.6 | 54.2±0.4 | 50.1±0.6 | 75.1±0.1 | 66.9±0.7 | 56.9±0.7 | 78.3±0.1 | 63.9±0.1 |
| ODS | 49.4±0.5 | 63.6±0.7 | 70.2±0.4 | 54.9±0.3 | 65.1±0.7 | 65.2±0.3 | 52.9±0.1 | 49.5±0.6 | 73.4±0.2 | 66.5±0.6 | 55.9±0.5 | 78.1±0.3 | 62.1±0.2 |
| ODS+DART (ours) | 49.9±0.4 | 64.8±0.9 | 71.0±0.4 | 55.4±0.4 | 65.6±0.6 | 65.7±0.2 | 53.4±0.2 | 49.8±0.6 | 73.7±0.2 | 66.7±0.5 | 56.0±0.5 | 78.3±0.3 | 62.5±0.2 |
| SAR | 49.8±0.3 | 62.8±0.5 | 71.2±0.2 | 52.5±0.6 | 64.1±0.3 | 64.9±0.1 | 51.2±0.5 | 49.1±0.3 | 72.2±0.4 | 65.0±0.3 | 54.1±0.5 | 76.6±0.2 | 61.1±0.0 |
| SAR+DART (ours) | 49.1±0.4 | 64.3±0.4 | 72.4±0.1 | 53.3±0.3 | 64.5±0.3 | 65.8±0.3 | 52.0±0.9 | 49.3±0.3 | 72.7±0.5 | 64.9±0.3 | 54.7±0.4 | 77.1±0.2 | 61.7±0.2 |

Table 24: Average accuracy (%) on CIFAR-100C-imb (IR1)

| Method | gaussian_noise | shot_noise | impulse_noise | defocus_blur | glass_blur | motion_blur | zoom_blur | snow | frost | fog | brightness | contrast | elastic_transform | pixelate | jpeg_compression |
|---|---|---|---|---|---|---|---|---|---|---|---|---|---|---|---|
| NoAdapt | 16.3±0.1 | 17.4±0.2 | 7.6±0.2 | 67.3±0.7 | 27.8±0.5 | 55.9±0.3 | 70.4±0.4 | 50.5±0.4 | 42.4±0.7 | 33.6±0.4 | 63.7±0.2 | 18.8±0.5 | 52.6±0.5 | 50.9±0.2 | 47.8±0.6 |
| BNAdapt | 53.4±0.6 | 53.9±0.5 | 43.6±0.4 | 69.3±0.8 | 55.0±0.7 | 64.4±0.5 | 71.9±0.5 | 58.0±0.6 | 62.1±0.6 | 53.1±0.2 | 67.8±0.5 | 63.4±0.7 | 59.5±0.8 | 67.3±0.7 | 53.9±0.2 |
| BNAdapt+DART-split (ours) | 53.4±0.6 | 53.9±0.5 | 43.0±0.4 | 69.3±0.8 | 55.0±0.7 | 64.4±0.5 | 71.9±0.5 | 58.0±0.6 | 62.1±0.6 | 53.1±0.2 | 67.8±0.5 | 63.4±0.7 | 59.5±0.8 | 67.3±0.7 | 53.9±0.2 |
| TENT | 55.1±0.4 | 56.1±0.4 | 47.2±0.9 | 69.4±0.7 | 55.9±0.9 | 65.4±0.6 | 71.2±0.3 | 61.9±1.1 | 63.3±0.5 | 61.7±0.6 | 68.8±0.3 | 68.2±0.9 | 59.9±0.9 | 68.0±0.5 | 56.9±0.7 |
| TENT+DART-split (ours) | 55.1±0.4 | 56.1±0.4 | 46.6±1.0 | 69.4±0.7 | 55.9±0.9 | 65.4±0.6 | 71.2±0.3 | 61.9±1.1 | 63.3±0.5 | 61.7±0.6 | 68.8±0.3 | 68.2±0.9 | 59.9±0.9 | 68.0±0.5 | 56.9±0.7 |
| SAR | 53.9±1.3 | 54.8±0.7 | 46.0±1.5 | 69.1±0.4 | 54.9±1.2 | 65.3±0.5 | 70.7±0.6 | 60.3±0.4 | 62.4±0.7 | 60.2±0.4 | 67.9±0.5 | 66.9±0.3 | 58.6±0.7 | 67.2±0.8 | 56.1±0.9 |
| SAR+DART-split (ours) | 53.9±1.3 | 54.8±0.7 | 45.1±1.3 | 69.1±0.4 | 54.9±1.2 | 65.3±0.5 | 70.7±0.6 | 60.3±0.4 | 62.4±0.7 | 60.2±0.4 | 67.9±0.5 | 66.9±0.3 | 58.6±0.7 | 67.2±0.8 | 56.1±0.9 |
| ODS | 55.2±0.3 | 56.3±0.2 | 47.0±0.3 | 70.6±0.7 | 56.6±0.8 | 66.6±0.6 | 72.6±0.3 | 62.2±0.8 | 64.1±0.7 | 61.1±0.4 | 69.8±0.3 | 66.6±1.1 | 61.1±0.9 | 68.8±0.5 | 57.3±0.3 |
| ODS+DART-split (ours) | 55.2±0.3 | 56.3±0.2 | 46.8±0.2 | 70.6±0.7 | 56.6±0.8 | 66.6±0.6 | 72.6±0.3 | 62.2±0.8 | 64.1±0.7 | 61.1±0.4 | 69.8±0.3 | 66.6±1.1 | 61.1±0.9 | 68.8±0.5 | 57.3±0.3 |

Table 25: Average accuracy (%) on CIFAR-100C-imb (IR200)

| Method | gaussian_noise | shot_noise | impulse_noise | defocus_blur | glass_blur | motion_blur | zoom_blur | snow | frost | fog | brightness | contrast | elastic_transform | pixelate | jpeg_compression |
|---|---|---|---|---|---|---|---|---|---|---|---|---|---|---|---|
| NoAdapt | 16.1±0.4 | 17.2±0.4 | 7.5±0.1 | 67.3±0.7 | 27.8±0.6 | 55.8±0.4 | 70.3±0.5 | 50.3±0.3 | 42.4±0.4 | 33.8±0.4 | 63.4±0.3 | 18.7±0.3 | 52.5±0.4 | 50.8±0.3 | 47.6±0.6 |
| BNAdapt | 26.6±0.4 | 26.8±0.5 | 22.7±0.5 | 33.7±0.6 | 26.4±0.5 | 31.2±0.4 | 34.6±0.8 | 28.0±0.3 | 30.1±0.5 | 26.3±0.5 | 32.8±0.0 | 30.0±0.6 | 28.5±0.6 | 32.5±0.5 | 26.3±0.3 |
| BNAdapt+DART-split (ours) | 45.9±0.7 | 46.1±0.4 | 41.2±1.6 | 44.7±1.4 | 45.5±1.2 | 46.6±1.1 | 46.0±0.5 | 45.6±1.1 | 47.8±1.2 | 45.7±1.2 | 44.4±1.8 | 47.2±0.7 | 47.5±0.9 | 46.0±1.3 | 46.2±0.6 |
| TENT | 23.3±0.6 | 23.9±0.7 | 20.2±0.3 | 29.1±0.7 | 23.2±0.6 | 26.8±0.5 | 29.7±0.7 | 24.5±0.7 | 25.7±0.4 | 24.9±0.1 | 28.2±0.8 | 25.5±0.8 | 24.6±0.4 | 28.4±0.4 | 23.7±0.7 |
| TENT+DART-split (ours) | 44.6±1.2 | 44.2±0.3 | 39.7±1.5 | 42.8±1.9 | 43.3±1.0 | 44.3±0.7 | 43.6±0.8 | 44.1±1.4 | 46.3±1.4 | 44.2±0.9 | 41.7±2.1 | 45.3±1.1 | 45.8±1.0 | 44.0±1.6 | 43.9±1.0 |
| SAR | 24.1±0.7 | 24.7±1.2 | 20.8±0.6 | 30.5±0.8 | 23.8±0.8 | 28.6±0.8 | 31.3±0.5 | 25.9±1.2 | 27.6±0.3 | 25.5±0.8 | 30.0±0.8 | 28.1±0.8 | 25.7±0.8 | 29.6±0.4 | 24.6±0.4 |
| SAR+DART-split (ours) | 45.3±0.9 | 45.9±0.6 | 40.5±2.0 | 42.9±2.0 | 44.5±0.7 | 44.9±0.9 | 44.0±0.6 | 44.9±1.3 | 46.8±1.6 | 45.7±1.6 | 42.9±1.9 | 45.8±1.0 | 46.2±0.4 | 44.6±1.5 | 45.8±0.5 |
| ODS | 43.3±1.7 | 44.2±0.9 | 36.0±0.7 | 58.0±0.6 | 44.9±0.6 | 52.7±0.8 | 60.1±1.1 | 48.7±0.4 | 51.0±0.3 | 48.5±0.7 | 57.3±0.5 | 47.5±0.7 | 48.8±1.1 | 56.0±0.8 | 46.6±0.5 |
| ODS+DART-split (ours) | 45.3±2.8 | 46.3±1.0 | 37.3±0.5 | 61.0±0.9 | 47.2±1.0 | 56.2±1.1 | 62.9±1.5 | 52.4±0.9 | 54.1±0.6 | 52.8±1.0 | 60.6±1.2 | 48.8±0.8 | 52.0±1.7 | 59.0±1.3 | 51.3±0.4 |

Table 26: Average accuracy (%) on CIFAR-100C-imb (IR500)

| Method | gaussian_noise | shot_noise | impulse_noise | defocus_blur | glass_blur | motion_blur | zoom_blur | snow | frost | fog | brightness | contrast | elastic_transform | pixelate | jpeg_compression |
|---|---|---|---|---|---|---|---|---|---|---|---|---|---|---|---|
| NoAdapt | 16.1±0.3 | 17.2±0.3 | 7.5±0.2 | 67.2±0.6 | 27.7±0.5 | 55.8±0.4 | 70.1±0.6 | 50.2±0.6 | 42.3±0.6 | 33.5±0.5 | 63.4±0.4 | 18.6±0.4 | 52.5±0.4 | 50.8±0.3 | 47.8±0.6 |
| BNAdapt | 17.5±0.5 | 17.6±0.4 | 14.9±0.3 | 21.4±0.4 | 17.0±0.5 | 20.1±0.5 | 22.0±0.3 | 18.4±0.4 | 19.3±0.5 | 17.5±0.3 | 20.8±0.5 | 19.2±0.5 | 18.1±0.6 | 20.7±0.6 | 17.0±0.4 |
| BNAdapt+DART-split (ours) | 47.6±1.1 | 47.7±0.5 | 42.4±1.3 | 56.1±1.2 | 45.7±1.3 | 53.5±0.6 | 55.1±0.7 | 50.1±0.4 | 52.2±1.2 | 47.7±1.8 | 53.6±1.1 | 50.8±0.6 | 48.3±1.0 | 56.1±0.7 | 45.7±0.6 |
| TENT | 14.2±0.5 | 14.7±0.6 | 12.6±0.1 | 17.8±0.4 | 13.8±0.3 | 16.2±0.8 | 18.1±0.7 | 14.9±1.2 | 15.9±0.7 | 15.4±0.7 | 17.0±0.7 | 14.6±0.9 | 15.1±0.3 | 17.1±0.6 | 14.3±0.2 |
| TENT+DART-split (ours) | 46.2±1.6 | 46.1±0.8 | 41.7±1.0 | 54.4±1.7 | 44.1±1.6 | 51.9±0.7 | 53.5±0.3 | 48.8±1.4 | 50.3±1.5 | 46.4±2.3 | 52.5±1.6 | 49.2±1.5 | 46.6±1.1 | 54.2±0.4 | 44.1±1.4 |
| SAR | 15.3±0.9 | 15.5±0.5 | 12.7±0.3 | 19.1±0.5 | 14.9±0.5 | 17.4±0.4 | 19.2±0.1 | 16.3±0.3 | 16.9±0.8 | 16.1±0.6 | 18.4±0.6 | 16.6±0.5 | 15.9±0.6 | 18.1±0.6 | 14.9±0.4 |
| SAR+DART-split (ours) | 47.3±1.5 | 47.6±0.4 | 42.3±1.1 | 56.1±1.6 | 45.5±1.1 | 53.3±0.6 | 55.1±0.7 | 49.8±0.7 | 51.9±0.8 | 47.5±2.0 | 53.6±1.3 | 50.8±0.6 | 47.8±0.8 | 56.0±0.7 | 45.4±0.7 |
| ODS | 38.3±0.6 | 39.2±0.6 | 31.7±0.5 | 52.7±0.9 | 38.6±0.9 | 46.9±0.9 | 54.8±1.4 | 42.1±1.5 | 45.6±1.0 | 42.8±2.0 | 51.6±1.3 | 38.8±2.2 | 42.2±0.3 | 49.5±0.9 | 40.7±0.8 |
| ODS+DART-split (ours) | 43.9±1.0 | 44.6±1.0 | 35.8±0.8 | 61.5±1.1 | 44.7±0.5 | 54.7±1.1 | 64.1±1.8 | 49.9±2.4 | 53.5±1.8 | 49.8±2.7 | 61.1±1.4 | 42.7±3.0 | 49.5±1.1 | 57.4±1.3 | 47.8±1.0 |

Table 27: Average accuracy (%) on CIFAR-100C-imb (IR50000)

| Method | gaussian_noise | shot_noise | impulse_noise | defocus_blur | glass_blur | motion_blur | zoom_blur | snow | frost | fog | brightness | contrast | elastic_transform | pixelate | jpeg_compression |
|---|---|---|---|---|---|---|---|---|---|---|---|---|---|---|---|
| NoAdapt | 16.1±0.5 | 17.3±0.2 | 7.4±0.2 | 67.3±0.6 | 27.8±0.5 | 55.9±0.2 | 70.3±0.4 | 50.3±0.5 | 42.5±0.4 | 33.4±0.3 | 63.7±0.3 | 18.7±0.3 | 52.5±0.3 | 51.0±0.3 | 47.9±0.4 |
| BNAdapt | 9.1±0.2 | 8.9±0.1 | 8.0±0.4 | 10.2±0.2 | 8.5±0.4 | 9.7±0.2 | 10.3±0.2 | 9.2±0.1 | 9.5±0.3 | 9.0±0.2 | 9.9±0.2 | 9.6±0.2 | 8.8±0.3 | 9.9±0.1 | 8.5±0.2 |
| BNAdapt+DART-split (ours) | 46.8±0.9 | 46.4±0.9 | 40.8±1.3 | 55.4±1.2 | 44.0±1.3 | 52.3±1.5 | 54.5±0.7 | 48.8±0.7 | 51.7±2.0 | 46.6±2.0 | 53.0±1.7 | 49.1±1.5 | 47.5±0.5 | 54.9±1.6 | 45.0±1.4 |
| TENT | 6.7±0.3 | 7.0±0.2 | 6.1±0.4 | 7.9±0.2 | 6.3±0.3 | 7.3±0.2 | 8.0±0.4 | 7.0±0.1 | 7.1±0.2 | 7.1±0.2 | 7.6±0.4 | 6.9±0.5 | 6.8±0.1 | 7.6±0.3 | 6.6±0.2 |
| TENT+DART-split (ours) | 45.9±0.6 | 45.3±1.1 | 40.5±1.5 | 54.5±0.9 | 42.2±1.6 | 50.4±1.8 | 53.2±0.6 | 47.3±0.4 | 50.3±2.9 | 45.5±1.7 | 51.6±1.7 | 47.0±1.7 | 45.8±0.6 | 53.3±1.6 | 43.6±0.5 |
| SAR | 7.7±0.4 | 7.7±0.5 | 6.6±0.2 | 8.6±0.3 | 7.0±0.2 | 7.9±0.3 | 8.8±0.4 | 7.9±0.3 | 8.2±0.1 | 7.6±0.3 | 8.3±0.2 | 7.8±0.3 | 7.4±0.3 | 8.5±0.4 | 7.4±0.3 |
| SAR+DART-split (ours) | 46.8±0.9 | 46.4±0.9 | 41.0±1.3 | 55.4±1.2 | 44.0±1.3 | 52.2±1.3 | 54.5±0.6 | 48.8±0.7 | 51.7±2.0 | 46.6±2.0 | 53.0±1.8 | 49.1±1.5 | 47.3±0.6 | 54.8±1.7 | 44.9±1.3 |
| ODS | 33.6±0.9 | 33.6±0.9 | 26.7±1.2 | 47.3±0.9 | 33.1±1.5 | 41.5±0.9 | 50.8±1.9 | 37.4±0.6 | 39.7±1.2 | 36.2±2.6 | 46.5±1.2 | 33.0±1.7 | 38.7±1.0 | 45.6±1.3 | 36.6±0.6 |
| ODS+DART-split (ours) | 42.6±1.3 | 43.3±1.4 | 33.6±1.5 | 60.4±1.5 | 42.5±1.6 | 53.4±0.7 | 63.3±1.6 | 49.1±1.0 | 51.2±1.4 | 46.8±4.1 | 58.7±1.6 | 41.2±2.8 | 49.5±1.4 | 57.4±1.3 | 46.7±0.3 |

Table 28: Average accuracy (%) on ImageNet-C-imb (IR1)

| Method | gaussian_noise | shot_noise | impulse_noise | defocus_blur | glass_blur | motion_blur | zoom_blur | snow | frost | fog | brightness | contrast | elastic_transform | pixelate | jpeg_compression |
|---|---|---|---|---|---|---|---|---|---|---|---|---|---|---|---|
| NoAdapt | 2.2±0.0 | 2.9±0.0 | 1.8±0.0 | 18.0±0.1 | 9.8±0.1 | 14.8±0.1 | 22.5±0.1 | 17.0±0.1 | 23.4±0.1 | 24.5±0.0 | 59.0±0.2 | 5.4±0.1 | 17.0±0.1 | 20.6±0.1 | 31.7±0.1 |
| BNAdapt | 15.2±0.1 | 15.9±0.1 | 15.8±0.1 | 15.1±0.0 | 15.3±0.1 | 26.3±0.2 | 38.8±0.1 | 34.4±0.2 | 33.2±0.1 | 48.0±0.1 | 65.2±0.1 | 16.9±0.1 | 44.1±0.2 | 48.9±0.2 | 39.7±0.2 |
| BNAdapt+DART-split (ours) | 15.2±0.1 | 15.8±0.1 | 15.8±0.1 | 15.1±0.0 | 15.3±0.1 | 26.3±0.2 | 38.8±0.1 | 34.4±0.2 | 33.2±0.1 | 48.0±0.1 | 65.2±0.1 | 16.9±0.1 | 44.1±0.2 | 48.9±0.2 | 39.7±0.2 |
| TENT | 28.9±0.7 | 32.6±0.6 | 31.9±0.6 | 27.8±1.1 | 26.9±0.1 | 45.4±0.2 | 51.1±0.2 | 50.1±0.3 | 38.8±0.5 | 59.1±0.1 | 67.6±0.1 | 14.9±1.7 | 56.8±0.2 | 60.1±0.1 | 54.2±0.1 |
| TENT+DART-split (ours) | 28.9±0.7 | 32.2±0.7 | 31.9±0.6 | 27.8±1.1 | 26.9±0.1 | 45.4±0.2 | 51.1±0.2 | 50.1±0.3 | 38.8±0.5 | 59.1±0.1 | 67.6±0.1 | 14.9±1.7 | 56.8±0.2 | 60.1±0.1 | 54.2±0.1 |
| SAR | 26.5±0.1 | 25.8±0.5 | 27.2±0.1 | 25.4±0.2 | 24.7±0.2 | 37.2±0.1 | 46.6±0.1 | 43.5±0.2 | 39.8±0.1 | 55.3±0.1 | 66.7±0.1 | 32.6±0.2 | 51.4±0.2 | 56.1±0.1 | 49.4±0.1 |
| SAR+DART-split (ours) | 26.5±0.1 | 25.7±0.4 | 27.2±0.1 | 25.4±0.2 | 24.7±0.2 | 37.2±0.1 | 46.6±0.1 | 43.5±0.2 | 39.8±0.1 | 55.3±0.1 | 66.7±0.1 | 32.6±0.2 | 51.4±0.2 | 56.1±0.1 | 49.4±0.1 |
| ODS | 29.7±0.5 | 32.7±0.5 | 32.2±0.6 | 28.7±0.7 | 28.0±0.2 | 44.4±0.2 | 51.4±0.1 | 49.4±0.3 | 40.3±0.5 | 59.1±0.1 | 68.3±0.1 | 17.3±1.6 | 56.7±0.2 | 60.0±0.1 | 54.0±0.2 |
| ODS+DART-split (ours) | 29.7±0.5 | 32.6±0.5 | 32.2±0.6 | 28.7±0.7 | 28.0±0.2 | 44.4±0.2 | 51.4±0.1 | 49.4±0.3 | 40.3±0.5 | 59.1±0.1 | 68.3±0.1 | 17.3±1.6 | 56.7±0.2 | 60.0±0.1 | 54.0±0.2 |

Table 29: Average accuracy (%) on ImageNet-C-imb (IR1000)

| Method | gaussian_noise | shot_noise | impulse_noise | defocus_blur | glass_blur | motion_blur | zoom_blur | snow | frost | fog | brightness | contrast | elastic_transform | pixelate | jpeg_compression |
|---|---|---|---|---|---|---|---|---|---|---|---|---|---|---|---|
| NoAdapt | 2.2±0.0 | 3.0±0.0 | 1.8±0.0 | 17.9±0.1 | 9.8±0.1 | 14.8±0.1 | 22.6±0.2 | 17.0±0.1 | 23.3±0.1 | 24.5±0.1 | 59.0±0.2 | 5.4±0.0 | 17.0±0.2 | 20.6±0.2 | 31.8±0.2 |
| BNAdapt | 9.8±0.0 | 10.2±0.1 | 10.1±0.1 | 9.2±0.1 | 9.4±0.1 | 16.2±0.1 | 23.8±0.2 | 21.9±0.1 | 21.2±0.1 | 30.1±0.1 | 41.9±0.2 | 10.5±0.1 | 27.1±0.2 | 30.2±0.1 | 24.9±0.2 |
| BNAdapt+DART-split (ours) | 9.8±0.0 | 9.9±0.1 | 10.0±0.1 | 9.2±0.1 | 9.4±0.1 | 16.2±0.1 | 23.8±0.2 | 21.9±0.1 | 21.4±0.1 | 30.5±0.2 | 43.7±0.7 | 10.5±0.1 | 27.3±0.2 | 30.2±0.1 | 26.6±0.6 |
| TENT | 8.6±0.2 | 8.9±0.8 | 10.3±0.6 | 8.0±0.4 | 6.8±0.4 | 11.7±1.7 | 24.7±0.6 | 22.7±1.0 | 12.6±0.5 | 32.5±0.2 | 39.9±0.2 | 2.4±0.2 | 29.9±0.4 | 33.1±0.1 | 28.5±0.4 |
| TENT+DART-split (ours) | 8.7±0.1 | 10.0±0.7 | 10.3±0.8 | 8.0±0.4 | 6.8±0.4 | 11.8±1.7 | 24.9±0.6 | 22.8±0.8 | 13.5±0.9 | 32.9±0.2 | 42.4±0.7 | 2.4±0.2 | 30.0±0.4 | 33.1±0.1 | 30.5±0.7 |
| SAR | 15.9±0.1 | 13.4±0.3 | 16.2±0.3 | 14.3±0.3 | 14.2±0.2 | 22.2±0.1 | 28.2±0.2 | 26.9±0.2 | 24.9±0.1 | 34.5±0.1 | 42.6±0.2 | 14.5±1.8 | 31.6±0.3 | 34.6±0.1 | 30.6±0.1 |
| SAR+DART-split (ours) | 15.7±0.2 | 13.1±0.4 | 16.0±0.2 | 14.3±0.3 | 14.1±0.2 | 22.2±0.1 | 28.2±0.2 | 26.9±0.2 | 24.9±0.1 | 34.6±0.1 | 44.3±0.7 | 14.5±1.8 | 31.3±0.4 | 34.6±0.2 | 30.8±0.6 |
| ODS | 14.8±0.3 | 14.9±1.1 | 17.3±0.9 | 14.1±0.4 | 12.1±0.9 | 20.4±2.5 | 40.5±0.6 | 36.7±1.2 | 22.2±0.9 | 51.0±0.2 | 61.3±0.2 | 4.5±0.3 | 47.6±0.4 | 51.8±0.2 | 45.3±0.5 |
| ODS+DART-split (ours) | 14.8±0.3 | 15.2±1.2 | 17.3±0.9 | 14.1±0.4 | 12.1±0.9 | 20.5±2.5 | 40.5±0.6 | 36.8±1.2 | 22.4±1.0 | 50.9±0.2 | 61.2±0.2 | 4.5±0.3 | 47.3±0.4 | 51.8±0.2 | 45.2±0.6 |

Table 30: Average accuracy (%) on ImageNet-C-imb (IR5000)

| Method | gaussian_noise | shot_noise | impulse_noise | defocus_blur | glass_blur | motion_blur | zoom_blur | snow | frost | fog | brightness | contrast | elastic_transform | pixelate | jpeg_compression |
|---|---|---|---|---|---|---|---|---|---|---|---|---|---|---|---|
| NoAdapt | 2.2±0.1 | 2.9±0.0 | 1.8±0.0 | 18.0±0.1 | 9.8±0.1 | 14.8±0.1 | 22.5±0.1 | 16.8±0.1 | 23.2±0.1 | 24.4±0.1 | 58.9±0.2 | 5.4±0.1 | 17.0±0.0 | 20.6±0.1 | 31.7±0.1 |
| BNAdapt | 4.4±0.0 | 4.6±0.0 | 4.5±0.1 | 3.9±0.1 | 3.9±0.1 | 6.8±0.1 | 9.7±0.1 | 9.5±0.1 | 9.7±0.0 | 12.8±0.1 | 17.6±0.1 | 4.5±0.1 | 11.1±0.1 | 12.3±0.1 | 10.6±0.1 |
| BNAdapt+DART-split (ours) | 4.2±0.1 | 4.9±0.3 | 4.2±0.2 | 3.9±0.1 | 3.9±0.1 | 7.2±0.2 | 12.7±0.7 | 12.5±0.9 | 14.2±0.8 | 20.1±1.2 | 31.5±1.9 | 4.5±0.1 | 16.5±0.9 | 14.0±0.4 | 18.0±1.3 |
| TENT | 2.1±0.1 | 2.2±0.2 | 2.4±0.2 | 1.8±0.2 | 1.6±0.1 | 1.9±0.0 | 5.3±0.3 | 2.6±0.1 | 2.9±0.2 | 7.9±0.4 | 11.9±0.3 | 0.6±0.1 | 6.1±0.4 | 7.7±0.3 | 6.9±0.2 |
| TENT+DART-split (ours) | 2.8±0.3 | 3.4±0.2 | 2.8±0.1 | 1.8±0.1 | 1.6±0.0 | 3.0±0.2 | 10.8±0.7 | 7.2±1.3 | 9.3±1.2 | 18.4±1.4 | 29.8±2.2 | 0.6±0.1 | 14.5±1.0 | 11.7±0.7 | 16.4±1.5 |
| SAR | 5.8±0.1 | 4.5±0.1 | 5.8±0.2 | 5.0±0.1 | 4.8±0.1 | 8.5±0.2 | 11.3±0.2 | 11.2±0.1 | 10.9±0.1 | 14.4±0.2 | 17.9±0.1 | 3.9±0.3 | 12.8±0.1 | 14.0±0.2 | 12.6±0.2 |
| SAR+DART-split (ours) | 5.3±0.1 | 5.4±0.4 | 5.1±0.1 | 4.9±0.1 | 4.7±0.1 | 8.4±0.3 | 13.4±0.7 | 13.1±0.8 | 14.6±0.9 | 20.9±1.2 | 31.6±1.9 | 3.9±0.3 | 17.0±0.9 | 15.1±0.6 | 18.6±1.3 |
| ODS | 6.4±0.3 | 6.5±0.4 | 7.2±0.3 | 5.7±0.4 | 4.8±0.2 | 6.3±0.1 | 18.9±1.2 | 8.2±0.4 | 9.1±0.5 | 28.0±1.1 | 41.9±0.7 | 1.8±0.2 | 22.4±1.1 | 28.1±0.8 | 24.0±0.9 |
| ODS+DART-split (ours) | 7.1±0.2 | 8.2±0.4 | 8.2±0.3 | 6.0±0.2 | 4.9±0.2 | 6.9±0.2 | 21.8±1.5 | 11.4±0.9 | 13.2±0.6 | 33.5±1.7 | 47.1±1.7 | 1.8±0.2 | 26.8±1.5 | 29.2±1.2 | 30.6±1.1 |

Table 31: Average accuracy (%) on ImageNet-C-imb (IR500000)

| Method | gaussian_noise | shot_noise | impulse_noise | defocus_blur | glass_blur | motion_blur | zoom_blur | snow | frost | fog | brightness | contrast | elastic_transform | pixelate | jpeg_compression |
|---|---|---|---|---|---|---|---|---|---|---|---|---|---|---|---|
| NoAdapt | 2.2±0.1 | 2.9±0.1 | 1.9±0.0 | 18.0±0.0 | 9.8±0.1 | 14.9±0.1 | 22.6±0.2 | 17.0±0.1 | 23.4±0.1 | 24.5±0.1 | 59.1±0.1 | 5.5±0.1 | 16.9±0.1 | 20.7±0.1 | 31.9±0.1 |
| BNAdapt | 2.1±0.0 | 2.1±0.0 | 2.1±0.0 | 1.6±0.1 | 1.6±0.1 | 2.8±0.1 | 3.7±0.1 | 3.9±0.1 | 4.4±0.0 | 5.1±0.1 | 6.7±0.1 | 2.1±0.0 | 4.2±0.0 | 4.5±0.1 | 4.3±0.1 |
| BNAdapt+DART-split (ours) | 2.4±0.1 | 3.5±0.3 | 2.5±0.3 | 1.6±0.1 | 1.6±0.0 | 4.5±0.2 | 9.9±0.8 | 9.7±0.6 | 11.4±0.9 | 15.4±1.2 | 22.1±1.4 | 2.1±0.0 | 12.0±0.9 | 10.9±0.7 | 12.7±1.0 |
| TENT | 0.8±0.0 | 0.8±0.1 | 0.8±0.1 | 0.6±0.0 | 0.6±0.0 | 0.7±0.1 | 1.6±0.0 | 0.9±0.1 | 1.0±0.0 | 2.1±0.1 | 3.2±0.1 | 0.3±0.0 | 1.6±0.1 | 1.8±0.1 | 1.8±0.1 |
| TENT+DART-split (ours) | 1.4±0.1 | 2.2±0.3 | 1.5±0.2 | 0.7±0.0 | 0.7±0.0 | 2.4±0.2 | 8.0±0.7 | 6.1±0.7 | 7.2±1.1 | 13.0±1.3 | 19.5±1.5 | 0.3±0.0 | 9.8±1.0 | 8.7±0.8 | 10.3±1.0 |
| SAR | 2.2±0.0 | 2.0±0.0 | 2.2±0.0 | 1.7±0.1 | 1.7±0.1 | 2.9±0.0 | 3.9±0.1 | 4.1±0.1 | 4.0±0.3 | 5.4±0.1 | 6.7±0.1 | 1.5±0.2 | 4.4±0.1 | 4.7±0.0 | 4.5±0.1 |
| SAR+DART-split (ours) | 2.6±0.2 | 3.7±0.3 | 2.7±0.3 | 1.8±0.1 | 1.7±0.1 | 4.6±0.3 | 10.1±0.8 | 9.8±0.6 | 11.4±1.0 | 15.8±1.3 | 22.0±1.5 | 1.5±0.2 | 12.1±0.9 | 11.2±0.8 | 13.0±1.1 |
| ODS | 4.0±0.0 | 4.1±0.4 | 4.4±0.3 | 3.5±0.2 | 3.3±0.0 | 3.7±0.3 | 12.4±0.3 | 4.8±0.3 | 5.6±0.1 | 17.5±0.4 | 28.9±0.7 | 1.2±0.0 | 13.2±0.6 | 16.4±0.7 | 13.0±0.6 |
| ODS+DART-split (ours) | 5.4±0.3 | 6.4±0.5 | 6.3±0.7 | 3.9±0.2 | 3.7±0.2 | 5.2±0.5 | 18.6±1.1 | 9.3±0.7 | 10.8±0.6 | 26.9±1.4 | 37.5±1.6 | 1.2±0.0 | 19.9±1.2 | 22.5±1.1 | 21.3±1.5 |

