# OpenReview forum: "Label Distribution Shift-Aware Prediction Refinement for Test-Time Adaptation"
_TMLR — Accepted by TMLR_

### Review · Reviewer_tTqA · 2024-11-25

**Summary Of Contributions:**

In this paper, the authors tackle the problem of TTA (Test-Time Adaptation) when there are severe class distribution shifts. They introduce a distribution-aware shift module that is trained using artificially created unbalanced batches of training images.
Their module can be used in combination with most existing TTA approaches, leading to consistent gains in accuracy.

**Audience:**

Yes

**Broader Impact Concerns:**

There are no broader impact concerns for this work.

**Claims And Evidence:**

Yes

**Requested Changes:**

I think the paper is ready to be published as is. There are already numerous experiments in the supplementary material that address most of the concerns that a reviewer could have.

**Strengths And Weaknesses:**

Strengths: The paper is clearly written, well motivated and contains convincing experiments on multiple datasets and combinations with many TTA methods. Multiple ablation studies confirm the importance of the various elements of the proposed solution.

Weaknesses: It is not a strong weakness, but since the observed degradation of performance when using BNAdapt is mostly due to severe shifts, the motivation can be seen as limited with limited expected gains in mild cases.

---

> ### Author Response · Authors · 2024-12-16
> **Response to Reviewer tTqA**
>
> > **W1: It is not a strong weakness, but since the observed degradation of performance when using BNAdapt is mostly due to severe shifts, the motivation can be seen as limited with limited expected gains in mild cases.**
>
> Thank you for your positive feedback. We emphasize that DART achieves significant performance gains in severe cases without performance degradation in mild cases. As shown in the CIFAR-10C-LT experiments in Table 1, DART achieves substantial performance improvements of 5.7% and 18.1% under severe shifts ($\rho=10/100$), while maintaining performance in balanced scenarios ($\rho=1$). This demonstrates DART's effectiveness under a wide range of test-time label distribution shifts.

---

### Review · Reviewer_oJqT · 2024-11-27

**Summary Of Contributions:**

The paper attempts to address performance degradation in test-time adaptation (TTA) when the test label distribution becomes im-balanced. The authors analyze the confusion matrix of TTA under label distribution shifts and highlight that inherent misclassification patterns contribute to poorer performance. They propose a metric to estimate the imbalance ratio of the test label distribution and introduce a new TTA method, DART, which leverages this metric and the original prediction distribution to refine the predictions. Empirical results on multiple benchmarks (CIFAR-10C, PACS, OfficeHome, and ImageNet) demonstrate significant improvements in accuracy for imbalanced test batches while maintaining stable performance when no label distribution shift occurs. Additionally, DART can easily be integrated with existing TTA methods designed for covariate shift.

**Audience:**

Yes

**Claims And Evidence:**

Yes

**Requested Changes:**

1. Add a symbol reference table.
2. It would be beneficial to include an illustration of the propositions in 'Impact of Label Distribution Shifts on TTA'
3. Experiment on the relationship between performance and training epochs.
4. Clarification on the imbalance ratio selection of label distribution during training and performance on unseen label distributions.

**Strengths And Weaknesses:**

#### Strengths

1. The paper addresses an important problem of test-time adaptation (TTA) under arbitrary label distributions.
2. The motivation behind the proposed method is clearly clarified both theoretically and experimentally, highlighting its ability to correct inherent classification confusion patterns caused by domain and label shifts.
3. The experimental results are robust, showing consistent improvements across multiple benchmarks, with stable gains over existing TTA methods.

#### Weaknesses

1. Section 2 of the paper is somewhat difficult to follow. It would be helpful if the authors could include a symbol reference at the beginning of Section 2. Additionally, regarding the theoretical results in 'Impact of Label Distribution Shifts on TTA', where data distributions are assumed to be two-dimensional Gaussian, it would be valuable to visualize some sampled instances and the Bayesian-optimal decision boundary in the 2D plane to directly demonstrate the inequalities in the conclusions.
2. There is a lack of some necessary analysis for the proposed method.
   1. **Convergence speed**: The method requires multiple training iterations (at least 50 epochs), increasing the computational load. Would reducing the number of training epochs lead to a significant drop in performance?
   2. **Generalization to unseen label distributions**: Can the method adapt to label distributions that were not seen during training, especially in highly imbalanced cases? For example, if the model is trained on distributions with an imbalance ratio below 50, can it generalize to a test set with a higher imbalance ratio, such as 100, and vice versa?

---

> ### Author Response · Authors · 2024-12-16
> **Response to Reviewer oJqT**
>
> > **W1. Section 2 of the paper is somewhat difficult to follow. It would be helpful if the authors could include a symbol reference at the beginning of Section 2. Additionally, regarding the theoretical results in 'Impact of Label Distribution Shifts on TTA', where data distributions are assumed to be two-dimensional Gaussian, it would be valuable to visualize some sampled instances and the Bayesian-optimal decision boundary in the 2D plane to directly demonstrate the inequalities in the conclusions.**
>
> As suggested by Reviewer oJqT, we have added a symbol table and a 2D illustration for the toy example to Appendix D.1.
>
>
> > **W2-1. Convergence speed: The method requires multiple training iterations (at least 50 epochs), increasing the computational load. Would reducing the number of training epochs lead to a significant drop in performance?**
>
> To analyze the sensitivity of the number of epochs for intermediate-time training, we conduct experiments on CIFAR-10C-LT with different numbers of epochs from 1 to 25, and summarize the results in Table R4. We can observe that increasing the number of epochs leads to better performance, but training for only 10 epochs achieves comparable results to 50 epochs, with just a 1.8% difference for $\rho=100$. This small gap highlights the efficiency of DART's intermediate-time training, demonstrating that it can achieve strong performance with minimal training epochs.
>
>
> **Table R4.  Performance (%) on CIFAR-10C-LT of DART-applied BNAdapt with different numbers of epochs for intermediate-time training**
>
> |  #epochs            | $\rho=1$ | $\rho=10$ | $\rho=100$ |
> |--------------|----------|-----------|------------|
> | 1            | 82.0     | 80.6      | 75.7       |
> | 5            | 85.3     | 84.5      | 81.7       |
> | 10           | 85.4     | 84.9      | 83.3       |
> | 25           | 85.3     | 84.8      | 84.8       |
> | 50           | 85.2     | 84.7      | 85.1       |
>
> > **W2-2. Generalization to unseen label distributions: Can the method adapt to label distributions that were not seen during training, especially in highly imbalanced cases? For example, if the model is trained on distributions with an imbalance ratio below 50, can it generalize to a test set with a higher imbalance ratio, such as 100, and vice versa?**
>
> DART aims to experience as many diverse class distributions as possible during intermediate time by using Dirichlet sampling for batch generation. As shown in Figure 7, Dirichlet-sampled batches cover a wide range of class distribution shifts. As suggested by Reviewer oJqT, we also consider a DART variant where the prediction refinement module experienced only three types of class distributions during intermediate time: uniform, long-tailed with $\rho = 20$, and inversely long-tailed. The results, summarized in Table 4, show that this variant effectively mitigates performance degradation on CIFAR-10C-LT. For example, with $\rho = 100$, the DART variant applied to BNAdapt achieves a 20.3% improvement compared to naive BNAdapt. However, the DART variant underperforms the original DART on CIFAR-10C-imb, where test-time label distributions are more diverse and severely imbalanced. These findings highlight the critical role of Dirichlet sampling in exposing the module to a wide range of class distributions during the intermediate time, which is crucial for robust generalization to unseen label distribution shifts.

---

> > ### Comment · Reviewer_oJqT · 2024-12-19
> > **Response to Authors**
> >
> > I appreciate the authors’ feedback. The new experimental results, figures, and clarifications have effectively addressed my concerns.

---

### Review · Reviewer_dzaf · 2024-12-03

**Summary Of Contributions:**

This paper analyzes Test-time adaptation (TTA) methods under label distribution shifts. Based on the analysis, this paper introduces label Distribution shift-Aware prediction Refinement for Test-time adaptation (DART), which refines the predictions by focusing on class-wise confusion patterns. DART trains a prediction refinement module that is intended to be used during test time to detect and correct class distribution shifts for better pseudo-label accuracy. Experimental results show that DART can correct inaccurate predictions caused by test-time distribution shifts.

**Audience:**

Yes

**Broader Impact Concerns:**

There is no concerns on the ethical implications.

**Claims And Evidence:**

Yes

**Requested Changes:**

Please address the questions in Cons and revise the manuscript accordingly.

**Strengths And Weaknesses:**

Pros:
+ The proposed label Distribution shift-Aware prediction Refinement for Test-time adaptation (DART) can be used to improve the TTA under the label distribution shift setting, broadening the application scenarios of the TTA strategies. It may interest some TMLR audiences.
+ The refinement module and the prediction deviation have some merits that may motivate follow-up researchers to design some new TTA methods.
+ The analysis using the toy example in Sec. 2 helps to illustrate the motivation of the proposed method, e.g., the prediction deviation metric, and how it works.
+ The experiments are extensive, with the ablation study to verify the effectiveness of the key components in DART, and further experimental analysis to justify the design of DART.
+ The overall performance is better than the other SOTA methods on various datasets.
+ Many implementation details and in-depth analysis (e.g., theoretical analysis) are provided in the Suppl., which facilitates the re-implementation and further understanding of the proposed method.

Cons:

1. In the Introduction, why can the model learn to adjust inaccurate predictions caused by label distribution shifts by experiencing several batches with diverse class distributions using the labeled training dataset before the start of test time? This statement is not easy to understand because the relationship between adjusting inaccurate predictions and the diverse class distributions is not presented. In addition, the necessity of doing it before the start of test time is not clarified.

2. Can we simply use some methods specific for long-tailed distribution to address the class distribution shift stated in the paper, e.g., balanced batch-level sampling? To answer this question, please also consider comparing with some long-tailed methods.

3. This paper adopts the balanced/diverse sampling strategy (IID or Dirichlet distribution) using training data. How much does this sampling strategy contribute to the performance (e.g., compared with removing IID or Dirichlet-based sampling)?

4. Why intermediate time rather than directly train it during the training phase? In Eq. (1), why are the parameters of the trained classifier fixed? These two questions may share the same answer.

5. What is the accuracy obtained by setting different $\alpha$ values?

6. Why not directly use DART-split as the standard pipeline for both the small-scale and large-scale datasets? What are the results of DART-split on small-scale datasets, like CIFAR-10-CLT, OfficeHome, and so on?

---

> ### Author Response · Authors · 2024-12-16
> **Response to Reviewer dzaf (1/3)**
>
> > **W1.  In the Introduction, why can the model learn to adjust inaccurate predictions caused by label distribution shifts by experiencing several batches with diverse class distributions using the labeled training dataset before the start of test time? This statement is not easy to understand because the relationship between adjusting inaccurate predictions and the diverse class distributions is not presented. In addition, the necessity of doing it before the start of test time is not clarified.**
>
> DART learns how to mitigate the performance degradation of BN-adapted classifiers caused by test-time label distribution shifts by utilizing labeled training data during an intermediate time, since addressing this degradation is challenging at test time without any labeled data.
> Figure 1 shows that BN-adapted classifiers exhibit class-wise confusion patterns caused by label distribution shifts, regardless of the presence or type of test corruption. To correct the inaccurate predictions, DART applies an affine transformation that reverses the effects of class-wise confusion, as in label noise training. However, inferring these patterns from unlabeled test data is challenging, as the type or severity of test-time label distribution shifts is unknown.
> To address this challenge, DART uses labeled training data to generate batches with diverse class distributions before the test time. Since the test-time label distribution shift is unpredictable before the test time and each shift can result in distinct class-wise confusion patterns, it is crucial to experience these variations as much as possible during intermediate time. These batches are used to train a prediction refinement module to adjust inaccurate predictions caused by label distribution shifts. The learned prediction adjustment technique can be effectively applied to any test corrupted test data. This is because the patterns caused by label distribution shifts remain consistent, regardless of test corruption types.
> In summary, DART improves the prediction accuracy of BN-adapted classifiers under test-time label distribution shifts by training on diverse class distributions during an intermediate phase, effectively mitigating the performance degradation.
>
>
> > **W2. Can we simply use some methods specific for long-tailed distribution to address the class distribution shift stated in the paper, e.g., balanced batch-level sampling? To answer this question, please also consider comparing with some long-tailed methods.**
>
> In TTA setups, where labels are unavailable, long-tail learning-based methods rely on predictions as pseudo-labels, making them vulnerable to performance degradation under label distribution shifts. However, DART can be integrated as a plug-in with these methods to correct inaccurate predictions and mitigate this issue directly.
> Long-tail learning methods usually address class imbalance through re-weighting or re-sampling. Re-weighting assigns higher weights to infrequent classes and lower weights to frequent ones, while re-sampling adjusts the class distribution by oversampling minor classes or undersampling major ones.
> As described in Section 4 and Appendix A.5, ODS and DELTA are re-weighting methods that monitor the frequency of each class during test time and dynamically adjust their weights, effectively mitigating the dominance of frequent classes while ensuring better representation for infrequent ones. On the other hand, NOTE is a re-sampling approach using a prediction-balancing memory to maintain an equal number of test samples per class for fine-tuning. As shown in Table 1 of the manuscript, these methods outperform other TTA approaches like TENT and SAR under label distribution shift scenarios for most cases.
> Despite their effectiveness, these methods depend on predictions, making them vulnerable to degradation caused by inaccurate pseudo-labels under label distribution shifts. DART, by directly correcting BNAdapt's performance, can be integrated with these methods to enhance overall performance under such scenarios significantly.

---

> ### Author Response · Authors · 2024-12-16
> **Response to Reviewer dzaf (2/3)**
>
> > **W3. This paper adopts the balanced/diverse sampling strategy (IID or Dirichlet distribution) using training data. How much does this sampling strategy contribute to the performance (e.g., compared with removing IID or Dirichlet-based sampling)?**
>
> Batches sampled from the Dirichlet distribution exhibit diverse class distributions, as shown in Figures 6 and 7 of Appendix A. Using these batches, the prediction refinement module, $g_\phi$, which learns to detect label distribution shifts and mitigate the performance degradation of BN-adapted classifiers (Eq. (1)). However, exposure to class-imbalanced batches from Dirichlet sampling can cause $g_\phi$ to become biased towards such scenarios, leading to unnecessary modifications when processing nearly class-balanced batches. In the absence of label distribution shifts, no significant performance degradation or consistent class-wise confusion patterns are observed across test corruptions. In such cases, unnecessary adjustments can negatively impact TTA performance. To address this issue, we introduce a regularization term (Eq. (2)) to prevent $g_\phi$ from modifying predictions for nearly class-balanced batches.
>
> To evaluate the impact of each loss term (Eq. (1) and (2)), we summarize the experimental results in Table R1 by removing each term individually. Without Eq. (1) (with Dirichlet sampled batches), the prediction refinement module fails to improve the degraded performance of BNAdapt under label distribution shifts. For instance, DART-applied BNAdapt and TENT perform the same as the original BNAdapt and TENT without Dirichlet sampling, emphasizing the importance of experiencing diverse label distributions. On the other hand, without Eq. (2) (with IID sampled batches), the module performs unnecessary prediction refinement on class-balanced batches, which degrades TTA performance. For example, while DART does not reduce the performance of BNAdapt, it decreases TENT’s performance by 2.1% on CIFAR-10C-LT with $\rho=1$. This highlights the need for regularization to avoid unnecessary adjustments.
> To better understand these results, we measure the Expected Calibration Error (ECE) for both BNAdapt and DART-applied BNAdapt, with results summarized in Table R2. ECE quantifies the discrepancy between a model’s confidence and its accuracy, with lower values indicating better-calibrated predictions. Recent TTA research [1, 2] emphasizes the importance of prediction calibration, showing that well-calibrated predictions improve TTA performance. Table R2 demonstrates that without regularization (Eq. (2)), the ECE increases after prediction refinement when $\rho=1$, indicating degraded prediction calibration and reduced TTA performance. In contrast, with regularization, the ECE remains stable, preventing performance degradation.
> In summary, both loss terms, leveraging different sampling strategies, are critical for effective prediction refinement in DART under test-time label distribution shifts.
>
> **Table R1. Ablation studies highlighting the Importance of IID and Dirichlet sampling for prediction refinement of DART on CIFAR-10C-LT. We report the average accuracy (%) over 15 common corruptions.**
> | Method       | Dirichlet (Eq. (1)) | IID (Eq. (2))  | $\rho=1$ | $\rho=10$ | $\rho=100$  |
> |--------------|---------------------|----------------|----------|-----------|-------------|
> | BNAdapt      |                     |                | 85.2     | 79.0      | 67.0        |
> | BNAdapt+DART | $\checkmark$                   |                | 85.3     | 85.5      | 85.8        |
> |              |                     | $\checkmark$              | 85.2     | 79.0      | 67.0        |
> |              | $\checkmark$                   | $\checkmark$              | 85.2     | 84.7      | 85.1        |
> | TENT         |                     |                | 86.3     | 82.9      | 70.4        |
> | TENT+DART    | $\checkmark$                   |                | 84.4     | 86.2      | 88.3        |
> |              |                     | $\checkmark$              | 86.3     | 83.0      | 70.2        |
> |              | $\checkmark$                   | $\checkmark$              | 86.5     | 86.7      | 88.2        |
>
>
> **Table R2. Expected Calibration Error (ECE) (%) of BNAdapt and DART-applied BNAdapt on CIFAR-10C-LT. A lower ECE indicates better calibration.**
>
> |   Method              | $\rho=1$ | $\rho=10$ | $\rho=100$ |
> |-----------------------------|----------|-----------|------------|
> | BNAadapt                    | 8.7      | 12.6      | 21.9       |
> | BNAadapt+DART               | 8.8      | 7.8       | 6.4        |
> | BNAadapt+DART (w/o Eq. (2)) | 12.0     | 10.6      | 7.5        |
>
> [1] Tan et al., “Uncertainty-Calibrated Test-Time Model Adaptation without Forgetting”, ArXiv 2024
>
> [2] Yang et al., "Towards Test Time Adaptation via Calibrated Entropy Minimization", KDD 2024

---

> ### Author Response · Authors · 2024-12-16
> **Response to Reviewer dzaf (3/3)**
>
> > **W4. Why intermediate time rather than directly train it during the training phase? In Eq. (1), why are the parameters of the trained classifier fixed? These two questions may share the same answer.**
>
> The prediction refinement module, $g_\phi$, aims to adjust the outputs of the BN-adapted classifier degraded under label distribution shifts. Thus, we fix all model parameters of the pre-trained classifier except for the BN statistics during intermediate-time training.
> As suggested by Reviewer dzaf, it is also possible to fine-tune the classifier's parameters during $g_\phi$. However, this may lead to instability since the classifier outputs continuously change. Additionally, fine-tuning the classifier using outputs of a poorly trained $g_\phi$ could amplify this instability. Thus, freezing the classifier’s parameters during $g_\phi$ training ensures stable and effective learning.
>
> > **W5. What is the accuracy obtained by setting different $\alpha$ values?**
>
> $\alpha$ is a hyperparameter that balances the two loss terms (Eq. (1) and (2)) during the training of $g_\phi$. To evaluate the sensitivity of $\alpha$, we conduct experiments on CIFAR-10C-LT with $\alpha$ values ranging from 0.01 to 1, as summarized in Table R3.
> For example, at $\rho = 100$, DART-applied BNAdapt shows only a 0.4% lower accuracy when $\alpha = 1$ compared to $\alpha= 0.1$, which is negligible relative to the overall 18% performance gain. In contrast, when $\alpha = 0$ (*i.e.*, IID-sampled batches are not used for $g_\phi$ training, as discussed in response to W2 of Reviewer dzaf), unnecessary prediction refinement on class-balanced batches negatively impacts TTA performance. For instance, DART-applied TENT shows a 2.1% performance drop on CIFAR-10C-LT with $\rho = 1$ in Table R1.
> These results demonstrate that DART is robust to changes in $\alpha$ as long as $\alpha> 0$.
>
>
> **Table R3. Accuracy (%) of DART-applied BNAdapt and TENT with different $\alpha$ values on CIFAR-10C-LT.**
>
> | Method    | $\alpha$ | $\rho=1$   | $\rho=10$   | $\rho=100$  |
> |------------------|----------|-----|------|------|
> | BNAadapt+DART    | 0.01     | 85.4 | 85.1 | 85.7 |
> |                  | 0.1 (used)| 85.2 | 84.7 | 85.1 |
> |                  | 0.5      | 85.3 | 83.9 | 84.8 |
> |                  | 1        | 85.3 | 83.5 | 84.7 |
> | TENT+DART        | 0.01     | 86.1 | 86.9 | 88.7 |
> |                  | 0.1 (used)| 86.5 | 86.7 | 88.2 |
> |                  | 0.5      | 86.6 | 86.5 | 87.9 |
> |                  | 1      | 86.5 | 86.2 | 87.8 |
>
> > **W6. Why not directly use DART-split as the standard pipeline for both the small-scale and large-scale datasets? What are the results of DART-split on small-scale datasets, like CIFAR-10-CLT, OfficeHome, and so on?**
>
> In Appendix B.2, we compare the performance of DART and DART-split on CIFAR-10C-LT and discuss their applicability based on the benchmark scale.
> We recommend using the original DART for small to mid-scale benchmarks and DART-split for large-scale benchmarks. The key difference between the two lies in the structure of the prediction refinement module $g_\phi$. DART employs a single module to manage both shift detection and affine transformation for each batch, while DART-split separates these tasks into two modules: one for detecting label distribution shifts and another for generating affine transformations.
> For small-scale datasets like CIFAR-10C-LT, DART is more effective, outperforming DART-split across all $\rho$ values, as shown in Table 7. Although DART-split achieves a notable 10% improvement under severe shifts ($\rho = 100$) and performs comparably to naive BNAdapt under lower shifts ($\rho = 1$ or $10$), DART consistently shows superior performance. On the other hand, DART-split is better suited for large-scale datasets, as demonstrated in Table 2. Its modular design efficiently handles the complexity and scale of large datasets. Thus, we recommend applying the original DART to small to mid-scale benchmarks and DART-split to large-scale benchmarks.

---

> > ### Comment · Reviewer_dzaf · 2024-12-16
> > **My concerns have been addressed**
> >
> > Many thanks for the response. My concerns have been addressed, so I suggest an acceptance.

---

### Author Response · Authors · 2024-12-16
**The revised paper is uploaded**

We upload our revised paper with the following modifications:
* (Appendix D) A symbol table (Table 9) and an illustration (Figure 8) for the toy example described in Section 2 are added
* (Appendix A, Figure 7) Examples of class distributions sampled from the Dirichlet distribution are added
* (Appendix F) Experimental results analyzing the effect of varying $\alpha$  are added
* (Appendix G.4) Experimental results examining the impact of changing the number of training epochs for $g_\phi$ are added

All changes in the revised paper are marked in blue.

---

### Decision · Action_Editor_mJXk · 2025-01-21

**Recommendation:** Accept with minor revision

**Comment:**

Based on the above comments and the consensus among the reviewers, I would recommend the paper to be accepted.

However, it is still necessary to include some discussions regarding the feasibility of the proposed method when it applies to large-scale models and datasets as well as when the labeled training dataset is not provided even in a form of condensed data. Especially, for the former case, the experimental results on ImageNet are provided in Table 2, however, the proposed TTA method seems to perform worse than no adaptation when the imbalance ratios are high.

**Audience:**

TTA under label distribution shift would be an interesting topic, and especially the dynamic adaptation to diverse shifts based on the trained prediction refinement module can receive a lot of interest from the community including TMLR's audience.

**Claims And Evidence:**

This paper proposes a novel prediction refinement module for test-time adaptation especially when the label distribution shift occurs. In particular, the proposed refinement module uses both average pseudo-label distributions and prediction variances for each test batch obtained from the BN-adapted classifier as inputs and predicts affine transformation parameters for the refinement of the original prediction like reversing the effects of class-wise confusion. And, this refinement module is trained using the labeled training dataset with a simulation of diverse shifts by Dirichlet sampling.

Overall, the proposed method seems to be sound with clear motivations. Moreover, extensive experiments on various datasets and settings sufficiently support consistent improvements of the proposed method over the baseline TTA methods including the benefits of each component.